# Tumor-activated in situ synthesis of single-atom catalysts for $O_2$-independent photodynamic therapy based on water-splitting

Yiyan Yin[1], Xiyang Ge[1], Jin Ouyang[2] & Na Na [1] ✉

Single-atom catalysts (SACs) have attracted interest in photodynamic therapy (PDT), while they are normally limited by the side effects on normal tissues and the interference from the Tumor Microenvironment (TME). Here we show a TME-activated in situ synthesis of SACs for efficient tumor-specific water-based PDT. Upon reduction by upregulated GSH in TME, $C_3N_4$-Mn SACs are obtained in TME with Mn atomically coordinated into the cavity of $C_3N_4$ nanosheets. This in situ synthesis overcomes toxicity from random distribution and catalyst release in healthy tissues. Based on the Ligand-to-Metal charge transfer (LMCT) process, $C_3N_4$-Mn SACs exhibit enhanced absorption in the red-light region. Thereby, a water-splitting process is induced by $C_3N_4$-Mn SACs under 660 nm irradiation, which initiates the $O_2$-independent generation of highly toxic hydroxyl radical (·OH) for cancer-specific PDT. Subsequently, the ·OH-initiated lipid peroxidation process is demonstrated to devote effective cancer cell death. The in situ synthesized SACs facilitate the precise cancer-specific conversion of inert $H_2O$ to reactive ·OH, which facilitates efficient cancer therapy in female mice. This strategy achieves efficient and precise cancer therapy, not only avoiding the side effects on normal tissues but also overcoming tumor hypoxia.

Photodynamic therapy (PDT) has exhibited substantial advantages for its spatiotemporal selectivity, minimal invasiveness, and low biotoxicity in cancer therapy[1,2]. To induce the death of cancer cells, efficient photosensitizers (PSs) are aspired for generating adequate reactive oxygen species (ROS), such as superoxide, hydroxyl radical (·OH), singlet oxygen, etc[1,3]. Meanwhile, single-atom catalysts (SACs) have recently received enormous interest for unique electronic structures, such as well-defined and precisely situated metal centers, identical coordination environments, tailorable compositions, and versatile functionalities[4,5]. Especially, SACs are conducive to the catalytic generation of ROS, benefiting from the coordination environments of isolated single atoms[6,7]. For this reason, SACs promise to become unique nanomedicines with excellent catalytic activities for cancer therapy, with maximized metal utilization and minimized biotoxicity[8,9]. However, the therapeutic efficiency of SACs is still challenged by the difficulties of efficient and controllable delivery into tumors while maintaining high catalytic activities across biological systems. Thus, an improved SACs-based strategy is required for cancer-specific PDT with maximized therapeutic efficacy and minimal side effects on normal tissues.

In situ synthesis of nanomedicines inside specific cells has been aroused for chemotherapy, overcoming random distributions of nanomedicines in healthy tissue to minimize side effects[10–12]. Recently, small molecular PSs have also been reported to be in situ synthesized

[1]Key Laboratory of Radiopharmaceuticals, Ministry of Education, College of Chemistry, Beijing Normal University, Beijing 100875, China. [2]Department of Chemistry, College of Arts and Sciences, Beijing Normal University at Zhuhai, Zhuhai 519087, China. ✉e-mail: nana@bnu.edu.cn

from precursors for anticancer therapy[12]. However, in situ synthesis of SACs is still challenged by the difficulties of specific release of metals and ligands within the cancer sites. Moreover, the complex Tumor Microenvironment (TME), with variable acidic environments, hypoxic conditions, and redox species, would lead to poor stabilities and unpredictable side reactions during the in situ synthesis of SACs[13]. Therefore, the cancer-specific in situ synthesis of SACs and efficient therapy without interference by TME remains challenging.

Particularly, the therapeutic efficiency of ROS-dependent PDT is significantly influenced by tumor hypoxia, limiting the generation of therapeutic ROS reagents without sufficient $O_2$[14]. This is even significant in the Type-II PDT with $O_2$-related ROS (like $^1O_2$) generated by $O_2$-dependent PSs. Alternatively, Type-I PDT could relieve the dependence on intracellular $O_2$ by generating ·OH through the Fenton-like oxidation of $H_2O_2$[15,16]. However, it was still limited by the insufficient endogenous $H_2O_2$ at tumor sites[17]. Therefore, generating adequate ROS species independent of endogenous $O_2$ and $H_2O_2$ would considerably increase the PDT efficiency in TME. Consequently, a necessity for $O_2$-independent generation of ROS via the in situ synthesized SACs in cancer sites arises. This would become more desired with water as the simple oxygen source for efficient PDT across tumors.

In this work, a single-atom catalyst ($C_3N_4$-Mn SACs) is in situ prepared in TME for the highly specific Type-I PDT upon the oxygen-independent generation of ·OH with minimal invasiveness. Firstly, a nanomedicine precursor of 2D/2D $C_3N_4$-$MnO_2$ is prepared, which is lowly toxic to normal tissues (such as the kidney and liver) in the metabolism (Fig. 1). To implement the in situ synthesis and oxygen-independent generation of ·OH, $C_3N_4$-$MnO_2$ is enriched at the tumor site through the enhanced permeability and retention (EPR) effect. Subsequently, the in situ release of $C_3N_4$ and $Mn^{2+}$ from $C_3N_4$-$MnO_2$ is employed, which responds to the upregulated GSH in TME. Consequently, atomically dispersed $Mn^{2+}$ is captured by $C_3N_4$, facilitating the in situ synthesis of $C_3N_4$-Mn SACs within TME. Especially based on the

Ligand-to-Metal charge transfer (LMCT) process from $C_3N_4$ to $Mn^{2+}$, $C_3N_4$-Mn SACs exhibit enhanced absorption in the red region for efficient $O_2$-independent generation of ·OH via water-splitting. The red shift of the absorption also facilitates the red light-irradiated applications (660 nm) with greater penetration depth than that of white light in PDT[18]. Thereby, being the most toxic ROS, adequate ·OH induces efficient cancer cell death through the powerful Lipid Peroxidation (LPO) process. Furthermore, PDT mechanisms are studied by extracellular and intracellular tests, and the in vivo therapeutic effects are further validated. Therefore, this work can inspire efficient, tumor-specific, and $O_2$-independent PDT through the in situ synthesis of SACs.

## Results and discussion

### SACs synthesis via degradation of $C_3N_4$-$MnO_2$ precursors in TME

The tumor-specific synthesis of SACs was employed through the in situ degradation of precursors within TME (Fig. 2a). Initially, the nanomedicine precursor of 2D/2D $C_3N_4$-$MnO_2$ nanocomposite was prepared by depositing $MnO_2$ sediment on the surface of the as-prepared g-$C_3N_4$ nanosheet according to the reported methods (Supplementary Figs. 1, 2, 3, 4, Eq. 1)[19,20]. The successful preparation of g-$C_3N_4$ nanosheets (Supplementary Fig. 2) and the deposition of $MnO_2$ on $C_3N_4$ (Supplementary Figs. 3 and 4) were confirmed by Transmission Electron Microscopy (TEM) and spectroscopic characterizations. This was also in accordance with the observing of the lattice fringe of $MnO_2$ on $C_3N_4$, with a lattice stripe spacing of ~0.31 nm (Fig. 2b). With $MnO_2$ deposited on $C_3N_4$, the thickness of the nanosheets increases from ~1 nm to ~2 nm (Fig. 2c). These morphological features of $C_3N_4$-$MnO_2$ precursors were altered during the degradation by endogenous glutathione (GSH) in TME (Eq. 2). As evidenced by the in vitro studies in TME simulated conditions, the suspension of $C_3N_4$-$MnO_2$ gradually changed from black to white within 5 min after GSH was added, demonstrating the quick destruction of $C_3N_4$-$MnO_2$ precursors by GSH (Supplementary

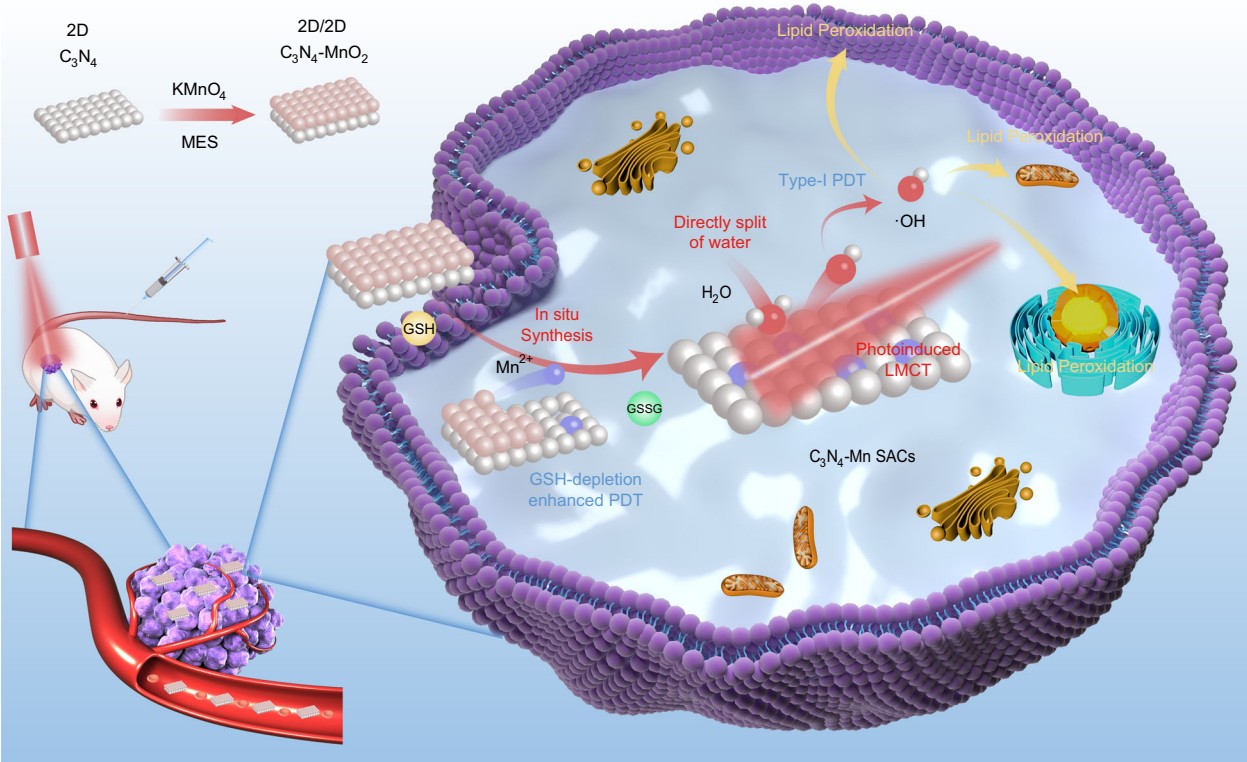

**Fig. 1 | Schematic diagram of in situ synthesis of $C_3N_4$-Mn SACs and applications in $O_2$-independent type-I PDT.** Different colors of balls represented $C_3N_4$ (white), $MnO_2$ (pink), and $Mn^{2+}$ (purple), respectively.

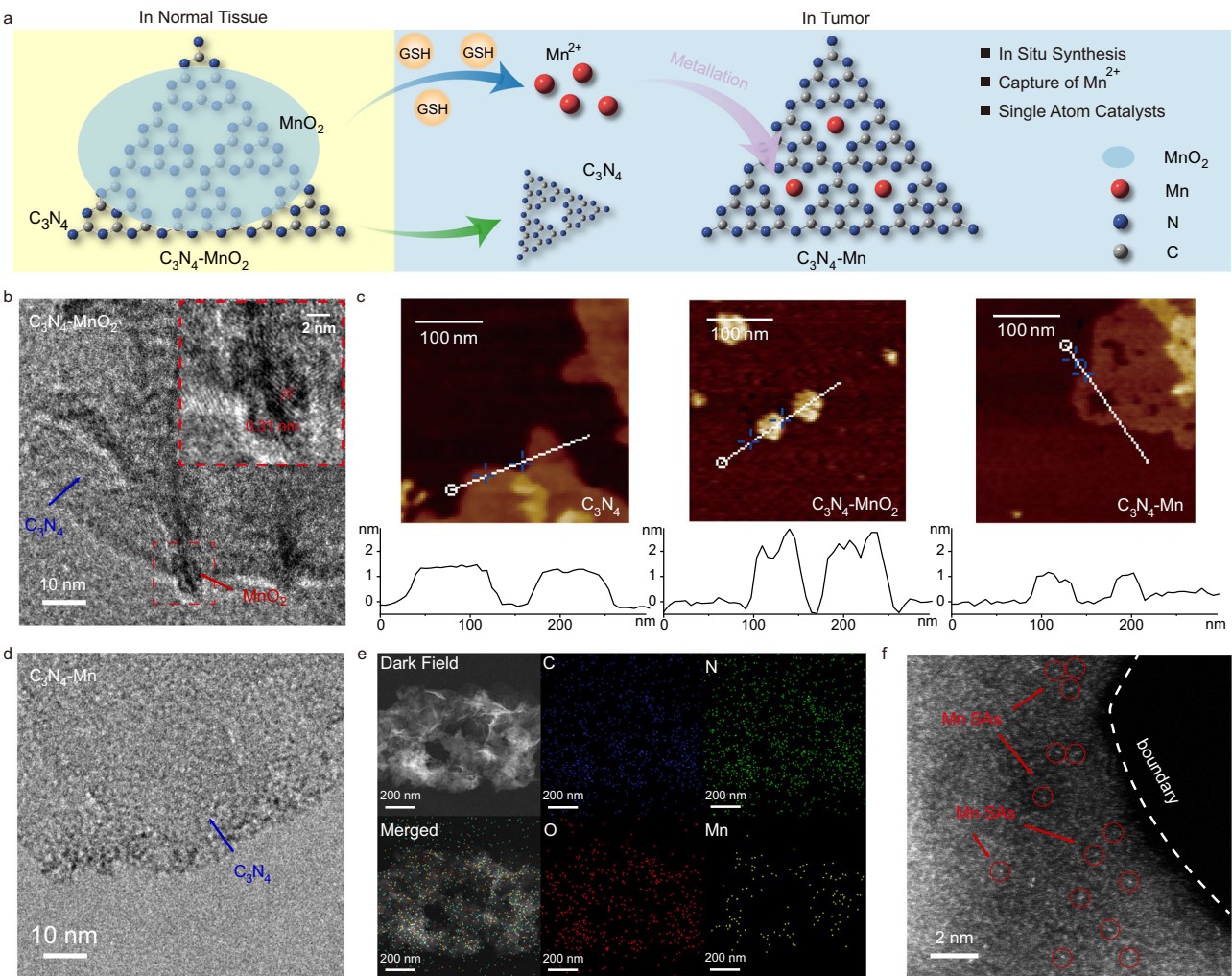

**Fig. 2 | The synthesis of C₃N₄-Mn SACs by the reduction of C₃N₄-MnO₂ by GSH and the morphology characterizations. a** The mechanism illustration. MnO₂ was reduced into Mn²⁺ by GSH, and Mn²⁺ was captured by C₃N₄ nanosheets via the coordination into the N₆-cavity. **b** TEM images of C₃N₄-MnO₂. **c** AFM images and the heights of C₃N₄ nanosheets, C₃N₄-MnO₂ precursor, and C₃N₄-Mn SACs. **d** TEM images of C₃N₄-Mn SACs. **e** EDS mapping of C₃N₄-Mn SACs. **f** HAADF-STEM image of C₃N₄-Mn SACs. The atomically dispersed Mn was highlighted by red circles.

Figs. 5, 6, 7). This was also in accordance with the considerable recovery of fluorescence (FL) after the degradation of MnO₂ by GSH, which was once quenched by MnO₂ in C₃N₄-MnO₂ (Supplementary Fig. 8). In addition, the regained thickness from ~2 nm (C₃N₄-MnO₂) to ~1 nm (C₃N₄-Mn) (Fig. 2c) and the absence of the MnO₂ stripe (Fig. 2d) further confirmed the depletion of MnO₂ by GSH.

Interestingly, no free Mn²⁺ was detected in the supernatant after the reduction of C₃N₄-MnO₂ (Supplementary Fig. 9), and the Mn species were still present in the sample (demonstrated by elemental maps in Fig. 2e). The actual Mn content was 2.34 wt%, determined by Inductively Coupled Plasma Optical Emission Spectroscopy (ICP-OES) detections. This revealed that Mn²⁺ was probably captured by C₃N₄ to form C₃N₄-Mn. That is to say, C₃N₄ was simultaneously released and coordinated with Mn²⁺ together with the degradation of MnO₂ (Fig. 2a, Eq. 3). This was also in accordance with the increase in ζ potential from −18.9 mV to −14.2 mV due to the capture of Mn²⁺ by C₃N₄ (Supplementary Fig. 10). In fact, this increased ζ potential upon Mn²⁺ coordination decreased the electrostatic repulsion of Mn²⁺, which would somehow diminish the dispersibility of the original g-C₃N₄. Although some precipitates in the prepared C₃N₄-Mn solution were observed after 24 h (Supplementary Fig. 11), the dispersibility can still support the subsequent biological applications in several hours. Therefore, the C₃N₄-MnO₂ precursors were decomposed in the presence of GSH, and

the generated Mn²⁺ could be captured by C₃N₄ to obtain the C₃N₄-Mn SACs. The atomically dispersed Mn in C₃N₄-Mn nanosheets was further visualized by the high-angle annular dark field-scanning transmission electron microscopy (HAADF-STEM) (Fig. 2f). This would be beneficial to the tumor-specific in situ synthesis of SACs for subsequent photodynamic therapy.

$$C_3N_4 + MnO_4^- \xrightarrow{\text{MES buffer, pH = 6.0}} C_3N_4 - MnO_2 \qquad (1)$$

$$C_3N_4 - MnO_2 + 4GSH \rightarrow C_3N_4 + Mn^{2+} + 2GSSG + 2H_2O \qquad (2)$$

$$C_3N_4 + Mn^{2+} \xrightarrow{\text{very fast}} C_3N_4 - Mn \qquad (3)$$

Furthermore, the chemical structure of C₃N₄-Mn SACs was examined by a series of chemical characterizations. Compared with the Fourier Transform Infrared spectroscopy (FTIR) spectrum of C₃N₄, the identical peaks of tri-s-triazine units (805 cm⁻¹) and aromatic CN heterocycles (1200–1800 cm⁻¹) indicated the maintenance of C₃N₄ structure in C₃N₄-MnO₂ and C₃N₄-Mn SACs (Fig. 3a)[21–23]. Simultaneously, the Mn-O bands (499 cm⁻¹ and 441 cm⁻¹) in C₃N₄-MnO₂ were not observed in the FTIR spectrum of C₃N₄-Mn SACs, demonstrating

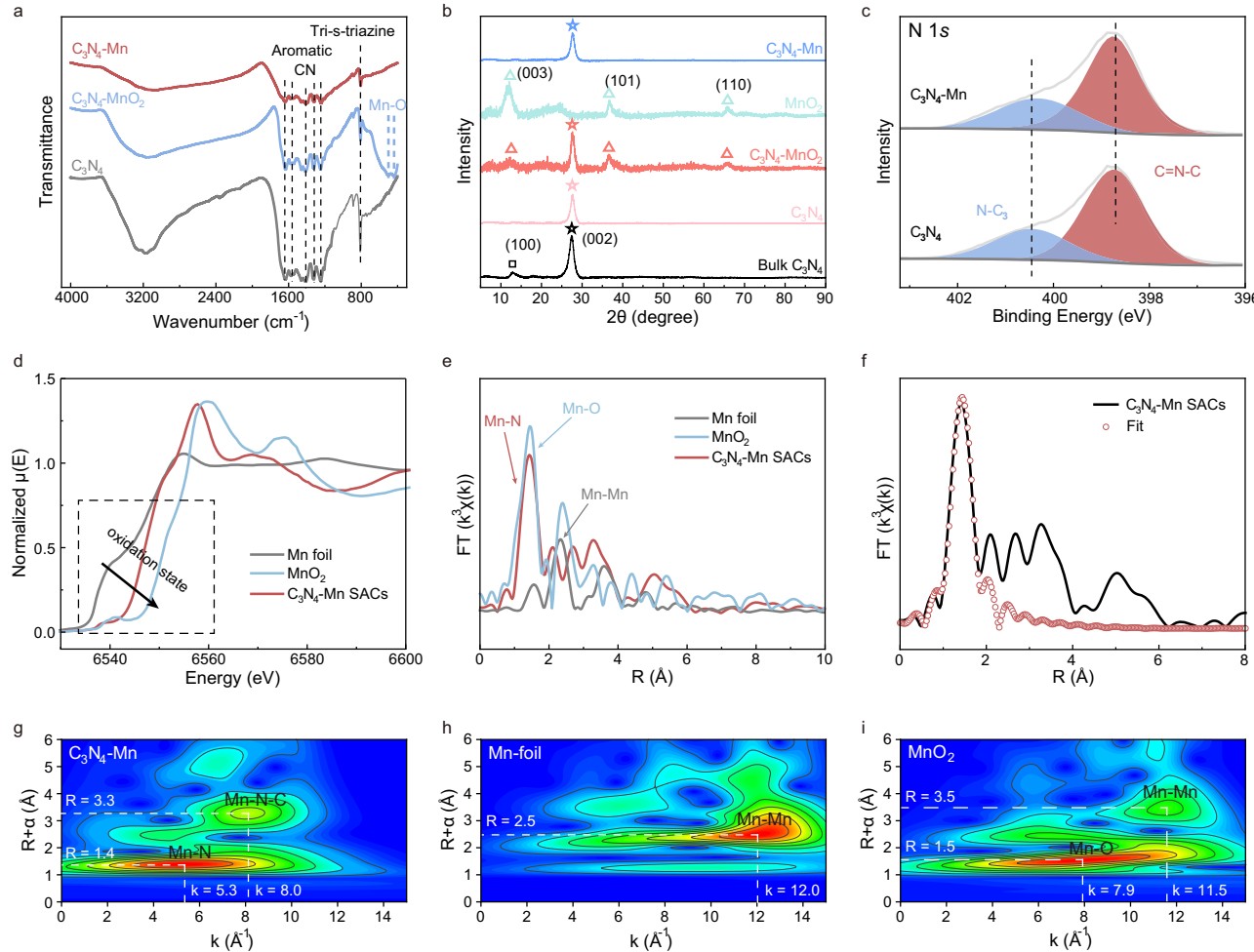

**Fig. 3 | Fine structure of Mn atom in C₃N₄-Mn SACs. a** FTIR spectra of g-C₃N₄ nanosheets, C₃N₄-MnO₂ precursor, and C₃N₄-Mn SACs. **b** XRD patterns of bulk C₃N₄, g-C₃N₄, C₃N₄-MnO₂, MnO₂, and C₃N₄-Mn SACs. **c** N 1s XPS spectra of C₃N₄-Mn SACs and g-C₃N₄. **d** XANES and (**e**) EXAFS spectra of C₃N₄-Mn SACs, MnO₂, and Mn foil at the Mn K-edge. **f** EXAFS fitting curves of C₃N₄-Mn SACs in R space. WT of (**g**) C₃N₄-Mn SACs, (**h**) Mn foil, and (**i**) MnO₂.

the successful reduction of MnO₂[22]. This was also demonstrated by the X-Ray Diffraction (XRD) patterns. As illustrated in Fig. 3b, the disappearance of the (100) peak at 13.4° in g-C₃N₄, C₃N₄-MnO₂, and C₃N₄-Mn showed the conversion of bulk phase to the 2D structure of C₃N₄[24]. Furthermore, the peaks of MnO₂ (12.3°, 36.7°, 65.8°, JCPDS 01-074-7889)[25] were not observed and no noticeable alteration of C₃N₄ (002) peak (27.6°) in C₃N₄-Mn SACs was recorded. This indicated that MnO₂ was consumed by GSH and the 2D structure of C₃N₄ was retained in C₃N₄-Mn SACs[22]. Additionally, the significant distinctive Raman peaks (278, 340, 421, 495 cm⁻¹) were identified in C₃N₄-Mn SACs, rather than the peaks of Mn-O bonds in C₃N₄-MnO₂ (563 cm⁻¹ and 644 cm⁻¹) (Supplementary Fig. 12)[21,26]. Therefore, after being reduced by GSH, Mn²⁺ was captured by C₃N₄ without damage to the structural units of C₃N₄.

To further investigate the interaction between Mn and C₃N₄, the X-ray Photoelectron Spectroscopy (XPS) analysis of C₃N₄-Mn, C₃N₄-MnO₂, and C₃N₄ was employed. As illustrated in Supplementary Fig. 13, the characteristic peak of Mn was observed in C₃N₄-MnO₂ and C₃N₄-Mn but was absent in C₃N₄. In addition, the electron transfer from C₃N₄ to MnO₂ was demonstrated based on the XPS characterizations, and particular peaks of Mn(II) (Mn 2p₁/₂ at 653.8 eV, Mn 2p₃/₂ at 641.5 eV) were observed in C₃N₄-Mn SACs[27]. This revealed that the valence state of Mn was not changed after being captured by C₃N₄. Moreover, after coordinating with Mn, the sp²-hybridized N peak of C₃N₄ (C-N=C, at 398.7 eV) shifted to high binding energies (Fig. 3c), which might be

attributed to the electron attraction by Mn²⁺[21]. Meanwhile, no noticeable shift was recorded in C 1s (288.2 eV) (Supplementary Fig. 13) and N-C₃ (400.5 eV) (Fig. 3c). This revealed that Mn²⁺ was coordinated with sp²-bonded N atoms rather than tertiary N or C atoms[21,22]. Besides, demonstrated by the O 1s XPS spectra (Supplementary Fig. 13), the observed O species in the EDS mapping of C₃N₄-Mn (Fig. 2e) were attributed to the absorbed oxygen of C₃N₄. In addition, the absent oxygen signal of metal oxides in C₃N₄-Mn SACs (Supplementary Fig. 13) further confirmed the in situ synthesis of C₃N₄-Mn SACs upon the reduction of MnO₂ into Mn²⁺. As reported, C₃N₄ might act as a ligand to capture, chelate, and fix metal ions within the center of the N₆-macroheterocycle[22]. Therefore, Mn²⁺ was caught in the N₆-macroheterocycle cavity of C₃N₄ by coordinating with sp²-hybridized N in C₃N₄.

To identify the local fine structure of Mn at the atomic level, C₃N₄-Mn SACs were further examined by X-ray Absorption Spectroscopy (XAS). As demonstrated by spectra of Mn K-edge X-ray Absorption Near-Edge Structure (XANES, Fig. 3d), the absorption edge of Mn in C₃N₄-Mn SACs placed between that of Mn foil and MnO₂, showing that the valence state of Mn was close to +2. The coordination between Mn and N in C₃N₄-Mn was demonstrated by the phase-uncorrected Fourier-transformed Extended X-ray Absorption Fine Structure (EXAFS) characterization. As shown in Fig. 3e, Mn foil presented the main peak at 2.3 Å, corresponding to the Mn-Mn bonds. While no corresponding Mn-Mn signal was observed in C₃N₄-Mn SACs, indicating the presence

of atomically dispersed Mn. Furthermore, the signal at 1.4 Å could be ascribed to the Mn-N bonds, which was in accordance with N 1s XPS spectra (Fig. 3c and Supplementary Fig. 13). To accurately characterize the coordination structure of the Mn atom, quantitative EXAFS curve fitting analysis in R spaces was employed. As depicted in Fig. 3f, the best-fitting results demonstrated that the peak of about 1.97 Å was ascribed to the coordination of Mn single atoms with N atoms in $C_3N_4$-Mn SACs (Supplementary Table 1). The Wavelet Transform (WT) results further indicated that there was no Mn-Mn bond in $C_3N_4$-Mn (Fig. 3g, h, i). The main peak of $C_3N_4$-Mn tended to have a lower k value ($-5.3$ Å$^{-1}$) than that of Mn-O in $MnO_2$ ($-8.0$ Å$^{-1}$), confirming the presence of Mn-N bond in $C_3N_4$-Mn SACs. In addition, the peaks of Mn-N-C with long lengths ($-3.3$ Å) were observed, which could be attributed to the long-range interaction of Mn-N-C in $C_3N_4$-Mn SACs. Therefore, the detailed Mn coordination by N in $C_3N_4$-Mn was strongly confirmed by the XAFS tests.

To further define the structure of atomically dispersed Mn in $C_3N_4$, the corresponding theoretical computations were utilized. Based on the dispersion and capture of Mn$^{2+}$ by nanosheets, the corrugated structure of $C_3N_4$ and $C_3N_4$-Mn SACs was optimized through Density Functional Theory (DFT) simulations[28,29]. The results revealed that Mn was localized in the cavity of $C_3N_4$ (Supplementary Fig. 14). Additionally, $C_3N_4$ was explored by Electrostatic Potential (ESP) analysis, which resulted in the highest negative ESP in the cavity for attracting cationic Mn$^{2+}$ (Supplementary Fig. 15). This also followed the sufficiently small size of Mn$^{2+}$ (83 pm), which could be captured by the $N_6$-macroheterocycle cavity in $C_3N_4$ nanosheets[30]. As a result, the Mn atom was coordinated to four pyridine-N atoms (with $d_{Mn-N} = 2.20$, 2.23, 2.25, and 2.34 Å, respectively), instead of being located in the center of the cavity (the distance from N was about 2.50 Å). This was in accordance with the quantitative EXAFS curve fitting analysis of Mn-N, which exhibited around 3.5 of the Mn-N coordination number with the smaller $d_{Mn-N}$ (Supplementary Table 1). Consequently, with a lower atomic radius, Mn was preferably coordinated to the four nearby pyridine-N, instead of located at the center of the $C_3N_4$ cavity upon uniform interaction with six pyridine-N[31]. Furthermore, the charge transfer between $C_3N_4$ and Mn atom can be further visualized by the Charge Density Difference (CDD) analysis. The apparent electron accumulation (yellow) at Mn and electron depletion (cyan) at the surrounding N atoms were illustrated in Supplementary Fig. 16, consistent with the XPS analysis (Fig. 3c). Consequently, the strong interactions and considerable charge redistribution between Mn and $C_3N_4$ were demonstrated, being responsible for the good stability and increased catalytic activity of $C_3N_4$-Mn SACs[32]. It should be emphasized that Mn$^{2+}$ was highly dispersed, and electrostatic repulsion among cations would effectively keep them from agglomeration. Therefore, Mn was quickly captured by $C_3N_4$ and atomically dispersed in the $N_6$-macroheterocycle cavity, which contributed to the synthesis of $C_3N_4$-Mn SACs.

## Examinations on $O_2$-independent generation of ·OH by SACs

To evaluate the photocatalytic performance of $C_3N_4$-Mn SACs, the electronic structure of $C_3N_4$-Mn SACs was studied first. As evidenced by Ultraviolet-Visible (UV-vis) absorption spectra (Fig. 4a), the absorption edge of $C_3N_4$ exhibited a red shift after Mn$^{2+}$ coordinated. The charge transfer bands displayed the electron distribution between $C_3N_4$ and Mn, which indicated improved harvesting of red light by $C_3N_4$-Mn SACs. Calculated by the Tauc Plot method, the band gaps of $C_3N_4$ and $C_3N_4$-Mn were 2.88 eV and 2.44 eV, respectively. Furthermore, the Cyclic Voltammetry (CV) measurements also revealed the charge transfer in $C_3N_4$-Mn SACs. As indicated in Supplementary Fig. 17, the peak current of Mn$^{4+}$/Mn$^{3+}$ and Mn$^{3+}$/Mn$^{2+}$ dramatically increased under red-light irradiation. This was attributed to the light-irradiated charge transfer through the coordination of Mn-N in the $C_3N_4$-Mn structure. Moreover, the Valence Band Maximum (VBM) was

+2.38 eV versus Normal Hydrogen Electrode (NHE) from the VB-XPS analysis (Fig. 4b), and the derived Conduction Band Minimum (CBM) was $-0.06$ eV versus NHE estimated by band gap (Fig. 4c). Such a substantial difference between $C_3N_4$ and $C_3N_4$-Mn in CBM ($-1.11$ eV to $-0.06$ eV versus NHE) could be attributed to the lower unoccupied orbital of Mn atom. Simultaneously, the tri-s-triazine structure of $C_3N_4$ featured rich π and nonbonding (n) molecular orbitals, facilitating the LMCT process from ligand molecular orbitals to metal d-orbitals[33]. Thus, the LMCT process from $C_3N_4$ ($sp^2$-hybridized N) to Mn might be assumed in Eq. 4.

$$C_3N_4 - Mn(II) \xrightarrow{\text{light,LMCT}} C_3N_4^+ - Mn(I) \qquad (4)$$

To evaluate the band structure for supporting the improved photocatalytic oxidation by $C_3N_4$-Mn, the calculations of $C_3N_4$ and $C_3N_4$-Mn were employed with the first-principles simulations based on the density functional theory level[28]. As calculated by the Perdew-Burke-Ernzerhof (PBE) method, the Mn is completely spin-polarized and in a high spin state (Supplementary Fig. 18). To avoid the underestimating of the band gap by GGA calculations, the band gaps of $C_3N_4$ and $C_3N_4$-Mn SACs were recalculated using the Heyd-Scuseria-Ernzerhof (HSE06) hybrid functional[29,34]. As shown in Supplementary Fig. 19, the band gap of $C_3N_4$-Mn (2.54 eV) was determined to be lower than that of $C_3N_4$ (2.96 eV), which was in accordance with the band energy obtained in Fig. 4c. With the decreased band gap of $C_3N_4$-Mn, the absorption of photons would be enhanced when exposed to red light irradiation (660 nm), which was consistent with the computed light-absorption spectra (Supplementary Fig. 20). Furthermore, after the coordination of Mn$^{2+}$ with the pyridine-N atoms, the valence band (VB) level was 0.44 eV lower than $C_3N_4$, which led to a significantly lower VBM of $C_3N_4$-Mn ($-6.84$ eV versus vacuum energy level) compared to the standard oxidation potential of $H_2O$/·OH ($-6.49$ eV versus vacuum energy level) (Supplementary Fig. 19). This would further confirm the oxidation capacity of the photo-generated holes. Thus, with Mn coordinated with $C_3N_4$, the band gap was dramatically decreased and photocatalytic oxidation capacity was significantly improved, which would consequently facilitate the efficient PDT process.

The LMCT mechanism was further demonstrated by the Electron Paramagnetic Resonance (EPR) characterization of $C_3N_4$-Mn SACs under red-light irradiation. The sextet EPR peaks at $g = 2.005$ (Fig. 4d) were observed, which was ascribed to high-spin Mn(II) in $C_3N_4$-Mn SACs[35,36]. While under light irradiation, the characteristic peaks substantially lowed (Fig. 4d), demonstrating the lowering of the Mn oxidation state by the LMCT process. For a deeper comprehension of the charge transfer under light irradiation, the density of states (DOS) for both $C_3N_4$ and $C_3N_4$-Mn were determined by the DFT calculations. As a result, a number of new energy levels appeared below the CB of $C_3N_4$ once Mn$^{2+}$ bound to the N atoms of $C_3N_4$, which were attributed to the hybridization of Mn with the $C_3N_4$ (Fig. 4e, Supplementary Fig. 21). Considering the d orbital of Mn is the primary contributor to the CB of $C_3N_4$-Mn, the decrease of the band gap could be attributed to the introduction of the lower empty orbital of Mn. Simultaneously, the VB of $C_3N_4$-Mn experienced a downward shift, mostly influenced by the p orbitals of the $sp^2$-hybridized N. The occupied d orbitals of Mn exhibited a low energy level, potentially attributed to the coordination influence of $C_3N_4$. This resulted in a large energy need (>3 eV) for the excitation of Mn$^{2+}$ itself (with spin changes from 5/2 to 3/2), thereby preventing any influence on the variations in the EPR signal induced by Mn's excitation. Therefore, based on the EPR results and calculation of $C_3N_4$-Mn, this electronic excitation from VB ($sp^2$-hybridized N) to CB (Mn) corresponds to the typical LMCT mechanism. Furthermore, to provide a more detailed illustration of the LMCT process, the photo-excited charge density transition from VB to CB of $C_3N_4$-Mn was

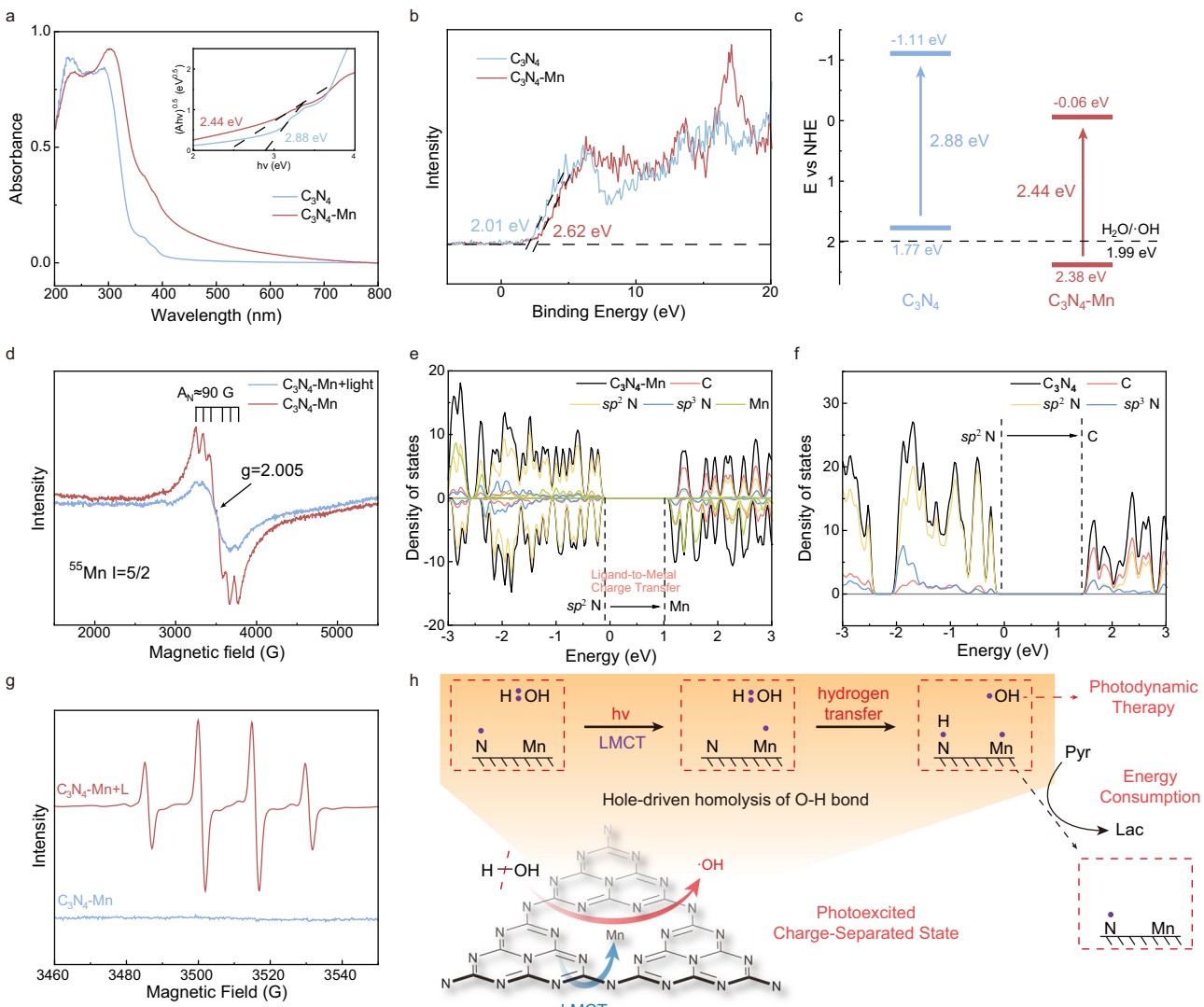

**Fig. 4 | Examinations on ·OH generation by C₃N₄-Mn SACs under light irradiation. a** UV-Vis absorption of $C_3N_4$ and $C_3N_4$-Mn SACs. **b** VB-XPS spectra of $C_3N_4$ and $C_3N_4$-Mn SACs. **c** Band position of $C_3N_4$ and $C_3N_4$-Mn SACs versus NHE. The band position versus NHE was calculated according to the following formula: $E_{NHE} = \varphi + E_{XPS} - 4.44$ (NHE level), where $\varphi$ is the work function of the instrument (4.2 eV). **d** EPR spectra of $C_3N_4$-Mn SACs before and after light irradiation. Density of states (DOS) of (**e**) $C_3N_4$-Mn and (**f**) $C_3N_4$ obtained by DFT calculations. **g** EPR spectra of $C_3N_4$-Mn SACs solution before and after the light irradiation for 30 min (660 nm, 0.4 W/cm²). DMPO acted as the trapping agent. **h** The LMCT-based photocatalytic generation of ·OH by water splitting over $C_3N_4$-Mn. A photoexcited charge reparation state was formed to produce ·OH via O-H homolysis.

examined. As presented in Supplementary Fig. 22, the excited electrons in $C_3N_4$ ($sp^2$-hybridized N) were transferred to the $d$ orbital of Mn, illustrating the characteristic photo-induced LMCT process. Notably, compared with the electron excitation process from C to N in $C_3N_4$ (Fig. 4f, Supplementary Fig. 21)[37], this LMCT process in $C_3N_4$-Mn would further enhance the separation of electrons and holes. This would facilitate the construction of the photoexcited charge-separation states, which are crucial for an efficient photocatalytic process. Therefore, the LMCT process might be achieved by inserting atomically dispersed Mn into $C_3N_4$, thereby facilitating the improvement of photocatalytic performance.

To evaluate the PDT performance of $C_3N_4$-Mn SACs, the generation of ROS under red-light irradiation was studied. Demonstrated by computation and characterization (Fig. 4c, Supplementary Figs. 19, 21), $C_3N_4$-Mn SACs possess charge-separated states and higher conduction band energy, demonstrating stronger photocatalytic oxidation capacity than $C_3N_4$. With a higher CBM potential than $\varphi(H_2O/·OH)$ (1.99 V versus NHE), $C_3N_4$-Mn SACs (CBM at 2.38 eV versus NHE) may oxidize water under light irradiation to generate ·OH. As indicated by EPR

analysis (Fig. 4g), strong signals of DMPO-·OH were obtained after the irradiation of the $C_3N_4$-Mn SACs solution (at 660 nm). This was also corroborated by enhanced FL signals of terephthalic acid (TA) upon ·OH oxidation (Supplementary Figs. 23, 24). Notably, no singlet oxygen (¹O₂) was found under light irradiation in the $C_3N_4$-Mn solution (Supplementary Fig. 25). This revealed that $C_3N_4$-Mn SACs favored electron transfer (Type-I) to generate ROS under light irradiation. In comparison, no evident ·OH was observed during light irradiation in the $C_3N_4$ solution (Supplementary Fig. 23). Therefore, the ligation of $Mn^{2+}$ could enhance ·OH generation under red-light irradiation on $C_3N_4$-Mn SACs. Excitingly, no apparent variation in the generation of ·OH by $C_3N_4$-Mn SACs was observed upon exposure to air, O₂, and N₂ atmospheres (Supplementary Fig. 26). Therefore, it can be proved that the principal oxygen source of ·OH was water, not atmospheric or dissolved O₂. Besides, no O₂ or H₂O₂ intermediates could be detected in this process (Supplementary Figs. 27, 28). This further demonstrated that ·OH was produced through the direct photocatalytic conversion of water by $C_3N_4$-Mn SACs. Therefore, ·OH was produced by an O₂-independent process upon the red-light

irradiation on $C_3N_4$-Mn SACs, which would overcome the influence of tumor hypoxia within TME.

Interestingly, ·H was also observed along with the generation of ·OH by $C_3N_4$-Mn SACs under light irradiation. This was proven by characteristic EPR signals of DMPO-H· ($A_N \approx 1.7$ mT, $A_{\beta H} \approx 1.3$ mT) upon the capture of ·H by 5,5-dimethyl-1-pyrroline N-oxide (DMPO) (Supplementary Fig. 29)[38]. Additionally, ·H was also validated by the capture with 2,2,6,6-Tetramethylpiperidine 1-oxyl (TEMPO), recording [TEMPOH + H]$^+$ (at m/z 158.1539) by a high-resolution mass spectrometer (HRMS) (Supplementary Fig. 30). The simultaneous generation of ·H suggested that ·OH was likely generated by the water-splitting process. Therefore, $C_3N_4$-Mn SACs may induce the homolysis of the H-O bond under light irradiation, promoting the efficient generation of ·OH.

Without light irradiation, Mn in $C_3N_4$ became stable after coordinating with N, which exhibited no obvious ROS signals in the $C_3N_4$-Mn SACs system (Supplementary Fig. 31). This was consistent with the lower Mn 3$d$ orbital energy after coordinating with $C_3N_4$ (Supplementary Fig. 21). Therefore, the light-induced LMCT process was regarded to be the fundamental mechanism for generating ·OH via water splitting. In this LMCT process, an exciting charge separation state was produced during red-light irradiation (Supplementary Figs. 21, 22), as demonstrated by the DOS analysis[38]. The charge separation state under light irradiation was considered to be responsible for initiating the water splitting, further corroborated by the recording of reduced ·OH signals after introducing a hole scavenger of methanol (Supplementary Fig. 32). Therefore, a mechanism of hole-driven homolysis of the O-H bond was proposed to generate ·OH under light irradiation (Eqs. 5, 6). In other words, a hole-driven hydrogen transfer from water to N in $C_3N_4$-Mn SACs triggered the generation of ·OH.

$$H_2O + C_3N_4^+ - Mn(I) \xrightarrow{\text{H transfer}} \cdot OH + C_3N_4^+(H) - Mn(I) \quad (5)$$

$$C_3N_4^+(H) - Mn(I) \longrightarrow \cdot H + C_3N_4 - Mn(II) \quad (6)$$

In addition, after irradiating $C_3N_4$-Mn for 30 min, the pH of the solution considerably dropped from 8.55 to 6.48. This suggested that in addition to producing ·H (Eq. 6), H$^+$ may also be generated by $C_3N_4^+$(H)-Mn(I) to decrease pH values (Eq. 7). As an electron donor, $C_3N_4^+$(H)-Mn(I) would react with the abundant electron receptors in cancer cells to complete the catalytic cycling of $C_3N_4$-Mn SACs. The cycle could be accelerated by pyruvic acid (Pyr), a crucial molecule for intracellular metabolism, which can acquire electrons from $C_3N_4^+$(H)-Mn(I) together with the reduction to lactic acid (Lac)[39]. This was further verified by recording the MS signal of Lac at m/z 89 ([Lac - H]$^-$) after irradiation at 660 nm (Supplementary Fig. 33), which demonstrated that $C_3N_4^+$(H)-Mn(I) can reduce Pyr to complete the catalytic cycle (Eq. 8).

$$C_3N_4^+(H) - Mn(I) \longrightarrow H^+ + C_3N_4 - Mn(I) \quad (7)$$

$$2C_3N_4^+(H) - Mn(I) + Pyr \longrightarrow Lac + 2C_3N_4 - Mn(II) \quad (8)$$

The mechanism of water splitting (Fig. 4h) was further explored using DFT calculations. The feasibility of hydrogen transfer by $C_3N_4$-Mn SACs was demonstrated by the lower hole energy (VBM level) of $C_3N_4$-Mn SACs (−6.84 eV versus vacuum energy level) than in $C_3N_4$ (−6.40 eV versus vacuum energy level) (Supplementary Fig. 19). This followed the enhanced generation of ·OH after the ligation of Mn to $C_3N_4$. Meanwhile, DFT simulations on the detailed reaction pathways (Supplementary Fig. 34) suggested that the introduction of Mn in $C_3N_4$ might significantly decrease the energy barrier of O-H homolysis in the critical step of ·OH generation. Therefore, $C_3N_4$-Mn SACs facilitated the direct and efficient water splitting for generating ·OH under red-light irradiation. Consequently, the $C_3N_4$-Mn SACs would demonstrate great potential for efficient PDT in TME due to the advantages of the $O_2$-independent generation of ·OH at hypoxia conditions.

## In situ synthesis of SACs and intracellular ROS generation

The $O_2$-independent generation of ROS by the in situ synthesized $C_3N_4$-Mn SACs in TME could facilitate a water-based photodynamic strategy for cancer therapy. As illustrated (Fig. 5a), $C_3N_4$-MnO$_2$ precursors were degraded by the upregulated endogenous GSH in TME to form $C_3N_4$-Mn SACs. This initiated a tumor-specific in situ synthesis of $C_3N_4$-Mn SACs upon the GSH depletion, avoiding the cellular scavenging of ROS to ensure efficient PDT in cancer cells. More significantly, $C_3N_4$-Mn SACs were proposed to facilitate the direct conversion of water, the most abundant species in tissues, into the most toxic ·OH under red-light irradiation. Therefore, the in situ synthesized $C_3N_4$-Mn SACs could fundamentally overcome tumor hypoxia in the oxygen-dependent PDT, initiating the specific and enhanced type-I PDT through the LMCT process.

Generally, the satisfied dark stability and light stability of the photocatalytic materials are essential premises for efficient PDT. As indicated (Supplementary Fig. 35), $C_3N_4$-Mn SACs showed excellent stability in DMEM cell culture medium and fetal bovine serum (FBS). Furthermore, $C_3N_4$-Mn also exhibited no photodegradation under red light irradiation (660 nm) in PBS. Meanwhile, to ensure the selective in situ synthesis in TME, $C_3N_4$-MnO$_2$ precursors should have strong stability and could not be degraded in the non-cancer environments without upregulated GSH. This was confirmed by the good stability of $C_3N_4$-MnO$_2$ in FBS and plasma (Supplementary Fig. 35), which would benefit the tumor-specific in situ synthesis of $C_3N_4$-Mn SACs and subsequent efficient PDT for cancer.

To further study the feasibility of intracellular synthesis of $C_3N_4$-Mn SACs, HeLa cells with upregulated GSH were selected as models of cancer cells. Meanwhile, HUV-EC cells with negative GSH acted as a control to simulate normal tissue cells. Firstly, cell imaging by Confocal Laser Scanning Microscopy (CLSM) was employed to examine the in situ synthesis of $C_3N_4$-Mn SACs in HeLa and HUV-EC cells with different treatments. As illustrated in Fig. 5b, treated with $C_3N_4$-MnO$_2$, blue signals of in situ synthesized $C_3N_4$-Mn SACs were observed on membranes and nuclei of HeLa cells. This could be attributed to the formation of sandwiched superstructures on the phospholipid bilayer (cell membrane and nuclear membrane)[20,40]. The in situ synthesis of SACs was validated by the identical cell images treated with the in situ synthesized (Fig. 5b, $C_3N_4$-MnO$_2$) and the pre-synthesized (Fig. 5b, $C_3N_4$-Mn) $C_3N_4$-Mn SACs. While after being treated with the GSH elimination reagent of N-Ethylmaleimide (NEM, 1 mM), no significant signal was observed in HeLa cells (Fig. 5b, NEM + $C_3N_4$-MnO$_2$). This revealed that $C_3N_4$-MnO$_2$ precursors could not be degraded to produce SACs without GSH. Similarly, in normal cells without high expression of GSH, the blue signal of $C_3N_4$-Mn SACs was not observed (Fig. 5c, $C_3N_4$-MnO$_2$). This revealed that the GSH-induced in situ synthesis of $C_3N_4$-Mn SACs was tumor-specific. Moreover, the GSH levels in HeLa cells were investigated by mass spectrometry (MS) before and after treated by $C_3N_4$-MnO$_2$ precursors. Compared to high GSH levels in HeLa cells, the concentration of GSH decreased relative to GSSG increased after the in situ synthesis of $C_3N_4$-Mn SACs (Fig. 5d). Afterwards, the corresponding intracellular GSH levels in HeLa cells and healthy HUV-EC cells were measured with and without the presence of $C_3N_4$-MnO$_2$. After $C_3N_4$-MnO$_2$ treatment, the GSH levels in HeLa cells exhibited a decrease of about 56%, whereas the GSH levels in the HUV-EC cells remained unchanged (Supplementary Fig. 36). This could be generated from the much lower expression of GSH in healthy cells than in the cancer cells[41]. Therefore, the synthesis of $C_3N_4$-Mn SACs was tumor-specific, triggered by the upregulated endogenous GSH in HeLa cells.

Subsequently, the in situ synthesized SACs would enable the water splitting to generate ·OH under irradiation. HeLa cells with

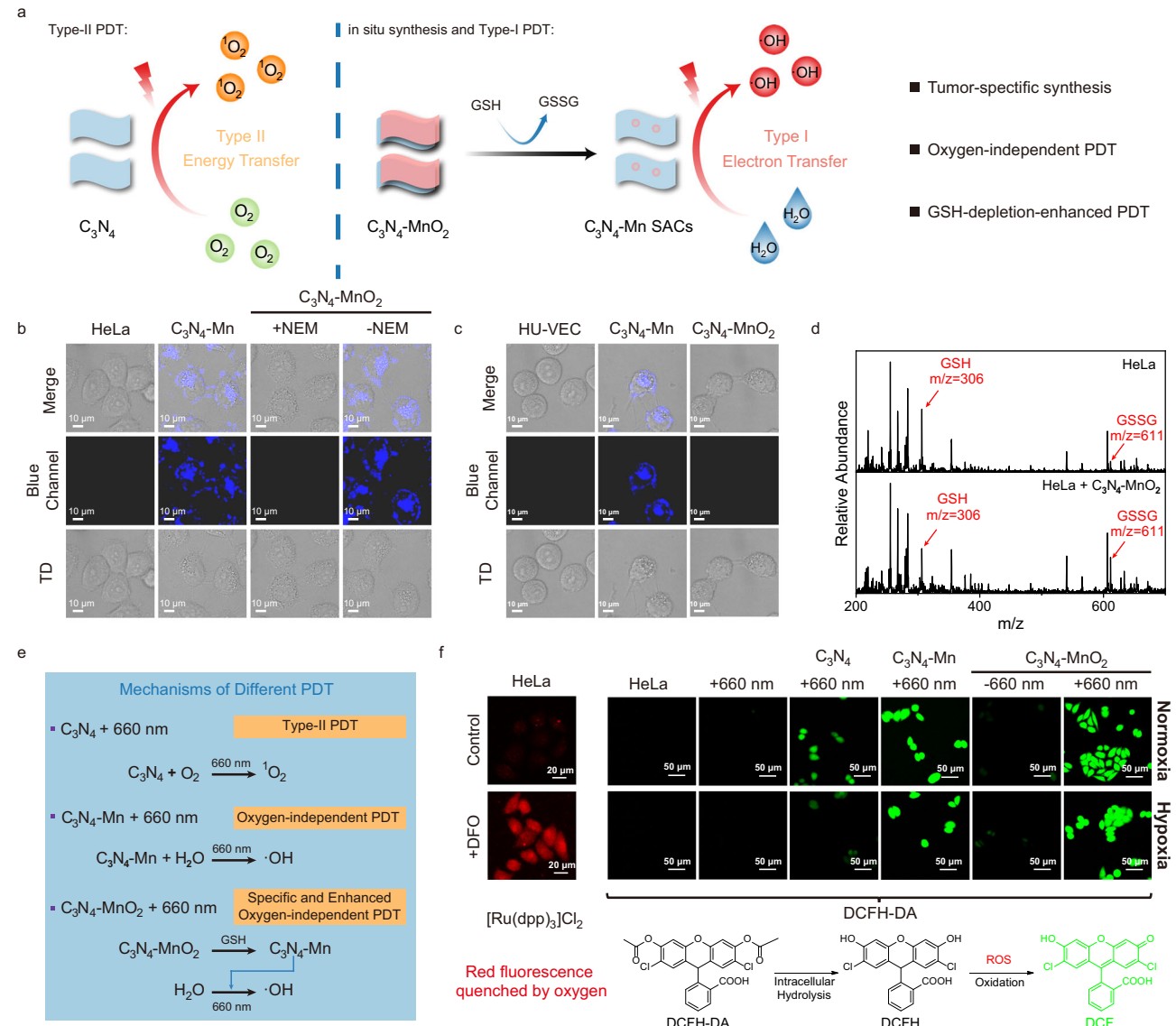

**Fig. 5 | Investigation on the GSH-induced in situ synthesis of C₃N₄-Mn SACs and the intracellular ROS generation in HeLa cells. a** Illustration on the water-based photodynamic strategy for cancer therapy. With the depletion of GSH, C₃N₄-Mn SACs were in situ generated to produce ·OH via water splitting. **b** CLSM images of HeLa cells incubated with PBS, C₃N₄-MnO₂ (in situ synthesis of SACs), pre-synthesized C₃N₄-Mn, and C₃N₄-MnO₂ + NEM (without in situ synthesis). $\lambda_{ex} = 408$ nm. **c** CLSM images of HUV-EC cells incubated with PBS, C₃N₄-MnO₂ (to evaluate the feasibility of in situ synthesis of SACs), and pre-synthesized C₃N₄-Mn. $\lambda_{ex} = 408$ nm. **d** MS spectra of HeLa cells' lysates before and after the in situ synthesis of C₃N₄-Mn SACs. **e** Mechanisms of different PDT in HeLa cells. **f** Intracellular oxygen detection of HeLa cells using Luminescent oxygen sensor ([Ru(dpp)₃]Cl₂) with and without DFO treatment, and the ROS imaging of HeLa cells after different treatments. DCFH-DA was selected as the ROS indicator. The equation is the mechanism for the ROS imaging by the indicator of DCFH-DA. [C₃N₄] = 10 μg/mL. c(NEM) = 1 mM. Light irradiation: 660 nm, 0.4 W/cm², 10 min. The experiment was repeated three times independently with similar results.

varied treatments (Fig. 5e) were imaged using CLSM to analyze the generation of ROS by the in situ synthesized C₃N₄-Mn SACs. For ROS evaluation, HeLa cells were treated with a ROS indicator of 2,7-dichlorodihydrofluorescein diacetate (DCFH-DA), generating green fluorescence from 2,7-dichlorofluorescein (DCF) upon ROS attack (Fig. 5f, equation)[42]. Under light irradiation, the maximum ROS level was found in HeLa cells treated by C₃N₄-MnO₂ precursors to support the in situ synthesis (Fig. 5f, C₃N₄-MnO₂ + 660 nm). At the same time, little ROS signal was observed without irradiation (Fig. 5f, C₃N₄-MnO₂), indicating the essential roles of irradiation for intracellular ROS generation. Furthermore, the pre-synthesized C₃N₄-Mn SACs exhibited greater ROS signals (Fig. 5f, C₃N₄-Mn + 660 nm) than C₃N₄ (Fig. 5f, C₃N₄ + 660 nm), which confirmed the improved ROS generation upon coordination of Mn to C₃N₄. Even more, the in situ synthesized C₃N₄-Mn SACs exhibited relatively higher ROS signals (Fig. 5f, C₃N₄-

MnO₂ + 660 nm) than the pre-synthesized one (Fig. 5f, C₃N₄-Mn + 660 nm). This was generated from the in situ synthesis of C₃N₄-Mn SACs upon endogenous GSH depletion, which eased the ROS consumption by the upregulated GSH in TME. Therefore, along with GSH depletion, the in situ synthesized C₃N₄-Mn SACs enhanced the intracellular generation of ROS for the subsequent effective PDT.

To examine the oxygen source of the ROS in HeLa cells, the intracellular ROS production in hypoxic environments was investigated. In the experiment, HeLa cells were treated with deferoxamine (DFO) to induce hypoxic condition[43]. As shown in Fig. 5f, the DFO-treated HeLa cells exhibited clear red signals with [Ru(dpp)₃]Cl₂ as the indicator, while no signal was recorded without DFO treatment. This suggested the successful construction of hypoxic conditions. Subsequently, the PDT performance of the aforementioned groups was further examined under hypoxic conditions. As shown in Fig. 5f,

dramatically lower ROS levels were resulted in the $C_3N_4$ group under hypoxic conditions than that obtained in normoxic environments. This was reasonably attributed to the oxygen-dependent generation of singlet oxygen by $C_3N_4$ (Supplementary Fig. 37). Notably, the strong green signals of DCF were observed in the HeLa cells treated with $C_3N_4$-Mn under both normoxia and hypoxia (Fig. 5f). This demonstrated the efficient generation of ROS by $C_3N_4$-Mn via the light-induced water-splitting, which was not impacted by the intracellular oxygen levels. Importantly, the $C_3N_4$-$MnO_2$-treated HeLa cells with irradiation also demonstrated the strongest green fluorescence under both normoxia and hypoxia, showing the successful in situ synthesis of $C_3N_4$-Mn SACs for the oxygen-independent ROS generation. Therefore, under tumor hypoxia, the in situ synthesized $C_3N_4$-Mn SACs are effective for efficient PDT upon the oxygen-independent generation of ROS.

## The lipid peroxidation-based mechanism of PDT

Considering $C_3N_4$-Mn SACs are mainly localized at the cell membranes and nucleus (Fig. 5b, c)[20,40], the PDT mechanism could be attributed to the membrane disruption by ·OH. The generated ·OH would contribute to the LPO process (on the hydrophobic side) in the phospholipid bilayer, inducing cancer cell death by membrane damages. To explore the LPO process by SACs, linoleic acid (LA) was selected as the model lipid, which was one of the abundant lipids in the body[44]. The LPO product was detected by MS after red-light irradiation. As illustrated (Fig. 6a), after the irradiation for 5 min, oxidation products and peroxidation products of LA at m/z 293, 295, 311, and 327 were observed, attributing to [LA = O - H]⁻, [LA-OH - H]⁻, [LA-OOH - H]⁻ and [LA-OOH-OH - H]⁻, respectively. These oxidate and hydroperoxide species were generated from the attack of ·OH on

unsaturated C = C bonds of LA. Moreover, certain smaller products (such as m/z 97, 171, 225, and 241) were identified in the MS spectrum after 20 min of irradiation (Fig. 6b). Therefore, more ·OH was generated with further increasing irradiation time to provide advanced oxidation products (such as ketones and aldehydes). Significantly, malonaldehyde (MDA) (m/z 71) and 4-hydroxtnonenal (4-HNE) (m/z 155) were detected, which were important LPO markers of cell death during oxidative stress[45]. The corresponding structures were developed using Collision-Induced Dissociation (CID) MS (Supplementary Fig. 38). Therefore, the present $C_3N_4$-Mn SACs can enhance PDT through the LPO process under red-light irradiation.

To further study the LPO mechanism by $C_3N_4$-Mn SACs, dynamic changes of important species during LPO process were monitored by MS under red-light irradiation. Initially (Fig. 6c), [LA - H]⁻ (m/z 279) dramatically decreased along with the increase of lipid oxidate/hydroperoxide species in the first 10 min (such as [LA-OH - H]⁻ at m/z 295 and [LA-OOH - H]⁻ at m/z 311). This indicated the LA oxidation by the attack of ·OH. Subsequently, the ions at m/z 295 and m/z 311 decreased along with the increase of smaller ions at m/z 155 and 171 (Fig. 6c). This was generated from the advanced oxidation where smaller species were produced, including the classical LPO product of 4-HNE (at m/z 155) (Fig. 6d-i). Besides, the generation of [LA = O - H]⁻ at m/z 293 was described in Fig. 6d-ii. The ions at m/z 225, 241, and 97 also displayed similar dynamic variations (Supplementary Fig. 39), suggesting the generation of another LPO product of MDA (Fig. 6d-iii). Therefore, $C_3N_4$-Mn SACs can induce LPO process, undergoing the lipid oxidation process to break the lipids.

To explore whether the LPO process was triggered by water splitting, the isotope labeling experiment was employed to determine

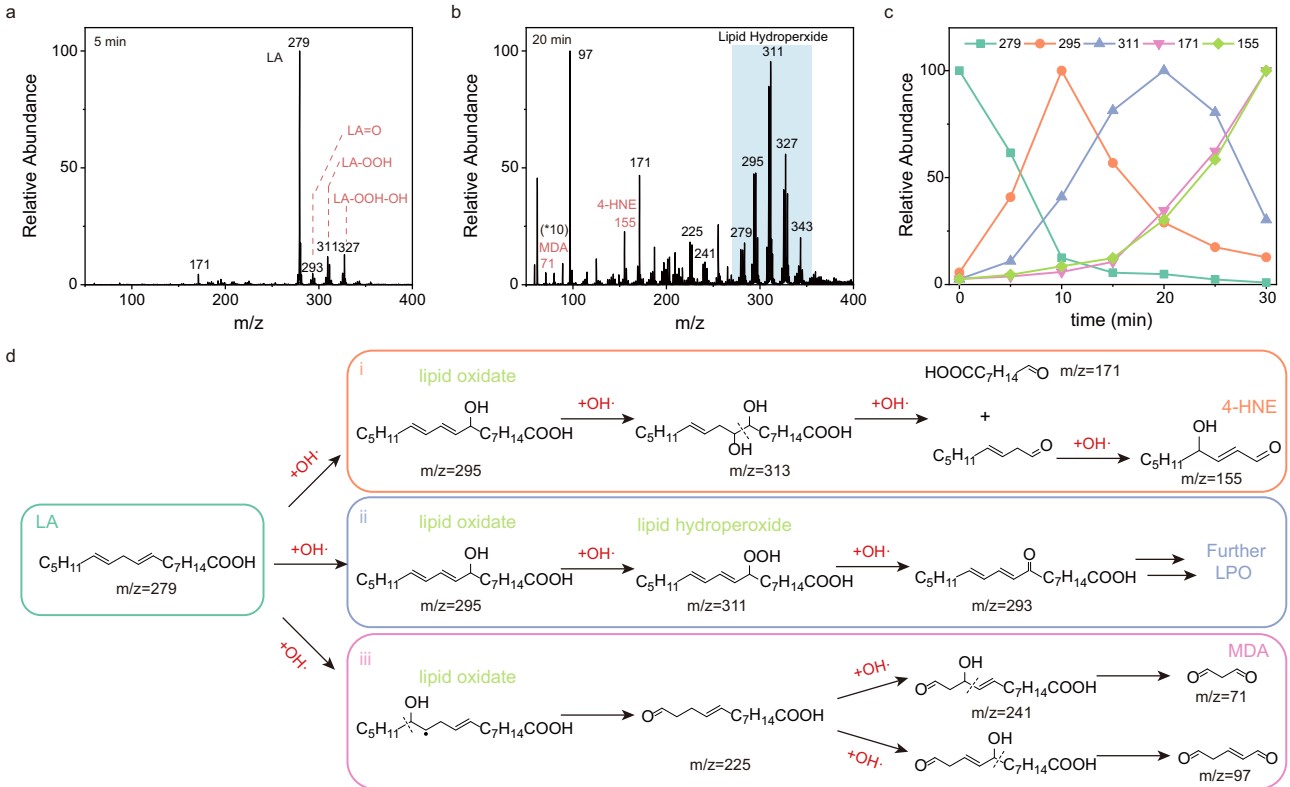

**Fig. 6 | Examination of LPO process on LA by $C_3N_4$-Mn SACs under red-light irradiation. a** MS spectrum of LA oxidized by $C_3N_4$-Mn after exposure to light (660 nm, 0.4 W/cm²) for 5 min, and products of lipid oxidate and lipid hydroperoxide were observed. **b** MS spectrum of LA oxidized by $C_3N_4$-Mn after exposure to light (660 nm, 0.4 W/cm²) for 20 min, and products of 4-HNE and MDA were recorded. **c** Time-dependent MS signal of ions at m/z 279 (LA), m/z 295 (lipid oxidate), m/z 311 (lipid hydroperoxide), m/z 171 (LPO intermediate), m/z 155 (4-HNE). **d** Mechanism illustration of the LPO process. Lipid oxide and hydroperoxide were first produced by ·OH attack, which was followed by the advanced oxidation into smaller molecules of MDA and 4-HNE. c(LA) = 100 μM, [$C_3N_4$] = 10 μg/mL. The experiment was repeated twice independently with similar results.

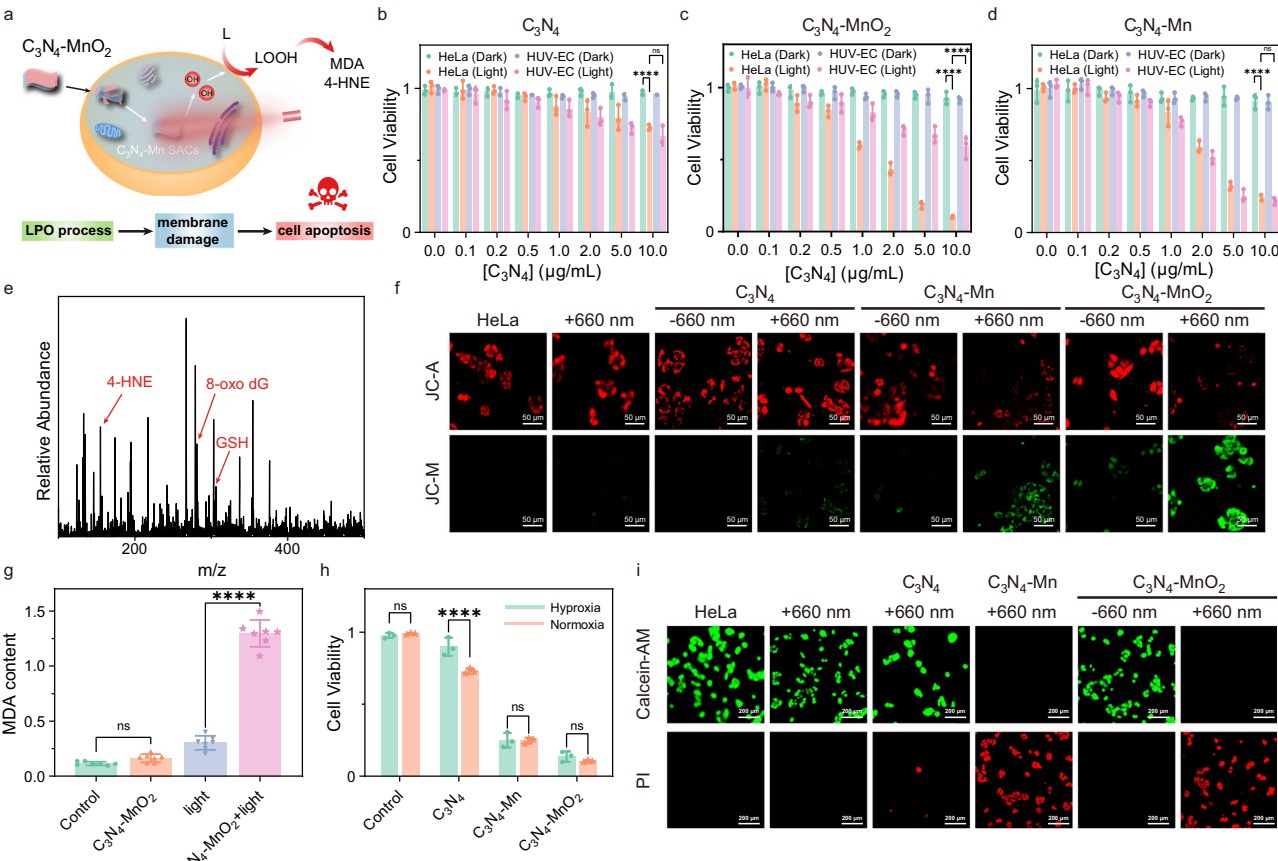

**Fig. 7 | Investigation of the intracellular LPO process by in situ synthesized C$_3$N$_4$-Mn SACs in cancer cells. a** Scheme of ·OH generation and LPO process by in situ synthesized C$_3$N$_4$-Mn SACs (obtained from intracellular C$_3$N$_4$-MnO$_2$ reduction). **b** MTT assay of HeLa and HUV-EC cells incubated with C$_3$N$_4$ ($n$ = 3 biologically independent samples). ****($p$ < 0.0001): HeLa (Dark) vs. HeLa (Light), ns ($p$ > 0.05): $p$ = 0.3040, HeLa (Light) vs. HUV-EC (Light), two-way ANOVA multiple comparison test. **c** MTT assay of HeLa and HUV-EC cells incubated with C$_3$N$_4$-MnO$_2$ precursors ($n$ = 3 biologically independent samples). ****($p$ < 0.0001): HeLa (Dark) vs. HeLa (Light), ****($p$ < 0.0001): HeLa (Light) vs. HUV-EC (Light), two-way ANOVA multiple comparison test. **d** MTT assay of HeLa and HUV-EC cells incubated with C$_3$N$_4$-Mn SACs ($n$ = 3 biologically independent samples). ****($p$ < 0.0001): HeLa (Dark) vs. HeLa (Light), ns ($p$ > 0.05): $p$ = 0.8378, HeLa (Light) vs. HUV-EC (Light), two-way ANOVA multiple comparison test. **e** MS spectrum of HeLa cells after PDT by the in situ synthesized C$_3$N$_4$-Mn SACs for 10 min. The experiment was repeated three times independently with similar results. **f** CLSM images of HeLa cells after different

treatments. The cells were stained with JC–1 ($\Delta\Psi_m$ probe, which indicated membrane damages by decreased $\Delta\Psi_m$ from red to green). **g** MDA content of HeLa cells after PDT for 10 min ($n$ = 7 biologically independent samples). ns ($p$ > 0.05): $p$ = 0.2155, Control vs. C$_3$N$_4$-MnO$_2$, ****($p$ < 0.0001): light vs. C$_3$N$_4$-MnO$_2$ + light, one-way ANOVA multiple comparison test. **h** Cell viability of HeLa cells treated with C$_3$N$_4$, C$_3$N$_4$-Mn, and C$_3$N$_4$-MnO$_2$ under light irradiation under hypoxia or normoxia environments ($n$ = 3 biologically independent samples). ns ($p$ > 0.05): $p$ = 0.7290, Control, ****($p$ < 0.0001): C$_3$N$_4$, ns ($p$ > 0.05): $p$ = 0.8953, C$_3$N$_4$-Mn, ns ($p$ > 0.05): $p$ = 0.2417, C$_3$N$_4$-MnO$_2$, two-way ANOVA multiple comparison test. **i** CLSM images of the HeLa cells with different treatments. The cells were stained with Calcein-AM (green signals indicated living cell) and propidium iodide (PI) (red signals indicated cell death). The experiment was repeated three times independently with similar results. Data are presented as mean ± SD. [C$_3$N$_4$] = 10 μg/mL. Light irradiation: 660 nm, 0.4 W/cm$^2$, 10 min.

the oxygen source in PDT. With introducing H$_2$$^{18}$O into the SACs system for the LPO process, the characteristic ions of Type-I PDT ([LA = O - H]$^-$ at m/z 293)[46] displayed 2-Da of up-shift to m/z 295 (Supplementary Fig. 40). This suggested that the LPO process was induced by ·OH generated from water splitting rather than oxygen or other oxygen sources. Therefore, upon water splitting, C$_3$N$_4$-Mn SACs initiated the LPO process for efficient PDT under red-light irradiation. The oxidate and hydroperoxide species were obtained through ·OH attack on the lipid molecules. Subsequently, these oxidation products were attacked by ·OH, triggering the breakdown of unsaturated C = C bonds and generating smaller LPO products. Through this LPO mechanism, the death of cancer cells would be induced, supporting the efficient O$_2$-independent PDT under red-light irradiation.

### Intracellular evaluation of LPO performance
The performance of the intracellular LPO process by C$_3$N$_4$-Mn SACs was further examined by cell imaging. As illustrated (Fig. 7a), the ·OH generated by the in situ synthesized C$_3$N$_4$-Mn SACs can attack lipids to

induce LPO-based membrane disruptions and initiate the subsequent cancer cell death. Significantly, C$_3$N$_4$-Mn SACs could be distributed on membranes of cancer cells, which in situ generated ·OH for attacking lipids efficiently. This would essentially avoid radical annihilation by relative long-distance transfers[47,48].

To test the light and dark toxicity of C$_3$N$_4$-Mn SACs, methylthia-zolyldiphenyltetrazolium bromide (MTT) assay was carried out for HeLa and HUV-EC cells incubated with different concentrations of C$_3$N$_4$, C$_3$N$_4$-MnO$_2$, and C$_3$N$_4$-Mn SACs under red light irradiation or in dark. Firstly, C$_3$N$_4$, C$_3$N$_4$-MnO$_2$, and C$_3$N$_4$-Mn demonstrated low dark toxicity to both HeLa cells and HUV-EC cells (Fig. 7b–d), showing the good biocompatibility of the medicine precursors. As revealed, C$_3$N$_4$ showed minimal light toxicity toward both HeLa and HUV-EC cells (Fig. 7b). Relatively, C$_3$N$_4$-Mn SACs demonstrated a potent ablation for both the HeLa cells and HUV-EC cells with an IC$_{50}$ value of about [C$_3$N$_4$] = 2.0 μg/mL (Fig. 7c). In addition, upon 660 nm of the irradiation, the in situ synthesized C$_3$N$_4$-Mn SACs (from C$_3$N$_4$-MnO$_2$) demonstrated improved cytotoxicity (IC$_{50}$ value of about

[$C_3N_4$] = 1.0 µg/mL) for HeLa cells than both $C_3N_4$ and pre-synthesized $C_3N_4$-Mn. This revealed the effective Type-I PDT after the introduction of single Mn atoms, which was further enhanced by GSH depletion. Notable, light toxicity of $C_3N_4$-$MnO_2$ for HUV-EC cells was inhibited, showing that the tumor-specific in situ synthesis of $C_3N_4$-Mn can selectively kill cancerous HeLa cells over normal HUV-EC cells. Therefore, tumor-specific in situ synthesis of $C_3N_4$-Mn SACs effectively facilitated the tumor-selective Type-I PDT.

To further reveal the mechanism of PDT by in situ synthesized $C_3N_4$-Mn SACs, the lysate of HeLa cells after PDT was studied by the MS detections. As demonstrated in Fig. 7e, GSH in HeLa cells was nearly depleted, and a series of oxidative products was identified within the HeLa cell lysates. The intracellular LPO process by in situ synthesized $C_3N_4$-Mn SACs was also validated by recording LPO products (markers) of 4-HNE (m/z 155) (Fig. 7e). Besides, 8-oxo dG (m/z 282), the marker of oxidative damage to deoxynucleotides, was also observed after light irradiation (Fig. 7e). This revealed that nucleic acid oxidation was employed during PDT in cancer cells[49]. The detection of these products of oxidative stress suggested the generation of significant ROS by $C_3N_4$-Mn SACs under light irradiation. Therefore, based on light-irradiated water splitting by in situ synthesized $C_3N_4$-Mn SACs, ·OH can be intracellularly generated for efficient Type-I PDT.

The performance of intracellular LPO process was further tested by dysfunction evaluation of mitochondria (LPO site) based on the lowered mitochondria-membrane potential ($\Delta\Psi_m$). With JC-1 as the $\Delta\Psi_m$ probe (Fig. 7f), the most significant membrane damage (the decreased $\Delta\Psi_m$ induced signal shift from red to green) was observed in the HeLa cells treated with in situ synthesized $C_3N_4$-Mn SACs under light irradiation ($C_3N_4$-$MnO_2$ + 660 nm). This indicated that the in situ synthesized $C_3N_4$-Mn SACs have remarkable performance for intracellular LPO process under light irradiation. However, HeLa cells treated by $C_3N_4$-$MnO_2$ showed significant red signals and weak green fluorescence, which demonstrated the low toxicity of the $C_3N_4$-$MnO_2$ precursors without light irradiation. The in situ synthesized $C_3N_4$-Mn SACs demonstrated dramatical decrease of the mitochondria-membrane potential, which was much obvious than the $C_3N_4$ and the pre-synthesized $C_3N_4$-Mn SACs groups (Fig. 7f). This was in accordance with the depletion of GSH and the generation of adequate ROS via water splitting (Fig. 5f). Moreover, the MDA content (markers of LPO process) within HeLa cells was also examined with an MDA test kit. As indicated in Fig. 7g, the most significant increase in MDA content was recorded in the in situ synthesized $C_3N_4$-Mn SACs group (treated with $C_3N_4$-$MnO_2$ precursors) under light irradiation. Therefore, the LPO process induced by the in situ synthesized $C_3N_4$-Mn SACs could severely affect mitochondria homeostasis and activate the subsequent death of HeLa cells.

To further investigate the PDT therapeutic efficiency by the in situ synthesized $C_3N_4$-Mn SACs in TME, the anticancer effect on HeLa cells under hypoxia was studied. Notably, both the pre-synthesized and in situ synthesized $C_3N_4$-Mn SACs demonstrated efficient cytotoxicity in both normoxia and hypoxia conditions, while hypoxia had a noticeable effect on the cell viability in the $C_3N_4$ group (Fig. 7h). This indicated that $C_3N_4$-Mn SACs with water as the oxygen source can still have strong phototoxicity in the tumor hypoxia environment. The PDT performance on the HeLa cells was further analyzed by the Calcein-AM/PI double staining (red signals represented cell death) (Fig. 7i), which suggested that the in situ synthesized $C_3N_4$-Mn SACs can initiate the efficient LPO process to induce the death of cancer cells. Therefore, with $C_3N_4$-$MnO_2$ as precursors, the in situ synthesis of $C_3N_4$-Mn SACs can be performed for efficient PDT in cancer cells.

## In vivo evaluations of the tumor treatment

The in vivo evaluation of the antitumor activity of the present in situ synthesized $C_3N_4$-Mn SACs in PDT was further conducted in the HeLa tumor-bearing mice (Fig. 8a). Tumor (HeLa) bearing Balb/c nu female mice were randomly divided into five groups ($n = 3$ biologically independent animals): saline (Saline), $C_3N_4$-$MnO_2$ without light irradiation ($C_3N_4$-$MnO_2$), $C_3N_4$, $C_3N_4$-Mn, and $C_3N_4$-$MnO_2$ under 660 nm of the irradiation ($C_3N_4$ + 660 nm, $C_3N_4$-Mn + 660 nm, $C_3N_4$-$MnO_2$ + 660 nm). The nanomedicines (100 µg/mL, 100 µL in saline) were injected into mice in each treatment group when the volume of mice tumors reached about 180 mm³. Firstly, the blood circulation and the contribution of the nanomedicine precursors of $C_3N_4$-$MnO_2$ were examined in the HeLa-bearing mice. As revealed by the pharmacokinetic analysis (Eq. 9) in Fig. 8b, the half-life of the precursors was approximately 1.4 h. The accumulation of $C_3N_4$-$MnO_2$ at the tumor site was revealed in Supplementary Fig. 41, which was attributed to the EPR effect. Furthermore, the relatively lower biodistribution at the liver (Supplementary Fig. 41) could be generated from the structural features of $C_3N_4$-$MnO_2$ (such as thinness, flexibility, and high dispersibility) for passing through the glomerular filtration barrier (DFB) and accumulating in the kidney[50]. Thereby, $C_3N_4$-$MnO_2$ could be rapidly eliminated by passing the renal filtration, decreasing the accumulation in normal tissues to avoid toxic side effects[51]. As displayed, no significant decrease of the Mn concentration was obtained in the tumor at 24 h compared to that after 12 h of the injection. To support the synthesis of $C_3N_4$-Mn SACs following the reaction of precursors with GSH after the enrichment at tumor sites, the irradiation of PDT was performed after 24 h of the injection.

$$y = 0.02227 + 0.01877*e^{(-2.777*x)} + 0.04879*e^{(-0.4951*x)} \tag{9}$$

Thereafter, the mice were subjected to the PDT treatment under light irradiation (660 nm, 0.4 W/cm², 10 min) after 24 h of the injection. After PDT treatment, tumor sizes, and body weights were measured every other day for 14 days. As illustrated in Fig. 8c, relatively stable body weight in all groups indicated the negligible side effects of these treatments on mice. For the control group of Saline, the tumor volumes increased significantly (Fig. 8d, e, Supplementary Fig. 42). However, in the presence of the nanomedicine, the tumors were effectively inhibited in light irradiation groups ($C_3N_4$ + 660 nm, $C_3N_4$-Mn + 660 nm, $C_3N_4$-$MnO_2$ + 660 nm). Specifically, $C_3N_4$-Mn exhibited more efficient tumor inhibition than $C_3N_4$, confirming that the atomically dispersed Mn in $C_3N_4$-Mn played a critical role in PDT. More notably, the in situ synthesized $C_3N_4$-Mn SACs from the reduction of $C_3N_4$-$MnO_2$ by GSH showed better therapeutic efficacy than the pre-synthesized $C_3N_4$-Mn SACs (Fig. 8d, e, Supplementary Fig. 42). This not only suggested the enhancement of PDT by GSH-depletion but also demonstrated the more efficient tumor inhibition by the adequate ·OH generated via in situ water splitting. Besides, the tumor growth inhibition value (TGI) of the $C_3N_4$-$MnO_2$ + 660 nm group was determined as 62.4%, which was greater than other groups (Supplementary Fig. 43). Two weeks later, the tumor weight of the $C_3N_4$-$MnO_2$ group under light irradiation was substantially lower than that of other groups (Fig. 8f). Therefore, the in situ synthesized $C_3N_4$-Mn SACs coupled with GSH-depletion have exhibited excellent antitumor effects, which contributed to the efficient Type-I PDT to generate adequate •OH via water splitting.

Furthermore, the PDT efficacy was examined by hematoxylin and eosin (H&E) staining assay, Ki67 assay, and TdT-mediated dUTP nick-end labeling (TUNEL) assay. After 24 h of the treatments, the mice were sacrificed to collect their tumors for histological investigation. As indicated by H&E staining of main organs (Heart, Liver, Spleen, Lung, Kidney), no evident tissue damage was observed to indicate the negligible side effects of these therapies on mice (Supplementary Fig. 44). While the dead or apoptotic cells with condensed nuclei were proven in the $C_3N_4$-$MnO_2$ + 660 nm group, whereas the massive viable cancer cells were exhibited in the control group (Fig. 8g). This suggested that the in situ synthesized $C_3N_4$-Mn SACs demonstrated significant

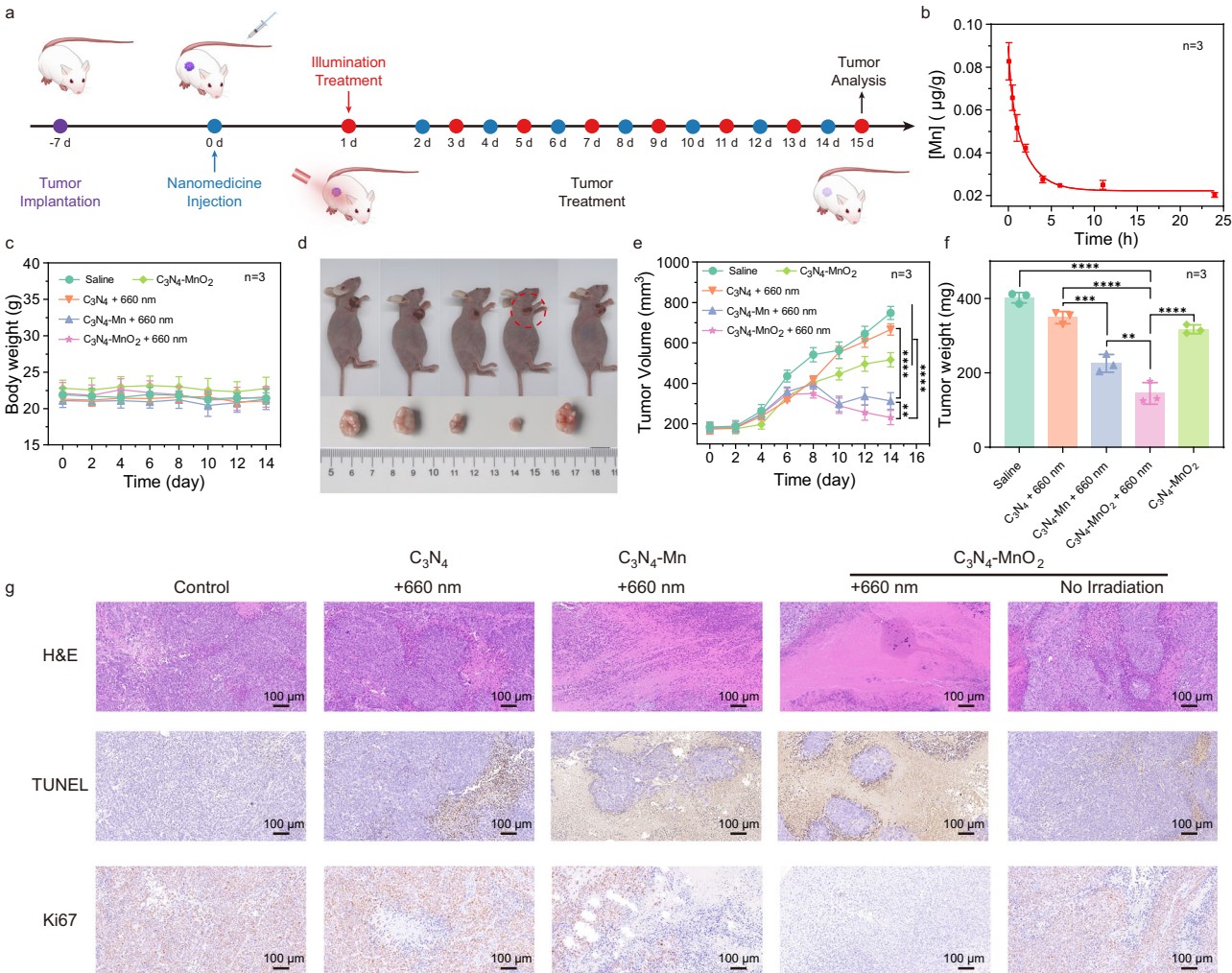

**Fig. 8 | In vivo PDT by in situ synthesized C$_3$N$_4$-Mn SACs. a** Schematic illustration of the establishment of the HeLa-bearing mice model and PDT treatments. **b** Blood circulation of C$_3$N$_4$-MnO$_2$ precursors by measuring Mn concentrations over 24 h after the intravenous injection. **c** The body weights of HeLa tumor-bearing mice after various treatments in 14 days. **d** Representative photos of the mice with different treatments (from left to right: Saline, C$_3$N$_4$ + 660 nm, C$_3$N$_4$-Mn + 660 nm, C$_3$N$_4$-MnO$_2$ + 660 nm, C$_3$N$_4$-MnO$_2$) and corresponding tumor tissues collected from different groups on the 14$^{th}$ day. **e** The tumor volume of HeLa tumor-bearing mice after various treatments in 14 days. ****($p < 0.0001$): Saline vs. C$_3$N$_4$-MnO$_2$ + 660 nm, ****($p < 0.0001$): C$_3$N$_4$ + 660 nm vs. C$_3$N$_4$-MnO$_2$ + 660 nm, ****($p < 0.0001$): C$_3$N$_4$ + 660 nm vs. C$_3$N$_4$-Mn + 660 nm, **($p < 0.033$): $p = 0.0040$, C$_3$N$_4$-Mn + 660 nm vs. C$_3$N$_4$-MnO$_2$ + 660 nm, ****($p < 0.0001$): C$_3$N$_4$-MnO$_2$ + 660 nm vs. C$_3$N$_4$-MnO$_2$, one-way ANOVA multiple comparison test. **f** Weight of dissected tumor from each group. ****($p < 0.0001$): Saline vs. C$_3$N$_4$-MnO$_2$ + 660 nm, ****($p < 0.0001$): C$_3$N$_4$ + 660 nm vs. C$_3$N$_4$-MnO$_2$ + 660 nm, ***($p < 0.0021$): $p = 0.0002$, C$_3$N$_4$ + 660 nm vs. C$_3$N$_4$-Mn + 660 nm, **($p < 0.033$): $p = 0.0042$, C$_3$N$_4$-Mn + 660 nm vs. C$_3$N$_4$-MnO$_2$ + 660 nm, ****($p < 0.0001$): C$_3$N$_4$-MnO$_2$ + 660 nm vs. C$_3$N$_4$-MnO$_2$, one-way ANOVA multiple comparison test. **g** Images of H&E-stained, TUNEL-stained, and Ki67-stained tumor slices. The slices were collected from different groups on the 14$^{th}$ day. The experiment was repeated three times independently with similar results. Data are presented as mean ± SD ($n = 3$ biologically independent animals). [C$_3$N$_4$] = 0.5 mg/kg. Light irradiation: 660 nm, 0.4 W/cm$^2$, 10 min.

damage on the tumor, which was also in accordance with the results of TUNEL staining with the lowest blue signals to indicate efficient cancer cell death. The tumor proliferation and malignancy were further investigated by Ki67 staining of tumor slices (Fig. 8g), which resulted in the lowest level of Ki67 (with the lowest red signals to signify the lowest proliferation) after being treated by the in situ synthesized C$_3$N$_4$-Mn SACs. Therefore, upon the in situ reduction of C$_3$N$_4$-MnO$_2$ by endogenous GSH, the in situ synthesized C$_3$N$_4$-Mn SACs exhibited an excellent antitumoral impact for cancer treatment through an O$_2$-independent Type-I PDT therapy.

In summary, a GSH-activated synthesis of SACs was developed for the cancer-cell-specific PDT through water splitting. The C$_3$N$_4$-Mn SACs were in situ synthesized by reducing C$_3$N$_4$-MnO$_2$ with upregulated GSH in cancer cells. The photocatalytic performance of C$_3$N$_4$ obtained a substantial boost by inserting atomically dispersed Mn into the C$_3$N$_4$ structure. Thus, under red-light irradiation, the in situ

synthesized SACs could efficiently induce the O$_2$-independent water splitting for efficient and selective ·OH generation. This facilitated the efficient Type-I PDT via the LPO process. Moreover, this in situ synthesis of SACs strategy can permit PDT with maximal therapeutic efficacy on tumors and minimal side effects over healthy tissues. Meanwhile, the water-based photodynamic strategy minimizes the limitations of TME hypoxia via O$_2$-independent water splitting for generating adequate ·OH. Therefore, the most active ROS of ·OH can be selectively and efficiently generated for efficient Type-I PDT. The present in situ synthesized C$_3$N$_4$-Mn SACs showed remarkable in vivo cancer therapy owing to the GSH depletion and efficient Type-I PDT upon water splitting. This in situ synthesis of SACs would inspire efficient and precise cancer therapy via directly converting inert H$_2$O into highly reactive ·OH in cancer cells. In the future, combined with multiple modifications for obtaining tumor-specific targeting or near-infrared light absorption with stronger penetrating capacity,

this in situ synthesis of SACs would inspire wider biological applications.

## Methods

### Cell lines and animals
The human cervical cancer cells (HeLa cells) and the human umbilical vein endothelial cells (HUV-EC cells) were purchased from the National Experimental Cell Resource sharing Platform (NICR). Female Balb/c nu mice (5 weeks) were purchased from Shanghai SLAC Laboratory Animal Co.,Ltd. All animal experimental protocols were reviewed and approved by the Ethics Committee of Beijing Normal University and complied with all relevant ethical regulations (permit no. BNUCC-EAW-2023-16). In accordance with the requirements, the size of the sub-cutaneous tumor must meet the specifications, with a maximum of 2000 mm$^3$. Euthanasia has to be performed once the size is achieved. In any of animal experiment described in this article, the maximal tumor size of the mouse was never reached.

### Synthesis of bulk C$_3$N$_4$ and g-C$_3$N$_4$ nanosheets
The yellow bulk C$_3$N$_4$ was first prepared by calcination of melamine. Briefly, white melamine (5 g) was annealed at 550 °C for 4 h. Then, 300 mg of bulk C$_3$N$_4$ was dispersed in 100 mL of HNO$_3$ (5 M) and the mixture was subjected to ultrasound for 1 h. Subsequently, the solution was refluxed for 12 h at 125 °C. The white product was centrifuged (7008 × $g$, 15 min) and washed with water to near-neutral pH. Finally, the resultant suspension was exfoliated in 30 mL water by performing a sonication for 10 h and then centrifuged at 985.5 × $g$ for 30 min to remove the unexfoliated and large-area nanosheets. The concentration of g-C$_3$N$_4$ solution was ~1 mg/mL.

### Preparation of the 2D/2D C$_3$N$_4$-MnO$_2$ nanocomposite
The 2D/2D nanocomposite of C$_3$N$_4$-MnO$_2$ was prepared by the reduction of KMnO$_4$ on g-C$_3$N$_4$. 2.5 mL g-C$_3$N$_4$ solution (100 μg/mL), 1 mL 10 mM KMnO$_4$, and 2.5 mL 0.1 M MES buffer (pH = 6.0) were added and dispersed into 10 mL water. Afterward, the solution was sonicated until it turned into a brown colloid. After centrifuging (7008 × $g$, 15 min) and purifying by water, the C$_3$N$_4$-MnO$_2$ precipitates were dispersed into 2.5 mL water for further use. The concentration of C$_3$N$_4$-MnO$_2$ suspension was ~500 μg/mL.

### Formation of C$_3$N$_4$-Mn SACs
C$_3$N$_4$-Mn SACs were prepared by mixing 10 mL C$_3$N$_4$-MnO$_2$ (500 μg/mL) with GSH (10 mM). After treatment for 30 min, the brown colloid began to fade and gradually turned into a white powder. The solution was centrifuged (7008 × $g$, 15 min) and washed with deionized water twice to remove the unbonded GSH. Finally, the product was dispersed into 10 mL at ~100 μg/mL for further experiments.

### GSH depletion of C$_3$N$_4$-MnO$_2$
For the GSH depletion, C$_3$N$_4$-MnO$_2$ (50 μg/mL) was mixed with different concentrations of GSH at room temperature. After incubating for 30 min, the fluorescence spectra were measured under the excitation at 312 nm and the mass spectra were collected to evaluate GSH oxidation. Online monitoring of GSH concentration was carried out using a homemade ionization source. Briefly, C$_3$N$_4$-MnO$_2$ (50 μg/mL) was mixed with GSH (1 mM), and the solutions were introduced into the mass spectrometer in real time for the detection of GSH.

### Electrochemical redox reactions
A standard three-electrode cell consisting of a Platinum filament working electrode was adopted for CV measurements. A calomel electrode acted as a reference electrode, while a platinum electrode was used as an auxiliary electrode. CV scanning was conducted between −1.00 and 1.50 V (scan rate: 100 mV/s). C$_3$N$_4$-Mn (10 mg/mL) or MnNO$_3$ (100 mM) was added for these CV measurements.

### Detection of extracellular ·OH
The extracellular radicals were determined by EPR analysis using 5,5-dimethyl-1-pyrroline N-oxide (DMPO) as the spin-trapping agents of ·OH. The ·OH generation under different light irradiation (0.2, 0.4, 0.7 W/cm$^2$) was determined by the fluorescence of terephthalic acid (TA). Typically, TA (10 μM) was mixed with C$_3$N$_4$-Mn (50 μg/mL) and then exposed to irradiation at 660 nm. After irradiation, the fluorescence spectra of TA were measured. To explore the catalytic activity of Mn$^{2+}$ after coordinating to C$_3$N$_4$, ·OH generated from H$_2$O$_2$ was determined by the 4,4'-diamino-3,3',5,5'-tetramethyl biphenyl (TMB) probe. Typically, TMB (10 μM, 10 mM in DMSO) was respectively mixed with Mn$^{2+}$ (1 μM) or C$_3$N$_4$-Mn (10 μg/mL), after which the H$_2$O$_2$ (100 μM) was added for collecting the UV-vis absorption spectra of TMB.

### Detection of extracellular ¹O₂
$^1$O$_2$ generated under irradiation was determined by the 1,3-diphenyli-sobenzofuran (DPBF) probe. Typically, DPBF (10 μM, 10 mM in MeOH/H$_2$O v-v = 1:1) was respectively mixed with C$_3$N$_4$ or C$_3$N$_4$-Mn (10 μg/mL), the UV-vis absorption spectra of DPBF was collected after irradiation (660 nm, 0.4 W/cm$^2$, 30 min).The extracellular $^1$O$_2$ generation by C$_3$N$_4$-Mn SACs (10 μg/mL) was determined by the EPR analysis. 2,2,6,6-tet-ramethylpiperide (TEMP) was used as a spin-trapping agent of $^1$O$_2$. The EPR signals were recorded after irradiating the solution (660 nm, 0.4 W/cm$^2$, 30 min).

### LPO process evaluations
The LPO process was monitored with a mass spectrometer. Briefly, 100 μM linoleic acid (LA) and 10 μg/mL C$_3$N$_4$-Mn were mixed in the solution of MeOH/H$_2$O (v:v = 1:1). The solution was irradiated by 660 nm laser (0.4 W/cm$^2$, 30 min), and the mass spectra were recorded every 5 min after irradiation. The MS spectra were collected in the negative ion mode and were acquired using Thermo Xcalibur software.

### Cell culture
Human cervical adenocarcinoma epithelial cells (HeLa) and Human Umbilical Vein Endothelial Cells (HUV-EC) were cultured with regular growth media consisting of high glucose DMEM. The cell growth media were supplemented with 10% FBS, 100 μ/mL penicillin, and 100 mg/mL streptomycin and cultured at 37 °C in a 5% CO$_2$ humidified environment. The media was changed every two days and the cells were digested by trypsin, which was then re-suspended in a fresh medium before plating.

### Intracellular imaging of C$_3$N$_4$-MnO$_2$
To monitor the intracellular uptake of the C$_3$N$_4$-MnO$_2$ precursors and in situ synthesis of C$_3$N$_4$-Mn SACs, the HeLa cells and the HUV-EC cells were incubated with C$_3$N$_4$-MnO$_2$ (50 μg/mL) or other treatment groups for 8 h. Briefly, 1 mL HeLa cells or HUV-EC cells (~5*10$^5$ cells) suspension were pre-incubated for 12 h in the glass bottom cell culture disk. Then, the cells were incubated with C$_3$N$_4$-MnO$_2$ or other nanomedicines for 12 h, and the fluorescence image of cells was recorded under 408 nm excitation.

### Simulation of the intracellular hypoxia condition
To investigate the influences of the hypoxia condition in tumors, deferoxamine (DFO, 100 μM) was added to induce a hypoxic environment. Intracellular hypoxia was evaluated by the [Ru(dpp)$_3$]Cl$_2$ (Luminescent oxygen sensor). It was non-fluorescent when cells were in a normal oxygen environment and became fluorescent when the oxygen levels were decreased.

### Cell viability test
To investigate the dark and the light toxicity of C$_3$N$_4$-MnO$_2$, HeLa cells and HUV-EC cells planted in 96-well plates were treated with different groups. Briefly, 100 μL HeLa cells or HUV-EC cells (~5000 cells)

suspension were pre-incubated for 12 h. Then, the cells were incubated with $C_3N_4$-$MnO_2$ or other nanomedicines for 12 h. Subsequently, light-related groups were treated with irradiation at 660 nm (0.4 W/cm$^2$, 10 min) and other groups were kept in the dark. Then, the solution in the wells was removed and 100 μL of 3-(4,5-dimethyl-2-thiazolyl)-2,5-diphenyl-2-H-tetrazolium bromide (MTT, 0.5 mg/mL in DMEM) solution was added to each plate. Then the solution in each well was carefully removed and replaced by 100 μL DMSO to dissolve the formazan. Cell cytotoxicity was measured by the absorbance at 490 nm. In contrast, to investigate the influences of the hypoxia condition in tumors, deferoxamine (DFO, 100 μM) was added to induce a hypoxic environment, and the cell viability was tested according to the aforementioned procedures.

### Cell imaging in vitro
For intracellular ROS detection, HeLa cells were stained by 2'7'-Dichlorodihydrofluorescein diacetate (DCFH-DA, 10 μM) under different treatments, and the intracellular ROS levels were indicated by fluorescence signal of DCF. For mitochondria membrane potential ($\Delta\Psi_m$) detection, the treated cells were stained with JC-1. The red signal of JC-1 aggregates or the green signal of JC-1 monomers was individually observed to indicate normal or decreased $\Delta\Psi_m$, respectively. To visualize the cell killing of PDT, cells were double stained with calcein-AM and PI to identify live/dead cells after 30 min of the treatments. Briefly, 1 mL HeLa cells (~5*10$^5$ cells) suspension were pre-incubated for 12 h in the glass bottom cell culture disk for cell imaging in vitro.

### Intracellular GSH and LPO levels
To determine intracellular GSH and LPO levels of HeLa cells after different treatments, cells (~10$^6$ cells) were seeded in 6-well plates and treated with/without $C_3N_4$-$MnO_2$ in light or dark. Subsequently, cells (~2.5*10$^6$ cells) were collected and the levels of lysed, intracellular GSH, and LPO levels were determined by mass spectrometer. Intracellular Malondialdehyde (MDA) level for HeLa cells was tested by using the MDA Assay Kit.

### In vivo antitumor performance
The tumor (HeLa) bearing Balb/c nu female mouse (6 weeks) model was built by subcutaneous injection of HeLa cells ($2 \times 10^7$ mL$^{-1}$, 100 μL) into the axilla of each mouse. Tumor (HeLa) bearing mice (tumor volume: ~180 mm$^3$) were randomly divided into five groups (3 mice in each group): saline group (control), $C_3N_4$-$MnO_2$ without laser irradiation group ($C_3N_4$-$MnO_2$), and groups of $C_3N_4$, $C_3N_4$-Mn, $C_3N_4$-$MnO_2$ with 660 nm irradiation ($C_3N_4$ + 660 nm, $C_3N_4$-Mn + 660 nm, $C_3N_4$-$MnO_2$ + 660 nm). Then the nanomedicine ([$C_3N_4$] = 100 μg/mL, 100 μL in saline) was injected into mice in each administration group by intravenous injections. After the injection of nanomedicine for 24 h, the tumor section was irradiated at 660 nm (0.4 W/cm$^2$) for 10 min. The tumor size was measured every two days and calculated by: V = 1/2 (tumor length) × (tumor width)$^2$. After two weeks, tumors and main organs were collected from the killed mice for further analysis.

### In vivo blood circulation and biodistribution
HeLa cancer-bearing mice were intravenously injected with $C_3N_4$-$MnO_2$ (500 μg/mL, 100 μL in saline). At indicated time points (0.1, 0.5, 1, 2, 4, 6, 11, 24 h), 50 μL blood was collected from the tail of each mouse. After intravenous injections for 12, 24, and 48 h, the mice were killed to measure the Mn amount in the liver, spleen, kidneys heart, lung, and tumor by ICP-OES.

### Histological analysis
All mice of different groups were sacrificed on the 14th day, and major organs and tumors were separated and made into slices for H&E, TUNEL, and Ki67 staining. Major organs were collected and fixed in 4% paraformaldehyde, which was then embedded into paraffin to obtain the slices at the thickness of 5 μm. The tissue slices were stained with H&E, TUNEL, and Ki67 and then imaged by optical microscopy and assessed by 3 independent pathologists.

### Statistical analysis
The results were presented as mean values of replicate experiments or replicate samples in one representative experiment, as indicated in the figure legends. Statistical analysis was performed using GraphPad Prism and $p < 0.05$ was considered statistically significant.

### Reporting summary
Further information on research design is available in the Nature Portfolio Reporting Summary linked to this article.

## Data availability
The authors declare that all data generated in this study are available within the article or the Supplementary Information. Other data related to this work are available from the corresponding authors upon request.

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

## Acknowledgements

This study was supported by the National Natural Science Foundation of China (NNSFC, 22274012 to N.N., 21974010 to J.O.), the Fundamental Research Funds for the Central Universities (No.2233300007 to N.N.), Key Project of Science and Technology Plan of Beijing Education Commission (No.KZ20231002807 to N.N.), and the 1W1B station at Beijing Synchrotron Radiation Facility (BSRF).

## Author contributions

Y.Yin and X.Ge conceived the idea and designed the experiments; J.Ouyang supported the characterizations; N.Na supervised the project and acquired the funding; Y.Yin and X.Ge conducted the experiments with the assistance from N.Na; Y.Yin conducted the Calculation; Y.Yin, X.Ge, and N.Na analyzed the data and interpreted the results. All authors read, discussed, and commented on the manuscript.

## Competing interests

The authors declare no competing interests.
