## [Peer Review File · Nature Communications]

Reviewers' Comments:

Reviewer #1:

Remarks to the Author:

This manuscript reports the in-situ synthesis of Mn single atom decorated g-C₃N₄ from the precursor 2D/2D g-C₃N₄/MnO₂ under the trigger of overexpressed GSH in the tumor microenvironment. The application of this g-C₃N₄-Mn SAC is for water splitting induced hydroxyl radical generation for killing cancer cell. Notably, this strategy is O₂-independent, overcoming the hypoxia TME challenge against efficient PDT. The MnO₂ reacts with GSH and depletes it for enhanced ROS effect inducing cancer apoptosis via lipid peroxidation. The strategy is well supported by both experimental and theoretical analysis. Although the in-situ SAC synthesis for water-based PDT is impressive, it still has major drawbacks like the lack of tumor targeting ability and light-penetration depth issue. It will be convincing to provide few discussions based on this downside in the Introduction part, so the readers get a clear perspective of the strategy. Also, the authors can provide their ideas about how to improve the design further considering these factors, in the Conclusion Section. More comments/suggestions are point-out as below:

1. As said earlier, the design lacks the tumor targeting ability. As mentioned in the introduction, the SAC in situ synthesis to overcome the challenge of random distribution of nanomedicine in healthy tissue, I would like the authors to comment on the ability of the designed 2D/2D g-C₃N₄/MnO₂ whether to target tumor site is possible? It is understood that overexpressed GSH will convert C₃N₄/MnO₂ to C₃N₄-Mn SAC, but first how does this nanomedicine design have ability to reach the tumor?
2. What is the blood circulation life time of the C₃N₄/MnO₂? How about the biocompatibility without any surface modifications?
3. Does the author have evidence of how MnO₂ is carried by C₃N₄? Is it just physical absorption or is there another chemical interaction?
4. What % of C₃N₄/MnO₂ is accumulated in tumor? Is it higher or lower compared to other organs? It is better to perform the ICP-MS in a time dependent manner to understand the C₃N₄/MnO₂ accumulation fashion.
5. The author claims that ligand-to-metal charge transfer occurs between C₃N₄-Mn, and the calculation determined that the energy gap is 660 nm. However, the absorbance spectrum in Fig. 2A shows a broad band (400-700 nm) enhancement. Please give more description for the absorbance spectrum.
6. The author should provide more explanation and detailed labels in Supplementary Fig. 18. Simply describing "(electron transfer from the conjugated pi orbital of sp²-hybridized N to Mn 3d_{z²} orbital), and (46 is the Mn atom and the red area represents sp² N)" are not enough to understand.
7. Why is the Mn peak missing in C₃N₄-Mn in XPS spectrum in Fig 13 (c) in supporting information? There is a marking as Mn indicated by arrow but the peak is not found. This is in contrast to the statement mentioned in line 112.
8. In XRD, the (100) plane corresponding to 2 theta 13.3 degree is not obvious amidst the noise. Please try to cancel noise in the XRD and redo. Also mark the corresponding (hkl) planes.
9. Line 105, there is no MnO₂ XRD, and how to infer that there is no characteristic peak of MnO₂ in C₃N₄-Mn. XRD cannot prove the point that MnO₂ is reduced. Change the statement in the manuscript.
10. In Figure 1(D), the HAADF image is not clear. A bright spot is even marked above the 2 nm scale label? The layering issue arises doubts. Clarity is missing. Provide the image with higher dpi for better clarity. Although other characterization supports the Mn-N formation and exclude Mn-Mn presence, HAADF-STEM image is more important to show the single atom formation.
11. Following the previous question, in Figure 10 (D), in Supporting information, the EDS mapping shows almost abundant Mn similar to C, N and O compositions. How can this support the argument that Mn is SAC with 2.34 wt%. Justify this.
12. What is the purpose of Supplementary Fig. 2? The author should explain the decrease in fluorescence and provide the absorbance spectrum of MnO₂ alone.
13. In animal experiment, the authors have chosen 24 h after nanomedicine to irradiate with light. Based on what data was this time point chosen? Is the accumulation higher at tumor after 24h? biodistribution?
14. According to Supplementary Fig. 20, the •OH signal is silent in the C₃N₄+L group. However, in Fig. 5E, F, G, C₃N₄+L shows toxicity in Hela cells. Maybe the source of toxicity is from singlet

- oxygen. It is better to include an additional group (C3N4+L) in Supplementary Fig. 22.
15. For the cell experiments, the mechanism for in situ synthesized C3N4-Mn SACs was GSH reacting with MnO₂ to form a single atom Mn on C3N4. Cells with negative GSH should be considered.
 16. I think Supplementary Fig. 12 was mislabeled, and the XRD pattern for the C3N4-MnO₂ group was missed.
 17. Supplementary Fig. 16 is incorrect; it shows a CV spectrum, not an XPS spectrum, and the labels A-D are missing. Please provide the correct figure.
 18. The labels in figure must be presented more clearly and I suggest to increase the font size inside the figures.
 19. P-3, line 62, "Cell experiment indicated that C3N4-Mn SACs were...". Check the statement clearly.
 20. Line 66, spelling error – adequate; line 203, charge separation.
 21. At least three repeats of the results should be provided in Supplementary Fig. 31. The label in Supplementary Fig. 29 (page 25) is wrong; it should be Supplementary Fig. 32.

Reviewer #2:

Remarks to the Author:

The authors have reported their work on Mn embedded g-C₃N₄ with photodynamic therapy a precise cancer treatment, which looks good for the bio-part. As I am not an expert in biochemistry or cancer, I would make some comments on the materials' characterizing. As shown in Figure 1f and 1k, it is hard to rule out MnO@g-C₃N₄, which is close to the experimentally reported TiO@g-C₃N₄ (DOI: 10.1039/C5SC03551A). Also, it is well known g-C₃N₄ is not a planar structure, which is reported both in the experiments (DOI: 10.1021/jp510187c, DOI: 10.1039/C3RA45776A, DOI: 10.1039/c3ta13188b) and in the calculations (DOI: 10.1007/s11467-018-0754-6, DOI: 10.1021/acs.jpcc.0c00411). It is thus believed that data from DFT calculations in this work should be questionable. One can see that either experimental or theoretical data could not support the authors' statement. I cannot recommend the current manuscript for publication.

I would like to suggest the authors to do a more rigorous calculation considering both Mn@g-C₃N₄ and MnO@g-C₃N₄ with a corrugated model for a more precise analysis on the materials. And then revised the manuscript for resubmitting.

Reviewer #3:

Remarks to the Author:

The manuscript "Water-based Photodynamic Therapy: Cancer-Cell- Activated In Situ Synthesis of Single-Atom Catalysts for O₂- Independent Precise Cancer Treatment" deals with the preparation and use of Mn-C₃N₄ single atom photocatalysts and their use in cancer therapy through the generation of .OH radicals following red light irradiation. The main claim of the manuscript is that the catalyst is synthesized in situ thanks to the presence of GSH in the tumor microenvironment (TME), that would reduce the MnO₂ overlayer, facilitating the anchoring of Mn²⁺ cations in C₃N₄. Then the Mn-C₃N₄ catalyst would be photoactivated to produce water splitting, generating .OH radicals in situ.

I am not recommending publication of this work, on account of i) limited novelty and ii) claims that are not sufficiently substantiated, regarding both the selective generation of the catalyst within cancer cells and the water splitting process described.

- Regarding the novelty:

1. Even though the authors seem to give the impression that their C₃N₄-Mn SAC material is novel, the fact is that it is not. Materials with very similar (if not the same) structure have been repeatedly reported (e.g. Mo et al. *App. Catal. B, Envir*, 2019; Wang et al. *ACS Sust. Chem. Eng.* 2018; Guo et al. *JACS*, 2019, to name a few). These manuscripts demonstrated SAC structure, often resorting to EXAFS/XANES, as in the manuscript. Even the preparation of the pre-catalyst (MnO₂ on top of the C₃N₄ layers) has been reported, using the same precursors, exfoliated C₃N₄ and KMnO₄ (Mo et al *Appl Catal B* 2019). Amazingly, they are not quoted, even though the material and the reactions reported are relevant to this work.

2. Nor is the generation of ROS with these materials new. Mn-C₃N₄ has been reported to generate

superoxide, singlet oxygen and hydroxyl radicals, depending on the conditions used.

- Regarding claims not sufficiently substantiated:

1. The main claim of the manuscript, the selective generation of the catalyst inside cancer cells is far from demonstrated. According to the authors the MnO₂/C₃N₄ construct would be reduced by GSH and since this is overexpressed in cancer cells the reduction that gives rise to the synthesis of the catalyst would be selective, and the catalyst (with the corresponding generation of .OH) would only take place within cancer cells. This is a central claim of the manuscript. However, the selectivity premise is wrong, since the MnO₂ is likely to be reduced by many other molecules present in biological environments. And in any case, although GSH is somewhat overexpressed in cancer cells, there are significant concentrations also in healthy cells (see for instance Bonet et al. Chemical Sci. 2022). Therefore, to be able to claim that the catalyst is only activated in the TME a demonstration with a large number of cancer and non cancer cells is needed, plus other representative environments (plasma, DMEM).

3. Related to this, it seems that the C₃N₄-Mn particles stayed mainly in the cancer cell membranes (something that the authors see as the reason of the lipidic oxidation observed). However, in this case they would not be exposed to the "high" GSH concentrations inside the cell and the catalyst would not be formed.

2. Another important claim is the water splitting under the conditions used. There is some doubts on this that are not dissipated by the results presented. For instance, if the hole-driven homolysis takes place as depicted in fig 2., what happens to the .H? is it desorbed as hydrogen (if so, they would have noticed, I guess), does it stay in the liquid as protons and lowers the pH of the environment? It should be noted that there are quite a few other reactions that can also give rise to .OH, among them the peroxone reaction reported by Guo et al with the same catalyst in the presence of H₂O₂, also present in the TME. Also, .OH radicals are readily formed by MnC₃N₄ from H₂O₂, present in the TME, as shown by Wang et al. ACS Sust Chem Eng 2018.

Other than that, there are other problems with this manuscript, starting with English language with so many flaws that sometimes the meaning is lost.

Reply to Reviewer 1's Comments

Comments to the Author

Although the in-situ SAC synthesis for water-based PDT is impressive, it still has major drawbacks like the lack of tumor targeting ability and light-penetration depth issue. It will be convincing to provide few discussions based on this downside in the Introduction part, so the readers get a clear perspective of the strategy. Also, the authors can provide their ideas about how to improve the design further considering these factors, in the Conclusion Section.

Response:

Thank you very much for the comments from the Reviewer. Just as the Reviewer indicated, the present strategy mainly focused on the *in situ* synthesis of the SACs and the oxygen-independent generation of $\cdot\text{OH}$. Therefore, the $\text{C}_3\text{N}_4\text{-MnO}_2$ precursors were simply enriched at the tumor site through the EPR effect without corresponding modifications for increasing tumor targeting ability. Although red light (660 nm) used in our strategy exhibited stronger penetration ability than the commonly used C_3N_4 (~450 nm), the light-penetration depth could be further improved in the future studies. As suggested by the Reviewer, to clear the perspective of the present strategy, the corresponding sentences have been added in both Introduction and Conclusions as follows:

(Page 3, Line 54-65, Highlighted Revised Manuscript)

Herein, a single-atom catalyst ($\text{C}_3\text{N}_4\text{-Mn SACs}$) was *in situ* prepared in TME for the highly specific Type-I PDT upon the oxygen-independent generation of $\cdot\text{OH}$ with minimal invasiveness. Firstly, a nanomedicine precursor of 2D/2D $\text{C}_3\text{N}_4\text{-MnO}_2$ was prepared, which was lowly toxic to normal tissues (such as the kidney and liver) in the metabolism (Scheme 1). To implement the *in situ* synthesis and oxygen-independent generation of $\cdot\text{OH}$, $\text{C}_3\text{N}_4\text{-MnO}_2$ was enriched at the tumor site through the enhanced permeability and retention (EPR) effect. Subsequently, the *in situ* release of C_3N_4 and Mn^{2+} from $\text{C}_3\text{N}_4\text{-MnO}_2$ was employed, which responded to the upregulated GSH in TME. Consequently, atomically dispersed Mn^{2+} was captured by C_3N_4 as a hexadentate ligand, facilitating the *in situ* synthesis of $\text{C}_3\text{N}_4\text{-Mn SACs}$ within TME. Especially based on Ligand-to-Metal charge transfer (LMCT) from C_3N_4 to Mn^{2+} , $\text{C}_3\text{N}_4\text{-Mn SACs}$ exhibited red-light harvesting for efficient O_2 -independent generation of $\cdot\text{OH}$ via water-splitting. The red shift of the absorption also facilitated the red light-irradiated applications (660 nm) with greater penetration depth than that of white light in PDT.¹⁸ ...

(Page 27, Line 565-567, Highlighted Revised Manuscript)

... In the future, combined with multiple modifications for obtaining tumor-specific targeting or near-infrared light absorption with stronger penetrating capacity, this *in situ* synthesis of SACs would inspire wider biological applications.

More comments/suggestions:

1. As said earlier, the design lacks the tumor targeting ability. As mentioned in the introduction, the SAC *in situ* synthesis to overcome the challenge of random distribution of nanomedicine in healthy tissue, I would like the authors to comment on the ability of the designed 2D/2D g-C₃N₄/MnO₂ whether to target tumor site is possible? It is understood that overexpressed GSH will convert C₃N₄/MnO₂ to C₃N₄-Mn SAC, but first how does this nanomedicine design have ability to reach the tumor?

Response:

Thank you very much for the comments from the Reviewer. Just as the Reviewer indicated, to focus on *in situ* synthesis and oxygen-independent ·OH generation, C₃N₄-MnO₂ was simply enriched at the tumor site through EPR effect without tumor-targeted modifications. According to the comment, the corresponding statements have been added in Introduction. In addition, the additional experiments on evaluating the accumulation of C₃N₄-MnO₂ at tumor sites was carried out. In the experiments, the biodistributions of C₃N₄-MnO₂ at different tissues of the mice (heart, liver, spleen, lung, kidney and tumor) were determined by measuring Mn concentrations over 12, 24, 48 h after the intravenous injection. As resulted (Supplementary Fig. 38), C₃N₄-MnO₂ can be accumulated at tumor within 12 h. Although C₃N₄-MnO₂ can also be distributed in other tissues, it could still be low-toxic due to the short of GSH for converting C₃N₄-MnO₂ into toxic C₃N₄-Mn SAC. Therefore, with the suitable size of approximately 80 nm (Fig. 1C, Supplementary Fig. 4), the C₃N₄-MnO₂ can be accumulated at tumor sites through EPR effect for the subsequent cancer therapy.

According to the comment, the corresponding revisions have been added in the revised manuscript as follows:

(Page 3, Line 57-60, Highlighted Revised Manuscript):

... To implement the *in situ* synthesis and oxygen-independent generation of ·OH, C₃N₄-MnO₂ was enriched at the tumor site through the enhanced permeability and retention (EPR)

effect. Subsequently, the *in situ* release of C_3N_4 and Mn^{2+} from $C_3N_4-MnO_2$ was employed, which responded to the upregulated GSH in TME. ...

(Page 24, Line 501-505, Highlighted Revised Manuscript)

... The accumulation of $C_3N_4-MnO_2$ at the tumor site was revealed in Supplementary Fig. 38, which was attributed to the EPR effect. As displayed, no significant decrease of the Mn concentration was obtained in the tumor at 24 h compared to that after 12 h of the injection. To support the synthesis of C_3N_4-Mn SACs following the reaction of precursors with GSH after the enrichment at tumor sites, the irradiation of PDT was performed after 24 h of the injection.

(Page S29, Supplementary Figure 38, Revised Supplementary Information)

Supplementary Figure 38. The biodistribution of $C_3N_4-MnO_2$ by measuring Mn concentrations over 12, 24, 48 h after the intravenous injection.

2. What is the blood circulation life time of the C_3N_4/MnO_2 ? How about the biocompatibility without any surface modifications?

Response:

According to the comment, the additional experiments on evaluating the blood circulation and the biocompatibility have been employed by the pharmacokinetic analysis and MTT assay. As

demonstrated by the pharmacokinetic analysis (Equation 9) in the HeLa-bearing mice, the half-life of the C₃N₄/MnO₂ precursors was approximately 1.4 h (Fig. 7B). In addition, the good biocompatibility of C₃N₄/MnO₂ was also demonstrated by cytotoxicity experiments (Fig. 6B, 6C, 6D) and H&E staining of normal tissue (Supplementary Fig. 41).

Consequently, the corresponding revisions have been added in the revised manuscript as follows:

(Page 24, Line 498-501, Highlighted Revised Manuscript)

... Firstly, the blood circulation and the contribution of the nanomedicine precursors of C₃N₄-MnO₂ were examined in the HeLa-bearing mice. As revealed by the pharmacokinetic analysis (Equation 9) in Fig. 7B, the half-life of the precursors was approximately 1.4 h. ...

(Page 24, Equation 9, Highlighted Revised Manuscript)

$$y = 0.02227 + 0.01877 * e^{(-2.777*x)} + 0.04879 * e^{(-0.4951*x)} \quad (9)$$

(Page 26, Figure 7B, Highlighted Revised Manuscript)

Fig. 7. In vivo PDT by *in situ* synthesized C_3N_4 -Mn SACs. (A) Schematic illustration of the establishment of the mice model and PDT treatments. **(B)** Blood circulation of C_3N_4 - MnO_2 precursors by measuring Mn concentrations over 24 h after the intravenous injection. **(C)** The body weights of HeLa tumor-bearing mice after various treatments in 14 days. ...

(Page 22, Line 439-444, Highlighted Revised Manuscript)

To test the light and dark toxicity of C_3N_4 -Mn SACs, methylthiazolyldiphenyltetrazolium bromide (MTT) assay was carried out for HeLa and HUV-EC cells incubated with different concentrations of C_3N_4 , C_3N_4 - MnO_2 , and C_3N_4 -Mn SACs under red light irradiation or in dark. Firstly, C_3N_4 , C_3N_4 - MnO_2 , and C_3N_4 -Mn demonstrated low dark toxicity to both HeLa cells and HUV-EC cells (Fig. 6B, 6C, 6D), showing the good biocompatibility of the medicine

precursors. As revealed, C₃N₄ showed minimal light toxicity toward both HeLa and HUV-EC cells (Fig. 6B). ...

(Page 21, Figure 6B, 6C, 6D, Highlighted Revised Manuscript)

Fig. 6. Investigation on the intracellular LPO by *in situ* synthesized C₃N₄-Mn SACs in cancer cells. (A) Scheme of ·OH generation and LPO by *in situ* synthesized C₃N₄-Mn SACs (obtained from intracellular C₃N₄-MnO₂ reduction). (B-D) MTT assay of HeLa and HUV-EC cells incubated with (B) C₃N₄, (C) C₃N₄-MnO₂ precursors, (D) C₃N₄-Mn SACs. (E) MS spectrum of HeLa cells after PDT by the *in situ* synthesized C₃N₄-Mn SACs for 10 min. The experiment was repeated three times independently with similar results.

3. Does the author have evidence of how MnO₂ is carried by C₃N₄? Is it just physical absorption or is there another chemical interaction?

Response:

We are sorry for the unclear description on the carrying of MnO₂ by C₃N₄. As reported, KMnO₄ can be reduced to MnO₂, resulting the deposition of MnO₂ on C₃N₄ (*Anal. Chem.* 2014, 86, 7, 3426). The chemical interaction between MnO₂ and C₃N₄ could be demonstrated based on the XPS

characterizations. As shown in Supplementary Fig. 13, the peaks of Mn(III) (Mn 2p_{1/2} at 653.1 eV and Mn 2p_{3/2} at 641.5 eV)¹² and Mn(IV) (Mn 2p_{1/2} at 655.7 eV and Mn 2p_{3/2} at 643.8 eV)^{13, 14} were recorded in C₃N₄-MnO₂ (Supplementary Fig. 13D). Besides, with MnO₂ deposited on C₃N₄, the characteristic peaks of N-C₃ (400.5 eV), C-N=C (398.7 eV) as well as C-N=C (288.2 eV) in C₃N₄ (Supplementary Fig. 13F) shifted to higher binding energies. Consequently, the electron transfer from C₃N₄ to MnO₂ was demonstrated. This is also in accordance with the reported works (*Angew. Chem. Int. Ed.* 2022, 61, e202116699; *ACS Sustainable Chem. Eng.* 2018, 6, 1, 965).

Consequently, the corresponding demonstration has been added in Supporting Information as follows:

(Page 7, Line 135-139, Highlighted Revised Manuscript)

... As illustrated in Supplementary Fig. 13, the characteristic peak of Mn was observed in C₃N₄-MnO₂ and C₃N₄-Mn but was absent in C₃N₄. In addition, the electron transfer from C₃N₄ to MnO₂ was demonstrated based on the XPS characterizations, and particular peaks of Mn(II) (Mn 2p_{1/2} at 653.8 eV, Mn 2p_{3/2} at 641.5 eV) were observed in C₃N₄-Mn SACs.²⁷ ...

(Page S9, Revised Supporting Information)

As shown in Supplementary Fig. 13, the peaks of Mn(III) (Mn 2p_{1/2} at 653.1 eV and Mn 2p_{3/2} at 641.5 eV)¹² and Mn(IV) (Mn 2p_{1/2} at 655.7 eV and Mn 2p_{3/2} at 643.8 eV)^{13, 14} were recorded in C₃N₄-MnO₂ (Supplementary Fig. 13D). Besides, with MnO₂ deposited on C₃N₄, the characteristic peaks of N-C₃ (400.5 eV), C-N=C (398.7 eV) as well as C-N=C (288.2 eV) in C₃N₄ (Supplementary Fig. 13F) shifted to higher binding energies. This could be due to the electron transfer from C₃N₄ to MnO₂.¹⁵

4. What % of C₃N₄/MnO₂ is accumulated in tumor? Is it higher or lower compared to other organs?
It is better to perform the ICP-MS in a time dependent manner to understand the C₃N₄/MnO₂ accumulation fashion.

Response:

As suggested by the Reviewer, the additional experiments have been added for examining the C₃N₄/MnO₂ accumulation fashion. In the experiments, the biodistribution of C₃N₄-MnO₂ in heart, liver, spleen, lungs, kidney, and tumor tissues at 12, 24, 48 h were evaluated by ICP-MS analysis. As shown in Supplementary Figure 38, the C₃N₄-MnO₂ can be accumulated at tumor within 12 h. Although C₃N₄-

MnO₂ can also be distributed in other tissues, it could still be low-toxic due to the short of GSH for converting C₃N₄-MnO₂ into toxic C₃N₄-Mn SAC. Therefore, the C₃N₄-MnO₂ can be accumulated at tumor sites through EPR effect, which facilitated the subsequent *in situ* conversion to the toxic C₃N₄-Mn SAC for tumor-specific therapy.

Consequently, the corresponding section has been revised as follows:

(Page 24, Line 501-505, Highlighted Revised Manuscript)

... The accumulation of C₃N₄-MnO₂ at the tumor site was revealed in Supplementary Fig. 38, which was attributed to the EPR effect. As displayed, no significant decrease of the Mn concentration was obtained in the tumor at 24 h compared to that after 12 h of the injection. To support the synthesis of C₃N₄-Mn SACs following the reaction of precursors with GSH after the enrichment at tumor sites, the irradiation of PDT was performed after 24 h of the injection.

(Page S29, Supplementary Figure 38, Revised Supplementary Information)

Supplementary Figure 38. The biodistribution of C₃N₄-MnO₂ by measuring Mn concentrations over 12, 24, and 48 h after the intravenous injection.

5. The author claims that ligand-to-metal charge transfer occurs between C_3N_4 -Mn, and the calculation determined that the energy gap is 660 nm. However, the absorbance spectrum in Fig. 2A shows a broad band (400-700 nm) enhancement. Please give more description for the absorbance spectrum.

Response:

We are sorry for the unclear descriptions and explanations. To make a better understanding of a broad band (400-700 nm) enhancement, the additional examinations have been carried out. In the revised works, the UV spectra in original Fig. 2A were studied by using the Tauc Plot method and the band gaps of C_3N_4 and C_3N_4 -Mn were calculated. As resulted, the band gap of C_3N_4 -Mn was determined to be 2.44 eV, which was much higher than the calculated one at 660 nm (1.870 eV in Supplementary Figure S19). To demonstrate the ligand-to-metal charge transfer between C_3N_4 and Mn, the VB-XPS testing for C_3N_4 -Mn and C_3N_4 was further conducted upon combining with UV-Vis spectra. Indicated by the resulted VBM and CBM energy, the band gap of C_3N_4 -Mn SACs was not as low as the theoretical value. The difference could be generated from that not all the C_3N_4 holes were coordinated with Mn, being somehow deviated from the theoretical calculations.

Consequently, according to the comment, the corresponding figure has been added in Figure 3 and the more description on the absorbance spectra has been added in the revised manuscript as follows:

(Page 10, Line 184-200, Highlighted Revised Manuscript)

To evaluate the photocatalytic performance of C_3N_4 -Mn SACs, the electronic structure of C_3N_4 -Mn SACs was studied first. As evidenced by Ultraviolet-Visible (UV-vis) absorption spectra (Fig. 3A), the absorption edge of C_3N_4 exhibited a red shift after Mn^{2+} coordinated. The charge transfer bands displayed the electron distribution between C_3N_4 and Mn, which indicated improved harvesting of red light by C_3N_4 -Mn SACs. **Calculated by the Tauc Plot method, the band gaps of C_3N_4 and C_3N_4 -Mn were 2.88 eV and 2.44 eV, respectively. Further Cyclic Voltammetry (CV) measurements also revealed the charge transfer in C_3N_4 -Mn SACs.** As indicated in Supplementary Fig. 16, the peak current of Mn^{4+}/Mn^{3+} and Mn^{3+}/Mn^{2+} dramatically increased under red-light irradiation. This was attributed to the light-irradiated charge transfer through the coordination of Mn-N in the C_3N_4 -Mn structure. Moreover, the Valence Band Maximum (VBM) was +2.38 eV versus Normal Hydrogen Electrode (NHE) from the VB-XPS analysis (Fig. 3B), and the derived Conduction Band Minimum (CBM) was -0.06 eV estimated by band gap (Fig. 3C). Such a substantial difference between C_3N_4 and C_3N_4 -Mn in CBM (-1.11 eV to -0.06 eV) could be attributed to the lower unoccupied orbital of

Mn atom. The different band gap values from the calculated one could be generated because not all the holes were coordinated with Mn. Simultaneously, the tri-s-triazine structure of C_3N_4 featured rich π and nonbonding (n) molecular orbitals, facilitating the LMCT process from ligand molecular orbitals to metal d-orbitals.³³ Thus, the LMCT process from C_3N_4 (sp^2 -hybridized N) to Mn might be assumed in Equation 4.

(Page 12, Figure 3A, 3B, 3C, Highlighted Revised Manuscript)

Fig. 3. Examinations on $\cdot OH$ generation by C_3N_4 -Mn SACs under light irradiation. (A) UV-Vis absorption of C_3N_4 and C_3N_4 -Mn SACs. The experiment was repeated three times independently with similar results. **(B)** VB-XPS spectra of C_3N_4 and C_3N_4 -Mn SACs. **(C)** Band position of C_3N_4 and C_3N_4 -Mn SACs versus NHE. **(D)** EPR spectra of C_3N_4 -Mn SACs before and after light irradiation. ...

6. The author should provide more explanation and detailed labels in Supplementary Fig. 18. Simply describing "(electron transfer from the conjugated pi orbital of sp²-hybridized N to Mn 3d_z² orbital), and (46 is the Mn atom and the red area represents sp² N)" are not enough to understand.

Response:

We are sorry for the unclear descriptions. To improve the explanation of Supplementary Fig. 18 (in the original Supporting Information), the calculation on the electron transfer was re-calculated and the corresponding labels and descriptions have been revised. In addition, the calculation was revised by arranging C, N and Mn in the more reasonable planar diagram for the better demonstration (also indicated by the Reviewer #2). The results were shown in Supplementary Fig. 20 (the revised version of original Supplementary Fig. 18), whose range of different electron transfers were divided by dotted lines for the better understanding.

The corresponding revisions are shown as follows:

(Page S17, Supplementary Figure 20, Revised Supplementary Information)

Supplementary Figure 20. The 2D colour-filled maps of transition density matrix in the S1 – S10 excited state transitions. (A) C_3N_4 . (B) C_3N_4 -Mn. Labels in the figure: C 1-18, N 19-45, Mn 46 (in the C_3N_4 -Mn), the rest is H.

(Page S18, Revised Supplementary Information)

For the excited state of C_3N_4 (Supplementary Fig. 20), the values of electron excitation in the $N \rightarrow N$, and $N \rightarrow C$ region are the strongest. This revealed that the electronic excited state of C_3N_4 is mainly the locally excited state (LE state). After the introduction of Mn, the strong electron excitation is totally within the $N \rightarrow Mn$ region, which shows the charge transfer

state (CT state) of the C_3N_4 -Mn. Besides, photoexcited holes (n orbital of N) in C_3N_4 -Mn are perpendicular to the C_3N_4 plane, displaying the easy symmetry matching with molecular orbitals of small molecules (like water) (Supplementary Fig. 19). Consequently, a smaller steric hindrance was obtained in C_3N_4 -Mn SACs, which facilitated the enhancement of water splitting to generate $\cdot OH$.

7. Why is the Mn peak missing in C_3N_4 -Mn in XPS spectrum in Fig 13 (c) in supporting information? There is a marking as Mn indicated by arrow but the peak is not found. This is in contrast to the statement mentioned in line 112.

Response:

We are sorry for the poor exhibition. According to the comment, the C_3N_4 -Mn in XPS spectrum has been re-examined in the revised works. As shown in Supplementary Figure 13C, a small peak of Mn was recorded. The low intensity of Mn was attributed to the coordination of Mn atoms into C_3N_4 , resulting low amount of Mn on the surface of the C_3N_4 -Mn.

Consequently, the corresponding figure has been revised as follows:

(Page S9, Supplementary Figure 13, Revised Supplementary Information)

Supplementary Figure 13. XPS spectra of g- C_3N_4 (A), C_3N_4 - MnO_2 (B) and C_3N_4 -Mn (C). Mn 2p spectra of C_3N_4 - MnO_2 (D), C_3N_4 -Mn SACs (E) and C 1s spectra of C_3N_4 , C_3N_4 - MnO_2 and C_3N_4 -Mn (F) at high resolution.

8. In XRD, the (100) plane corresponding to 2 theta 13.3 degree is not obvious amidst the noise. Please try to cancel noise in the XRD and redo. Also mark the corresponding (hkl) planes.

Response:

We are sorry for the poor XRD spectra. According to the comment, the XRD characterizations have been re-collected in the revised works. As shown in Figure 2B, the (100) plane corresponding to 2 theta 13.3 degree can be observed in the revised figure. In addition, the corresponding (hkl) planes have also been marked.

The revised figure is shown as follows:

(Page 9, Figure 2B, Highlighted Revised Manuscript)

Fig. 2. Fine structure of Mn atom in the C₃N₄-Mn SACs. **(A)** FTIR spectra of g-C₃N₄ nanosheets, C₃N₄-MnO₂ precursor, and C₃N₄-Mn SACs. **(B)** XRD patterns of bulk C₃N₄, g-C₃N₄, C₃N₄-MnO₂, MnO₂, and C₃N₄-Mn SACs. **(C)** N 1s XPS spectra of C₃N₄-Mn SACs and g-C₃N₄. ...

9. Line 105, there is no MnO₂ XRD, and how to infer that there is no characteristic peak of MnO₂ in C₃N₄-Mn. XRD cannot prove the point that MnO₂ is reduced. Change the statement in the manuscript.

Response:

Just as the Reviewer indicated, the characteristic peaks of MnO₂ in C₃N₄-Mn can not be well evaluated without the controlled XRD data of MnO₂. Consequently, additional experiments have been carried out for collecting XRD data of MnO₂, facilitating the demonstration of MnO₂ degradation during the *in situ* synthesis. In addition, the XRD spectra of other materials, including bulk C₃N₄, g-C₃N₄, C₃N₄-MnO₂, MnO₂ and C₃N₄-Mn SACs, have also been collected for the better demonstration.

As suggested by the Reviewer, the corresponding statement and figure have been revised as follows:

(Page 7, Line 125-130, Highlighted Revised Manuscript)

... This conclusion was also shown by the X-Ray Diffraction (XRD) patterns. As illustrated in Fig. 2B, the disappearance of the (100) peak at 13.4° in g-C₃N₄, C₃N₄-MnO₂, and C₃N₄-Mn showed the conversion of bulk phase to 2D structure of C₃N₄.²⁴ Furthermore, the peaks of MnO₂ (12.3°, 36.7°, 65.8°, JCPDS 01-074-7889)²⁵ were not observed and no noticeable alteration of C₃N₄ (002) peak (27.6°) in C₃N₄-Mn SACs was recorded. This indicated that MnO₂ was consumed by GSH and the 2D structure of C₃N₄ was retained in C₃N₄-Mn SACs.²² ...

10. In Figure 1(D), the HAADF image is not clear. A bright spot is even marked above the 2 nm scale label? The layering issue arises doubts. Clarity is missing. Provide the image with higher dpi for better clarity. Although other characterization supports the Mn-N formation and exclude Mn-Mn presence, HAADF-STEM image is more important to show the single atom formation.

Response:

We are sorry for the unclear exhibition. As suggested, the HAADF-STEM image has been re-collected at the thin edge, obtaining the single-layer image with higher dpi for the better clarity. The revised figure is shown in Figure 1F (the revised version of Figure 1D in original manuscript):

(Page 6, Figure 1F, Highlighted Revised Manuscript)

Fig. 1. The synthesis of C_3N_4 -Mn SACs by the reduction of C_3N_4 - MnO_2 by GSH and the morphology characterizations. (A) The mechanism illustration. MnO_2 was reduced into Mn^{2+} by GSH, and Mn^{2+} was captured by C_3N_4 nanosheets via the coordination at the N_6 -cavity. ... (F) HAADF-STEM image of C_3N_4 -Mn SACs with the atomic dispersed Mn highlighted by red circles.

11. Following the previous question, in Figure 10 (D), in Supporting information, the EDS mapping shows almost abundant Mn similar to C, N and O compositions. How can this support the argument that Mn is SAC with 2.34 wt%. Justify this.

Response:

We appreciate that the Reviewer's pointing out the mistakes. Just as the Reviewer indicated, the old version of the EDS mapping is not satisfied due to the over-scanning during the data collection.

Consequently, the EDS mapping has been re-collected in the revised works. The revised EDS diagram is shown in Figure 1E (the revised version of the original Supplementary Figure 10D):

(Page 6, Figure 1E, Highlighted Revised Manuscript)

Fig. 1. The synthesis of $\text{C}_3\text{N}_4\text{-Mn}$ SACs by the reduction of $\text{C}_3\text{N}_4\text{-MnO}_2$ by GSH and the morphology characterizations. (A) The mechanism illustration. MnO_2 was reduced into Mn^{2+} by GSH, and Mn^{2+} was captured by C_3N_4 nanosheets via the coordination at the N_6 -cavity. ... (E) EDS mapping of $\text{C}_3\text{N}_4\text{-Mn}$ SACs. (F) HAADF-STEM image of $\text{C}_3\text{N}_4\text{-Mn}$ SACs with the atomic dispersed Mn highlighted by red circles.

12. What is the purpose of Supplementary Fig. 2? The author should explain the decrease in fluorescence and provide the absorbance spectrum of MnO₂ alone.

Response:

We are sorry for the misleading demonstrations. The SEM and TEM images in Supplementary Fig. 2 was used to confirm the successful preparation of g-C₃N₄. The images also supported the deposition and the degradation of the MnO₂ on C₃N₄ upon comparing with the images of C₃N₄-MnO₂ and C₃N₄-Mn SACs. The corresponding sentences have been revised as follows:

(Page 4, Line 79-83, Highlighted Revised Manuscript):

... The successful preparation of g-C₃N₄ nanosheets (Supplementary Fig. 2) and the deposition of MnO₂ on C₃N₄ (Supplementary Fig. 3 and 4) were confirmed by Transmission Electron Microscopy (TEM) and spectroscopic characterizations. This was also in accordance with the observing of the lattice fringe of MnO₂ on C₃N₄, with a lattice stripe spacing of ~0.31 nm (Fig. 1B). ...

Besides, as suggested by the Reviewer, changes of C₃N₄ fluorescence and UV-vis absorption of MnO₂ alone have been collected in the additional experiments. As shown in Supplementary Figure 3, the fluorescence of C₃N₄ decreased with MnO₂ deposited on C₃N₄ upon the reduction of KMnO₄. It can be observed that MnO₂ have a broad absorption band, which overlaps well with the fluorescence emission of g-C₃N₄. Therefore, this FL quenching was generated upon accepting excited electrons by MnO₂, simultaneously confirming the successful deposition of MnO₂ on g-C₃N₄ nanosheets.

Consequently, the corresponding section has been revised as follows:

(Page S4, Revised Supplementary Information)

... The fluorescence (FL) intensity of C₃N₄ significantly decreased with the increase of KMnO₄ concentration till 1 mM (Supplementary Fig. 3). It can be observed that MnO₂ have a broad absorption band, which overlaps well with the fluorescence emission of the g-C₃N₄. Therefore, this FL quenching was induced by the absorption of excited electrons by MnO₂, simultaneously confirming the successful deposition of MnO₂ on g-C₃N₄ nanosheets.¹⁰ ...

(Page S3, Supplementary Figure 3B, Revised Supplementary Information)

Supplementary Figure 3. (A) Fluorescence spectra of C_3N_4 nanosheet after MnO_2 deposited upon the reduction of $KMnO_4$ at different concentrations. The excitation wavelength was 312 nm. The experiment was repeated three times independently with similar results. (B) The UV-Vis absorption of MnO_2 . $c(g-C_3N_4) = 25 \mu g/mL$, $c(MnO_2) = 100 \mu g/mL$.

13. In animal experiment, the authors have chosen 24 h after nanomedicine to irradiate with light. Based on what data was this time point chosen? Is the accumulation higher at tumor after 24 h? biodistribution?

Response:

We are sorry for the unclear description on the animal experiment. According to the comment, the additional experiment on examining the time-dependent biodistribution of the nanomedicine has been carried out. In the experiment, the content of Mn was determined by ICP detections for evaluating the content of the nanomedicine in tumor at 12, 24 and 48 h, respectively. As shown in Supplementary Figure 38, the nanomedicine can be enriched at tumor after 12 h and no significant decrease was observed at 24 h. Consequently, to support the generation of C_3N_4 -Mn SACs upon the reaction of precursors with GSH at tumor sites, the irradiation of PDT was applied after 24 h of the injection.

Therefore, the corresponding section has been revised as follows:

(Page 25, Line 501-505, Highlighted Revised Manuscript)

... The accumulation of C_3N_4 - MnO_2 at the tumor site was revealed in Supplementary Fig. 38, which was attributed to the EPR effect. As displayed, no significant decrease of the Mn concentration was obtained in the tumor at 24 h compared to that after 12 h of the injection. To support the synthesis of C_3N_4 -Mn SACs following the reaction of precursors with GSH

after the enrichment at tumor sites, the irradiation of PDT was performed after 24 h of the injection.

(Page S29, Supplementary Figure 38, Revised Supplementary Information)

Supplementary Figure 38. The biodistribution of $C_3N_4-MnO_2$ by measuring Mn concentrations over 12, 24, and 48 h after the intravenous injection.

14. According to Supplementary Fig. 20, the $\bullet OH$ signal is silent in the C_3N_4+L group. However, in Fig. 5E, F, G, C_3N_4+L shows toxicity in Hela cells. Maybe the source of toxicity is from singlet oxygen. It is better to include an additional group (C_3N_4+L) in Supplementary Fig. 22.

Response:

According to the comment, the additional experiments have been carried out for clearing the ROS species generated from the light irradiation of C_3N_4 (C_3N_4+L group). The specific reagent of DPBF for detecting singlet oxygen was selected, upon the oxidation by singlet oxygen to generate products with absorption at 410 nm (Supplementary Figure 34-B). As resulted (Supplementary Figure 34-A), the significant increased absorption at 410 nm was recorded after 30 min of the light irradiation on C_3N_4 , confirming the generation of singlet oxygen. Besides, the EPR analysis of singlet oxygen was carried out with TEMP as the trapping agent. As shown in Supplementary Figure 34-C, the significant triple peak of $TEMPO\bullet$ was observed in EPR spectrum after light irradiation for 30 min, verifying the presence of singlet oxygen in the additional group of C_3N_4+L .

Consequently, Supplementary Figure 34 and corresponding sentences have been added in the revised manuscript as follows:

(Page 18, Line 360-363, Highlighted Revised Manuscript)

... As shown in Fig. 4F, dramatically lower ROS levels were resulted in the C_3N_4 group under hypoxic conditions than that obtained in normoxic environments. This was reasonably attributed to the oxygen-dependent generation of singlet oxygen by C_3N_4 (Supplementary Fig. 34). ...

(Page S26, Supplementary Figure 34, Revised Supplementary Information)

Supplementary Figure 34. (A) UV-vis spectra of DPBF in C_3N_4 after irradiated for 30 min (635 nm, 0.4 W/cm^2). The experiment was repeated twice independently with similar results. (B) Schematic representation of 1O_2 detection by DPBF. (C) EPR spectra of C_3N_4 before and after the light irradiation (635 nm, 0.4 W/cm^2) for 30 min. TEMP acted as the trapping agent. (D) The schematic illustration on the generation of 1O_2 by C_3N_4 under light irradiation. $c(C_3N_4) = 100\ \mu g/mL$, $c(DPBF) = 10\ \mu M$.

(Page S26, Revised Supplementary Information)

As shown in Supplementary Fig. 34, C_3N_4 can convert O_2 to 1O_2 under 635 nm light irradiation, which facilitated the Type-II PDT. The specific reagent of DPBF was selected for detecting singlet oxygen, upon the absorption of the singlet oxygen oxidized produced at 410 nm (Supplementary Figure 34B). As a result (Supplementary Figure 34A), the significantly increased absorption at 410 nm was recorded after 30 min of the light irradiation on C_3N_4 , confirming the generation of singlet oxygen. Besides, the EPR analysis of singlet oxygen was carried out with TEMP as the trapping agent. As shown in Supplementary Figure 34C, the significant triple peak of $TEMPO\cdot$ was observed in the EPR spectrum after light irradiation for 30 min, verifying the presence of singlet oxygen. These results are in accordance with the light irradiated producing of ROS by C_3N_4 under normoxia relative to the failure of ROS generation under hypoxia.

15. For the cell experiments, the mechanism for *in situ* synthesized C_3N_4 -Mn SACs was GSH reacting with MnO_2 to form a single atom Mn on C_3N_4 . Cells with negative GSH should be considered.

Response:

As suggested by the Reviewer, the additional experiment on evaluating the *in situ* synthesis in the cells with negative GSH has been carried out. The HUV-EC cells with negative GSH was selected for evaluating the feasibility of the *in situ* synthesis of SACs, which was evaluated by cell imaging. As shown in Figure 4C, no significant blue FL signal was observed for the “HUV-EC” cells after incubated with the “ C_3N_4 - MnO_2 ” group. This demonstrated that MnO_2 was still presented on C_3N_4 , kept quenching blue FL signal of C_3N_4 by MnO_2 (Supplementary Figure 9). Consequently, the failure of *in situ* synthesis of C_3N_4 -Mn SACs was confirmed in the cells without GSH. This would endow the low-toxicity to normal cells, which was verified by the MTT assay in the revised Figure 6B.

According to the comment, Figure 4C and Figure 6B have been revised and the corresponding section has been revised as follows:

(Page 15, Line 308-321, Highlighted Revised Manuscript)

To further study the feasibility of intracellular synthesis of C_3N_4 -Mn SACs, HeLa cells with upregulated GSH were selected as models of cancer cells. Meanwhile, HUV-EC cells with negative GSH acted as a control to simulate normal tissue cells. Firstly, cell imaging by Confocal Laser Scanning Microscopy (CLSM) was employed to examine the *in situ* synthesis

of C₃N₄-Mn SACs in HeLa and HUV-EC cells with different treatments. As illustrated in Fig. 4B, treated with C₃N₄-MnO₂, blue signals of *in situ* synthesized C₃N₄-Mn SACs were observed on membranes and nuclei of HeLa cells. This could be attributed to the formation of sandwiched superstructures on the phospholipid bilayer (cell membrane and nuclear membrane).^{20, 41} The *in situ* synthesis of SACs was validated by the identical cell images treated with the *in situ* synthesized (Fig. 4B, C₃N₄-MnO₂) and the pre-synthesized (Fig. 4B, C₃N₄-Mn) C₃N₄-Mn SACs. While after being treated with the GSH elimination reagent of N-Ethylmaleimide (NEM, 1 mM), no significant signal was observed in HeLa cells (Fig. 4B, NEM+C₃N₄-MnO₂). This revealed that C₃N₄-MnO₂ precursors could not be degraded to produce SACs without GSH. Similarly, in normal cells without high expression of GSH, the blue signal of C₃N₄-Mn SACs was not observed (Fig. 4C, C₃N₄-MnO₂). This revealed that the GSH-induced *in situ* synthesis of C₃N₄-Mn SACs was tumor-specific. ...

(Page 17, Figure 4C, Highlighted Revised Manuscript)

Fig. 4. Investigation on the GSH-induced *in situ* synthesis of C_3N_4-Mn SACs and the intracellular ROS generation in HeLa cells. ... (B) CLSM images of HeLa cells incubated with PBS, $C_3N_4-MnO_2$ (*in situ* synthesis of SACs), pre-synthesized C_3N_4-Mn , and $C_3N_4-MnO_2+NEM$ (without *in situ* synthesis). $\lambda_{ex}=408$ nm. (C) CLSM images of HUV-EC cells incubated with PBS, $C_3N_4-MnO_2$ (*in situ* synthesis of SACs), and pre-synthesized C_3N_4-Mn . $\lambda_{ex}=408$ nm. ...

(Page 21, Figure 6B, 6C, 6D, Highlighted Revised Manuscript)

Fig. 6. Investigation on the intracellular LPO by *in situ* synthesized C_3N_4-Mn SACs in cancer cells. (A) Scheme of $\cdot OH$ generation and LPO by *in situ* synthesized C_3N_4-Mn SACs (obtained from intracellular $C_3N_4-MnO_2$ reduction). **(B-D)** MTT assay of HeLa and HUV-EC cells incubated with **(B)** C_3N_4 , **(C)** $C_3N_4-MnO_2$ precursors, **(D)** C_3N_4-Mn SACs.

16. I think Supplementary Fig. 12 was mislabeled, and the XRD pattern for the $C_3N_4-MnO_2$ group was missed.

Response:

We are sorry for the unsatisfied XRD spectra. Just as the Reviewer indicated, Supplementary Fig. 12 was mislabeled. As suggested, the original Supplementary Fig. 12 has been revised by adding the XRD pattern of $C_3N_4-MnO_2$ group.

The revised figure was added in the revised manuscript and renamed as Figure 2B:

(Page 9, Figure 2B, Highlighted Revised Manuscript)

Fig. 2. Fine structure of Mn atom in the C_3N_4 -Mn SACs. (A) FTIR spectra of g- C_3N_4 nanosheets, C_3N_4 - MnO_2 precursor, and C_3N_4 -Mn SACs. (B) XRD patterns of bulk C_3N_4 , g- C_3N_4 , C_3N_4 - MnO_2 , MnO_2 , and C_3N_4 -Mn SACs. (C) N 1s XPS spectra of C_3N_4 -Mn SACs and g- C_3N_4

17. Supplementary Fig. 16 is incorrect; it shows a CV spectrum, not an XPS spectrum, and the labels A-D are missing. Please provide the correct figure.

Response:

We are sorry for the mistake. As suggested, Supplementary Fig. 16 has been corrected as follows:

(Page S14, Supplementary Figure 16, Revised Supplementary Information)

Supplementary Figure 16. CV curves of Mn^{2+} and g- C_3N_4 with or without light irradiation (660 nm, 0.4 W/cm^2). $c(\text{C}_3\text{N}_4)=50 \mu\text{g}/\text{mL}$

18. The labels in figure must be presented more clearly and I suggest to increase the font size inside the Figures.

Response:

We appreciate the Reviewer’s comment. As suggested, all the figures have been revised for the better presentation. The revised figures are shown in the Revised Manuscript and Revised Supporting Information.

19. P-3, line 62, “Cell experiment indicated that C_3N_4 -Mn SACs were...”. Check the statement clearly.

Response:

According to the present comment and the aforementioned comments, the corresponding sentence has been revised as follows:

(Page 3, Line 57-60, Highlighted Revised Manuscript):

... To implement the *in situ* synthesis and oxygen-independent generation of $\cdot\text{OH}$, C_3N_4 - MnO_2 was enriched at the tumor site through the enhanced permeability and retention (EPR)

effect. Subsequently, the *in situ* release of C_3N_4 and Mn^{2+} from $C_3N_4-MnO_2$ was employed, which responded to the upregulated GSH in TME. ...

20. Line 66, spelling error – adequate; line 203, charge separation.

Response:

We are very sorry for the spelling errors and the corresponding spelling errors have been corrected. In addition, the spelling and grammar of the whole manuscript have been carefully brushed. We hope that the revised manuscript is substantially improved from a linguistic point of view.

21. At least three repeats of the results should be provided in Supplementary Fig. 31. The label in Supplementary Fig. 29 (page 25) is wrong; it should be Supplementary Fig. 32.

Response:

Thank you very much for pointing out the mistakes. As suggested, Supplementary Fig. 31 in the original manuscript has been revised and shown as Supplementary Fig. 30 in the revised version. In addition, the label in Supplementary Fig. 29 (named as Supplementary Fig. 32 in the revised Supporting Information) has been revised.

The revised figure is shown as follows:

(Page S23, Supplementary Figure 30, Revised Supplementary Information)

Supplementary Figure 30. The generation of $\cdot OH$ by C_3N_4-Mn after 30 min irradiation (660 nm, $0.4 W/cm^2$), which was evaluated by FL intensities of TA with and without methanol as the hole scavenger. Data are presented as mean \pm SD ($n=3$). $c(C_3N_4-Mn) = 10 \mu g/mL$, $c(TA) = 10 \mu M$.

Reply to Reviewer 2's Comments

Comments to the Author

The authors have reported their work on Mn embedded g-C₃N₄ with photodynamic therapy a precise cancer treatment, which looks good for the bio-part. As I am not an expert in biochemistry or cancer, I would make some comments on the materials' characterizing. As shown in Figure 1f and 1k, it is hard to rule out MnO@g-C₃N₄, which is close to the experimentally reported TiO@g-C₃N₄ (DOI: 10.1039/C5SC03551A). Also, it is well known g-C₃N₄ is not a planar structure, which is reported both in the experiments (DOI: 10.1021/jp510187c, DOI: 10.1039/C3RA45776A, DOI: 10.1039/c3ta13188b) and in the calculations (DOI: 10.1007/s11467-018-0754-6, DOI: 10.1021/acs.jpcc.0c00411). It is thus believed that data from DFT calculations in this work should be questionable. One can see that either experimental or theoretical data could not support the authors' statement. I cannot recommend the current manuscript for publication.

I would like to suggest the authors to do a more rigorous calculation considering both Mn@g-C₃N₄ and MnO@g-C₃N₄ with a corrugated model for a more precise analysis on the materials. And then revised the manuscript for resubmitting.

Response:

Thank you very much for the comments from the Reviewer. We are sorry for the unsatisfied characterization and demonstration on the fine structure of Mn in C₃N₄-Mn SACs. In fact, the present C₃N₄-Mn SACs were obtained by the reduction of MnO₂ by GSH to generate Mn²⁺ on the surface of C₃N₄. Thereafter, the generated atomically dispersed Mn²⁺ was coordinated to C₃N₄ via a hexadentate ligand, facilitating the *in situ* synthesis of C₃N₄-Mn SACs within TME. Therefore, the Mn-O bonds could not be obtained during this synthesis process. As characterized, no characteristic peak of Mn-O bond was observed in FT-IR (Figure 2A) and Raman (Supplementary Figure 12) spectra, which preliminarily proved the absence of Mn-O in C₃N₄-Mn SACs. To further confirm the presence of Mn-N bonds rather than Mn-O, the fine structures of Mn in C₃N₄-Mn SACs have been re-examined upon EXAFS data. Furthermore, quantitative EXAFS curve fitting analysis in R spaces was employed to accurately determine the coordination structure of Mn atom. As demonstrated by the fitting results (Figure 2F), the peak around 1.97 Å was ascribed to the coordination of Mn single atoms with N atoms in C₃N₄-Mn SACs (Supplementary Table 1). This coordination of single atoms with N atoms of C₃N₄ was similar to that of the reported C₃N₄-Cu SACs (DOI: 10.1002/anie.202207677). Besides, upon comparing WT data of C₃N₄-Mn SACs (Figure 2G), Mn foil (Figure 2H) and MnO₂ (Figure 2I), the

long-range interaction of Mn-N-C was demonstrated, further verifying the obtaining of C₃N₄-Mn SACs instead of C₃N₄-MnO ones.

Consequently, the original Figure 1 has been revised into Figure 2 and the corresponding descriptions have been revised as follows:

(Page 8, Line 147-164, Highlighted Revised Manuscript):

To identify the local fine structure of Mn at the atomic level, C₃N₄-Mn SACs were further examined by X-ray Absorption Spectroscopy (XAS). As demonstrated by spectra of Mn K-edge X-ray Absorption Near-Edge Structure (XANES, Fig. 2D), the absorption edge of Mn in C₃N₄-Mn SACs placed between that of Mn foil and MnO₂, showing that the valence state of Mn was close to +2. The coordination between Mn and N in C₃N₄-Mn was demonstrated by the phase-uncorrected Fourier transformed Extended X-ray Absorption Fine Structure (EXAFS) characterization. As shown in Fig. 2E, Mn foil presented the main peak at 2.3 Å, corresponding to the Mn-Mn bonds. While no corresponding Mn-Mn signal was observed in C₃N₄-Mn SACs, indicating the presence of atomically dispersed Mn. Furthermore, the signal at 1.4 Å could be ascribed to the Mn-N bonds, which was in accordance with N 1s XPS spectra (Fig. 2C and Supplementary Fig. 13). **To accurately characterize the coordination structure of the Mn atom, quantitative EXAFS curve fitting analysis in R spaces was employed. As depicted in Fig. 2F, the best-fitting results demonstrate that the peak of about 1.97 Å was ascribed to the coordination of Mn single atoms with N atoms in C₃N₄-Mn SACs (Supplementary Table 1).** The Wavelet Transform (WT) results further indicated that there was no Mn-Mn bond in C₃N₄-Mn (Fig. 2G, 2H, 2I). The main peak of C₃N₄-Mn tended to have a lower *k* value (~ 5.3 Å⁻¹) than that of Mn-O in MnO₂ (~ 8.0 Å⁻¹), confirming the presence of Mn-N bond in C₃N₄-Mn SACs. **In addition, the peaks of Mn-N-C with long lengths (~ 3.3 Å) were observed, which could be attributed to the long-range interaction of Mn-N-C in C₃N₄-Mn SACs.** Therefore, the detailed Mn coordination by N in C₃N₄-Mn was strongly confirmed by the XAFS tests.

(Page 9, Figure 2F, Highlighted Revised Manuscript)

Fig. 2. Fine structure of Mn atom in the C_3N_4 -Mn SACs. (A) FTIR spectra of g- C_3N_4 nanosheets, C_3N_4 - MnO_2 precursor, and C_3N_4 -Mn SACs. (B) XRD patterns of bulk C_3N_4 , g- C_3N_4 , C_3N_4 - MnO_2 , MnO_2 , and C_3N_4 -Mn SACs. (C) N 1s XPS spectra of C_3N_4 -Mn SACs and g- C_3N_4 . (D) XANES and (E) EXAFS spectra of C_3N_4 -Mn SACs, MnO_2 , and Mn foil at the Mn K-edge. (F) EXAFS fitting curves of C_3N_4 -Mn SACs in R space. WT of (G) C_3N_4 -Mn SACs, (H) Mn foil, and (I) MnO_2 .

(Page S9, Supplementary Table 1, Revised Supplementary Information)

Sample	Shell	CN ^a	R (Å) ^b	σ^2 (Å ²) ^c	ΔE_0 (eV) ^d	R factor
C_3N_4 -Mn SACs	Mn-N	3.5	1.97	0.0019	3.47	0.008

^a CN, coordination number; ^b R, the distance between absorber and backscatter atoms; ^c σ^2 , DebyeWaller factor to account for both thermal and structural disorders; ^d ΔE_0 , inner potential correction;

R factor indicated the goodness of the fit. S_0^2 was fixed to 0.8. Error bounds that characterize the structural parameters obtained by EXAFS spectroscopy were estimated as $CN \pm 20\%$; $\sigma^2 \pm 20\%$; $R \pm 0.04\text{\AA}$

In addition, we really appreciate the Reviewer's comment on the DFT calculations. Just as the Reviewer indicated, g-C₃N₄ is not a planar structure. Therefore, as suggested by the Reviewer, a more rigorous DFT calculation considering all the C₃N₄-related materials with a corrugated model has been carried out in the whole revised manuscript. Just as the Reviewer indicated, C₃N₄ is indeed not a planar structure, which could decrease the energy of molecular orbitals and excited states due to the lower symmetry. Consequently, the molecular orbitals and the LMCT process of C₃N₄-Mn SACs have been re-evaluated and discussed based on the new structures. The corresponding revisions are shown in all the DFT-related sections, including the corresponding contents in Page 10-14 (Revised Manuscript) and Page S10-S17, S24-S25 (Revised Supporting Information). All the data including Figure 3E, Supplementary Figure 14, 15, 17, 18, 19, 20, 32, and Supplementary Table 2, 3 have also been updated.

Reply to Reviewer 3's Comments

Comments to the Author

1. Even though the authors seem to give the impression that their C₃N₄-Mn SAC material is novel, the fact is that it is not. Materials with very similar (if not the same) structure have been repeatedly reported (e.g. Mo et al. *App. Catal. B, Envir*, 2019; Wang et al. *ACS Sust. Chem. Eng.* 2018; Guo et al. *JACS*, 2019, to name a few). These manuscripts demonstrated SAC structure, often resorting to EXAFS/XANES, as in the manuscript. Even the preparation of the pre-catalyst (MnO₂ on top of the C₃N₄ layers) has been reported, using the same precursors, exfoliated C₃N₄ and KMnO₄ (Mo et al *Appl Catal B* 2019). Amazingly, they are not quoted, even though the material and the reactions reported are relevant to this work.

Response:

Thank you very much for the comments from the Reviewer. We are sorry for the unclear demonstration on the perspective of the present strategy. Just as the reviewer indicated, C₃N₄-based SACs or MnO₂ on top of the C₃N₄ layers have been used in the field of catalysis. Nevertheless, we did not declare the fabrication of a novel material in the present work. While the main perspective of the present work was tumor-specific *in situ* synthesis of SACs within TME, which subsequently exhibited red-light harvesting for efficient O₂-independent generation of ·OH via water-splitting. This strategy not only overcame the toxicity from random distribution and catalyst release in healthy tissues, but also initiated the O₂-independent generation of highly toxic ·OH from the “inert” H₂O for cancer-specific PDT. Consequently, this strategy achieves the efficient and precise cancer therapy, not only overcoming tumor hypoxia, but also avoiding the side effects on normal tissues.

To make a better demonstration on the perspective of the present strategy, the corresponding section has been revised as follows:

(Page 1, Line 12-22, Highlight Revised Manuscript)

Single-atom catalysts (SACs) have attracted interest in photodynamic therapy (PDT), while they are normally limited by the side effects on normal tissues and the interfaces from the Tumor Microenvironment (TME). **Herein, tumor-activated *in situ* synthesis of SACs was reported to achieve the efficient tumor-specific water-based PDT. Upon *in situ* reduction by upregulated GSH in TME, C₃N₄-Mn SACs were obtained in TME with Mn atomically coordinated into the N₆-macroheterocycle cavity of C₃N₄ nanosheets. This *in situ* synthesis overcomes toxicity from random distribution and catalyst release in healthy tissues. Based**

on Ligand-to-Metal charge transfer (LMCT), C₃N₄-Mn SACs exhibited red-light harvesting for effective water splitting. This initiated the O₂-independent generation of highly toxic hydroxyl radical (\cdot OH) for cancer-specific PDT. Subsequently, the \cdot OH-initiated lipid peroxidation process was demonstrated to devote effective cancer cell death. The *in situ* synthesized SACs facilitated the precise cancer-specific conversion of “inert” H₂O to “reactive” \cdot OH. This strategy achieved efficient and precise cancer therapy, not only overcoming tumor hypoxia, but also avoiding the side effects on normal tissues.

2. Nor is the generation of ROS with these materials new. Mn-C₃N₄ has been reported to generate superoxide, singlet oxygen and hydroxyl radicals, depending on the conditions used.

Response:

We are sorry for the unclear demonstration on the perspective of the generation of hydroxyl radical by the present strategy. Although ROS has been reported by series of materials for PDT, the therapeutic efficacy of O₂-dependent PDT is significantly influenced by tumor hypoxia, limiting the generation of therapeutic ROS reagents without sufficient O₂ (Ref 14). This is particularly serious in the Type-II PDT with O₂-related ROS (like ¹O₂) generated by O₂-dependent photosensitizers. Alternatively, Type-I PDT could relieve the dependence on intracellular O₂ by developing \cdot OH through the Fenton-like oxidation of H₂O₂ (Ref 15, 16). However, it was still limited by the insufficient endogenous H₂O₂ at tumor sites (Ref 17). Therefore, generating adequate ROS species independent of endogenous O₂ and H₂O₂ would greatly avoid the effect of tumor hypoxia and boost the PDT efficiency in TME. Consequently, a higher requirement for O₂-independent generation of ROS using the *in situ* synthesized SACs in cancer sites arises. This would become more desirable with water as the simple oxygen source for efficient PDT across tumors.

Significantly, the present work reported the red-light irradiated water splitting to efficiently generate \cdot OH at tumor sites by the *in situ*-synthesized SACs. By virtue of LMCT and the good penetrability of red light, the O₂-independent generation of ROS was realized for efficient PDT. This would greatly overcome tumor hypoxia and avoid side effects on normal tissues. This is one of the main advantages of the present work, relative to the traditional O₂-dependent generation of ROS for PDT.

Consequently, for the clear demonstration, the corresponding section has been revised in the revised manuscript as follows:

(Page 2-3, Line 45-69, Highlighted Revised Manuscript):

Particularly, the therapeutic efficiency of ROS-dependent PDT is significantly influenced by tumor hypoxia, limiting the generation of therapeutic ROS reagents without sufficient O₂.¹⁴ This is even significant in the Type-II PDT with O₂-related ROS (like ¹O₂) generated by O₂-dependent PSs. Alternatively, Type-I PDT could relieve the dependence on intracellular O₂ by generating ·OH through the Fenton-like oxidation of H₂O₂.^{15, 16} However, it was still limited by the insufficient endogenous H₂O₂ at tumor sites.¹⁷ Therefore, generating adequate ROS species independent of endogenous O₂ and H₂O₂ would considerably increase the PDT efficiency in TME. Consequently, a necessity for O₂-independent generation of ROS via the *in situ*-synthesized SACs in cancer sites arises. This would become more desired with water as the simple oxygen source for efficient PDT across tumors.

Herein, a single-atom catalyst (C₃N₄-Mn SACs) was *in situ* prepared in TME for the highly specific Type-I PDT upon the oxygen-independent generation of ·OH with minimal invasiveness. Firstly, a nanomedicine precursor of 2D/2D C₃N₄-MnO₂ was prepared, which was lowly toxic to normal tissues (such as the kidney and liver) in the metabolism (Scheme 1). To implement the *in situ* synthesis and oxygen-independent generation of ·OH, C₃N₄-MnO₂ was enriched at the tumor site through the enhanced permeability and retention (EPR) effect. Subsequently, the *in situ* release of C₃N₄ and Mn²⁺ from C₃N₄-MnO₂ was employed, which responded to the upregulated GSH in TME. Consequently, atomically dispersed Mn²⁺ was captured by C₃N₄ as a hexadentate ligand, facilitating the *in situ* synthesis of C₃N₄-Mn SACs within TME. Especially based on Ligand-to-Metal charge transfer (LMCT) from C₃N₄ to Mn²⁺, C₃N₄-Mn SACs exhibited red-light harvesting for efficient O₂-independent generation of ·OH via water-splitting. The red shift of the absorption also facilitated the red light-irradiated applications (660 nm) with greater penetration depth than that of white light in PDT.¹⁸ Thereby, being the most toxic ROS, adequate ·OH induced efficient cancer cell death through the powerful Lipid peroxidation (LPO) process. Furthermore, PDT mechanisms have been studied by extracellular and intracellular tests, and the *in vivo* therapeutic effects were further validated. Therefore, this work would inspire efficient, tumor-specific, and O₂-independent PDT through the *in situ* synthesis of SACs.

3. The main claim of the manuscript, the selective generation of the catalyst inside cancer cells is far from demonstrated. According to the authors the MnO₂/C₃N₄ construct would be reduced by GSH

and since this is overexpressed in cancer cells the reduction that gives rise to the synthesis of the catalyst would be selective, and the catalyst (with the corresponding generation of .OH) would only take place within cancer cells. This is a central claim of the manuscript. However, the selectivity premise is wrong, since the MnO₂ is likely to be reduced by many other molecules present in biological environments. And in any case, although GSH is somewhat overexpressed in cancer cells, there are significant concentrations also in healthy cells (see for instance Bonet et al. Chemical Sci. 2022). Therefore, to be able to claim that the catalyst is only activated in the TME a demonstration with a large number of cancer and non cancer cells is needed, plus other representative environments (plasma, DMEM).

Response:

As suggested by the Reviewer, the additional experiments on the evaluation within cancer and non-cancer cells, as well as other representative environments of plasma, serum and DMEM culture medium have been added in the revised manuscript. In the revised works, the non-cancer cells of HUV-EC cells were selected for evaluating the feasibility of the *in situ* synthesis of SACs, which was evaluated by cell imaging. As shown in Figure 4C in the Revised Manuscript, no significant blue FL signal was observed for the “HUV-EC” cells after incubated with the “C₃N₄-MnO₂” group. This demonstrated that MnO₂ was still presented on C₃N₄, kept quenching blue FL signal of C₃N₄ by MnO₂ (Supplementary Figure 9). Consequently, the failure of *in situ* synthesis of C₃N₄-Mn SACs was confirmed in the non-cancer cells without GSH. This would endow the low-toxicity to normal cells, which was verified by the MTT assay in the revised Figure 6B.

In addition, the feasibility of *in situ* synthesis of SACs in other representative environments of plasma and culture medium have been carried out. In the experiments, the UV-vis absorption of C₃N₄-MnO₂ and C₃N₄-Mn were recorded after incubating in DMEM culture medium, FBS and mouse plasma for 0 to 4 h. As resulted, C₃N₄-Mn exhibited the good light and dark stability which would benefit the efficient PDT (Supplementary Figure 33A-C in Revised Supporting Information). In addition, C₃N₄-MnO₂ exhibited the good stability in the FBS and plasma without obvious change on the UV-vis absorption (Supplementary Figure 33D-F), which would benefit the tumor-specific *in situ* synthesis of C₃N₄-Mn SACs and subsequent efficient tumor-specific PDT.

Consequently, the corresponding section has been added in the revised manuscript as follows:

(Page 15, Line 308-321, Highlighted Revised Manuscript)

To further study the feasibility of intracellular synthesis of C₃N₄-Mn SACs, HeLa cells with upregulated GSH were selected as models of cancer cells. Meanwhile, HUV-EC cells

with negative GSH acted as a control to simulate normal tissue cells. Firstly, cell imaging by Confocal Laser Scanning Microscopy (CLSM) was employed to examine the *in situ* synthesis of C_3N_4 -Mn SACs in HeLa and HUV-EC cells with different treatments. As illustrated in Fig. 4B, treated with C_3N_4 -MnO₂, blue signals of *in situ* synthesized C_3N_4 -Mn SACs were observed on membranes and nuclei of HeLa cells. This could be attributed to the formation of sandwiched superstructures on the phospholipid bilayer (cell membrane and nuclear membrane).^{20, 41} The *in situ* synthesis of SACs was validated by the identical cell images treated with the *in situ* synthesized (Fig. 4B, C_3N_4 -MnO₂) and the pre-synthesized (Fig. 4B, C_3N_4 -Mn) C_3N_4 -Mn SACs. While after being treated with the GSH elimination reagent of N-Ethylmaleimide (NEM, 1 mM), no significant signal was observed in HeLa cells (Fig. 4B, NEM+ C_3N_4 -MnO₂). This revealed that C_3N_4 -MnO₂ precursors could not be degraded to produce SACs without GSH. Similarly, in normal cells without high expression of GSH, the blue signal of C_3N_4 -Mn SACs was not observed (Fig. 4C, C_3N_4 -MnO₂). This revealed that the GSH-induced *in situ* synthesis of C_3N_4 -Mn SACs was tumor-specific. ...

(Page 17, Figure 4C, Highlighted Revised Manuscript)

Fig. 4. Investigation on the GSH-induced *in situ* synthesis of C_3N_4-Mn SACs and the intracellular ROS generation in HeLa cells. ... (B) CLSM images of HeLa cells incubated with PBS, $C_3N_4-MnO_2$ (*in situ* synthesis of SACs), pre-synthesized C_3N_4-Mn , and $C_3N_4-MnO_2+NEM$ (without *in situ* synthesis). $\lambda_{ex}=408$ nm. (C) CLSM images of HUV-EC cells incubated with PBS, $C_3N_4-MnO_2$ (*in situ* synthesis of SACs), and pre-synthesized C_3N_4-Mn . $\lambda_{ex}=408$ nm. ...

(Page 15, Line 300-307, Highlighted Revised Manuscript)

Generally, the satisfied dark stability and light stability of the photocatalytic materials are essential premises for efficient PDT. As indicated (Supplementary Fig. 33), C_3N_4-Mn SACs showed excellent stability in DMEM cell culture medium and fetal bovine serum (FBS). Furthermore, C_3N_4-Mn also exhibited no photodegradation under red light irradiation (660 nm) in PBS. Meanwhile, to ensure the selective *in situ* synthesis in TME, $C_3N_4-MnO_2$ precursors

should have strong stability and could not be degraded in the non-cancer environments without upregulated GSH. This was confirmed by the good stability of $C_3N_4-MnO_2$ in FBS and plasma (Supplementary Fig. 33), which would benefit the tumor-specific *in situ* synthesis of C_3N_4-Mn SACs and subsequent efficient PDT for cancer.

(Page 25, Supplementary Figure 33, Revised Supplementary Information)

Supplementary Figure 33. UV-vis absorption of C_3N_4-Mn in DMEM Culture medium (A) and FBS (B) from 0 to 4 h. (C) UV-vis absorption of C_3N_4-Mn in PBS after light irradiation (660 nm, 0.4 W/cm²) for 0 to 40 min. UV-vis absorption of $C_3N_4-MnO_2$ in DMEM culture medium (D), FBS (E), and mouse plasma (F) from 0 to 4 h. The experiment was repeated twice independently with similar results.

(Page 22, Line 439-452, Highlighted Revised Manuscript)

To test the light and dark toxicity of C_3N_4-Mn SACs, methylthiazolyldiphenyltetrazolium bromide (MTT) assay was carried out for HeLa and HUV-EC cells incubated with different concentrations of C_3N_4 , $C_3N_4-MnO_2$, and C_3N_4-Mn SACs under red light irradiation or in dark. Firstly, C_3N_4 , $C_3N_4-MnO_2$, and C_3N_4-Mn demonstrated low dark toxicity to both HeLa cells and HUV-EC cells (Fig. 6B, 6C, 6D), showing the good biocompatibility of the medicine precursors. As revealed, C_3N_4 showed minimal light toxicity toward both HeLa and HUV-EC

cells (Fig. 6B). Relatively, C_3N_4 -Mn SACs demonstrated a potent ablation for both the HeLa cells and HUV-EC cells with an IC_{50} value of about $[C_3N_4] = 2.0 \mu\text{g/mL}$ (Fig. 6C). In addition, upon 660 nm of the irradiation, the *in situ* synthesized C_3N_4 -Mn SACs (from C_3N_4 - MnO_2) demonstrated improved cytotoxicity (IC_{50} value of about $[C_3N_4] = 1.0 \mu\text{g/mL}$) for HeLa cells than both C_3N_4 and pre-synthesized C_3N_4 -Mn. This revealed the effective Type-I PDT after the introduction of single Mn atoms, which was further enhanced by GSH depletion. Notable, light toxicity of C_3N_4 - MnO_2 for HUV-EC cells was inhibited, showing that the tumor-specific *in situ* synthesis of C_3N_4 -Mn can selectively kill cancerous HeLa cells over normal HUV-EC cells. Therefore, tumor-specific *in situ* synthesis of C_3N_4 -Mn SACs effectively facilitated the tumor-selective Type-I PDT.

(Page 21, Figure 6B, 6C, 6D, Highlighted Revised Manuscript)

Fig. 6. Investigation on the intracellular LPO by *in situ* synthesized C_3N_4 -Mn SACs in cancer cells. (A) Scheme of $\cdot OH$ generation and LPO by *in situ* synthesized C_3N_4 -Mn SACs (obtained from intracellular C_3N_4 - MnO_2 reduction). (B-D) MTT assay of HeLa and HUV-EC cells incubated with (B) C_3N_4 , (C) C_3N_4 - MnO_2 precursors, (D) C_3N_4 -Mn SACs. ...

4. Related to this, it seems that the C₃N₄-Mn particles stayed mainly in the cancer cell membranes (something that the authors see as the reason of the lipidic oxidation observed). However, in this case they would not be exposed to the "high" GSH concentrations inside the cell and the catalyst would not be formed.

Response:

We are sorry for the misleading data of cell imaging in the original manuscript. In the old version of the cell imaging, the time for incubating cells with the C₃N₄-Mn nanomaterials was 4 h, which was not adequate enough for the endocytosis. According to the comments, the additional cell imaging experiments have been carried out to clear the location of the C₃N₄-Mn SACs. In the experiments, cell imaging was employed after incubating HeLa cells with different groups for 8 h, including groups of PBS, C₃N₄-MnO₂ (for the *in situ* synthesis of SACs), pre-synthesized C₃N₄-Mn, and C₃N₄-MnO₂+NEM (without *in situ* synthesis due to the absence of GSH). As shown in Figure 4B in the Revised Manuscript, treated with C₃N₄-MnO₂ for 8 h, blue signals of *in situ* synthesized C₃N₄-Mn SACs were observed on membranes and nucleus of HeLa cells. The similar images were also exhibited in the pre-synthesized C₃N₄-Mn group. In addition, although the concentration of GSH in TME is not as high as the intracellular one, it is still sufficient to complete the *in situ* synthesis of SACs. Consequently, the C₃N₄-Mn SACs can be *in situ* synthesized in TME with upregulated GSH, and would facilitate the subsequent LPO for selective and efficient PDT.

According to the comment, the corresponding section has been revised as follows:

(Page 15, Line 312-321, Highlighted Revised Manuscript):

... As illustrated in Fig. 4B, treated with C₃N₄-MnO₂, blue signals of *in situ* synthesized C₃N₄-Mn SACs were observed on membranes and nuclei of HeLa cells. This could be attributed to the formation of sandwiched superstructures on the phospholipid bilayer (cell membrane and nuclear membrane).^{20, 41} The *in situ* synthesis of SACs was validated by the identical cell images treated with the *in situ* synthesized (Fig. 4B, C₃N₄-MnO₂) and the pre-synthesized (Fig. 4B, C₃N₄-Mn) C₃N₄-Mn SACs. While after being treated with the GSH elimination reagent of N-Ethylmaleimide (NEM, 1 mM), no significant signal was observed in HeLa cells (Fig. 4B, NEM+C₃N₄-MnO₂). This revealed that C₃N₄-MnO₂ precursors could not be degraded to produce SACs without GSH. Similarly, in normal cells without high expression of GSH, the blue signal of C₃N₄-Mn SACs was not observed (Fig. 4C, C₃N₄-MnO₂). This revealed that the GSH-induced *in situ* synthesis of C₃N₄-Mn SACs was tumor-specific. ...

Fig. 4. Investigation on the GSH-induced *in situ* synthesis of C_3N_4-Mn SACs and the intracellular ROS generation in HeLa cells. (A) Illustration on the water-based photodynamic strategy for cancer therapy. With the depletion of GSH, C_3N_4-Mn SACs were *in situ* generated, and then produced $\cdot OH$ via water splitting. **(B)** CLSM images of HeLa cells incubated with PBS, $C_3N_4-MnO_2$ (*in situ* synthesis of SACs), pre-synthesized C_3N_4-Mn , and $C_3N_4-MnO_2+NEM$ (without *in situ* synthesis). $\lambda_{ex}=408\text{ nm}$. **(C)** CLSM images of HUV-EC cells incubated with PBS, $C_3N_4-MnO_2$ (*in situ* synthesis of SACs), and pre-synthesized C_3N_4-Mn . $\lambda_{ex}=408\text{ nm}$. **(D)** MS spectra of HeLa cells' lysates before and after the *in situ* synthesis of C_3N_4-Mn SACs. **(E)** Mechanisms of different PDT in the HeLa cells. ...

5. Another important claim is the water splitting under the conditions used. There is some doubts on this that are not dissipated by the results presented. For instance, if the hole-driven homolysis

takes place as depicted in fig 2., what happens to the $\cdot\text{H}$? is it desorbed as hydrogen (if so, they would have noticed, I guess), does it stay in the liquid as protons and lowers the pH of the environment? It should be noted that there are quite a few other reactions that can also give rise to $\cdot\text{OH}$, among them the peroxone reaction reported by Guo et al with the same catalyst in the presence of H_2O_2 , also present in the TME. Also, $\cdot\text{OH}$ radicals are readily formed by MnC_3N_4 from H_2O_2 , present in the TME, as shown by Wang et al. ACS Sust Chem Eng 2018.

Response:

We appreciate the Reviewer's comment on the mechanism of $\cdot\text{OH}$ generation by water splitting. According to the comment, the additional examinations have been carried out to examine what happens to the $\cdot\text{H}$ after the generation of $\cdot\text{OH}$ by water splitting. Firstly, as suggested by the Reviewer, the pH value of the therapy system was measured. Just as the Reviewer predicted, a significant decrease of pH value was recorded after the irradiation. This indicated the formation of a proton, which could be generated during the detaching of H^+ from $\cdot\text{H}$ after the water splitting. This indicated that H^+ could be generated by $\text{C}_3\text{N}_4^+(\text{H})\text{-Mn(I)}$ to decrease pH (Equation 7 in the Revised Manuscript), which would consequently leave an electron on $\text{C}_3\text{N}_4\text{-Mn SACs}$.

Secondly, to further seek the receptors of the electrons from $\text{C}_3\text{N}_4\text{-Mn SACs}$ in cells, the change of the pyruvate acids (Pyr, a potential electron receptor in cancer cells) was examined during the therapy. As detected by mass spectrometry (Supplementary Figure 31 in the Revised Supporting Information), upon the light irradiation of $\text{C}_3\text{N}_4\text{-Mn SACs}$ for PDT, Pyr was reduced and the reduced product of lactic acid (Lac) at m/z 89 ($[\text{Lac} - \text{H}]^-$) was recorded. Consequently, as a significant substance for the intracellular metabolism, Pyr can receive electrons and protons from $\text{C}_3\text{N}_4\text{-Mn SACs}$ and to be reduced into Lac (the inset Equation in Supplementary Figure 31) (Ref 40). This would therefore facilitate the subsequent catalytic cycle to enhance PDT (Equation 8 in the Revised Manuscript).

To further examine the reduction of Pyr by receiving electrons and protons from $\text{C}_3\text{N}_4\text{-Mn SACs}$, the DFT calculation has been carried out to update the energy profiles of photocatalytic water splitting (Supplementary Figure 29 in Original Supporting Information). Especially, along with the $\cdot\text{OH}$ generation, the energies of $\cdot\text{H}$ generation as well as reduction of Pyr on surfaces of C_3N_4 and $\text{C}_3\text{N}_4\text{-Mn}$ have been calculated. As shown in Supplementary Fig. 32 (in the Revised Supporting Information), C_3N_4 acted as a better hydrogen bond acceptor to adsorb water molecules (with a free energy change of -1.18 eV, 1.65 to 2.83 eV) than $\text{C}_3\text{N}_4\text{-Mn}$ (with a free energy change of 0.08 eV, 1.95 to 1.87 eV). This could be generated from the receiving of an excited electron by Mn on C_3N_4 , which slightly prevent Mn from absorbing water molecules. While given the presence of sufficient water molecules

in the solution, this would not affect the catalytic water splitting. Nevertheless, at a free energy change of -0.44 eV (1.51 to 1.95 eV), the homolysis of the O-H bond over C₃N₄-Mn is much easier than over C₃N₄ (1.12 eV, 2.77 to 1.65 eV). It is the key step in the generation of ·OH and the significant difference indicates that the introduction of Mn can significantly enhance the catalytic generation of ·OH. The desorption of ·H was also calculated, and only a small amount of ·H was observed relative to ·OH (Supplementary Fig. 27, 28), which was in accordance with what's the Reviewer indicated. Correspondingly, C₃N₄-H-Mn* exhibits a good capability to reduce Pyr, which is confirmed by the lower free energy change (-2.21 eV, 2.43 to 4.64 eV). Therefore, the water splitting catalyzed by C₃N₄-Mn SACs can generate ·OH efficiently, and subsequently facilitate the reduction of Pyr by C₃N₄-H-Mn* to initiate the following catalytic cycles. Consequently, along with the maintaining of the photocatalytic cycles, cellular respiration of cancer cells can be blocked upon consuming a significant energy substance of Pyr.

Besides, to evaluate whether ·OH can be generated from H₂O₂ as indicated by the Reviewer (Wang et al., *ACS Sustainable Chem. Eng.* 2018, 6, 8754), the nanomaterials were dispersed into H₂O₂ (rather than H₂O) for the examinations. In the experiment, the 4,4'-diamino-3,3',5,5'-tetramethyl biphenyl (TMB) probe was selected for evaluating the generation of ·OH. The increased UV-vis absorption of TMB at 371 nm and 652 nm (by the charge transfer complex from TMB to oxTMB) indicated the generation of ·OH. As shown in Supplementary Figure 29 in the Revised Supporting Information, no significant characteristic signal of ·OH was recorded in the C₃N₄-Mn SACs-H₂O₂ system. This indicated that no ·OH could be generated from H₂O₂, further confirming the generation of ·OH from the water splitting via the light-induced LMCT.

Therefore, according to the comments, the corresponding sections have been added in the revised manuscript as follows:

(Page 13-14, Line 269-279, Highlighted Revised Manuscript)

In addition, after irradiating C₃N₄-Mn for 30 min, the pH of the solution considerably dropped from 8.55 to 6.48. This suggested that in addition to producing ·H (Equation 6), H⁺ may also be generated by C₃N₄⁺(H)-Mn(I) to decrease pH values (Equation 7). As an electron donor, C₃N₄⁺(H)-Mn(I) would react with the abundant electron receptors in cancer cells to complete the catalytic cycling of C₃N₄-Mn SACs. The cycle could be accelerated by pyruvic acid (Pyr), a crucial molecule for intracellular metabolism, which can acquire electrons from C₃N₄⁺(H)-Mn(I) together with the reduction to lactic acid (Lac).⁴⁰ This was further verified by recording the MS signal of Lac at *m/z* 89 ([Lac - H]⁻) after irradiation at 660 nm

(Supplementary Fig. 31), which demonstrated that $C_3N_4^+(H)-Mn(I)$ can reduce Pyr to complete the catalytic cycle (Equation 8).

(Page S22, Supplementary Figure 29, Revised Supplementary Information)

Supplementary Figure 29. (A) UV-vis absorption of TMB in H_2O_2 with Mn^{2+} or C_3N_4-Mn SACs presented. The experiment was repeated three times independently with similar results. (B) Schematic representation of $\cdot OH$ detection by 3,3',5,5'-Tetramethylbenzidine (TMB), the increased absorption of TMB at 371 nm and 652 nm (by the charge transfer complex from TMB to oxTMB) indicated the generation of $\cdot OH$. $c(C_3N_4) = 10 \mu g/mL$, $c(TMB) = 10 \mu M$.

(Page S22, Revised Supplementary Information)

Without the irradiation, no $\cdot OH$ could be generated from the Fenton-like reaction in H_2O_2 . This could attribute to the decreased electrostatic attractions between anionic H_2O_2 and metal sites with high electron density (Supplementary Fig. 29).

(Page S23, Supplementary Figure 31, Revised Supplementary Information)

Supplementary Figure 31. The MS spectra of Pyr and C₃N₄-Mn SACs before and after light irradiation (660 nm, 0.4 W/cm²) for 30 min. c(C₃N₄-Mn) = 10 µg/mL, c(Pyr) = 1 mM.

(Page S25, Revised Supplementary Information)

The detailed reaction path of the water splitting on the surface of C₃N₄ and C₃N₄-Mn was determined. As shown in Supplementary Fig. 32, C₃N₄ behaved as a better hydrogen bond acceptor to adsorb water molecules (with a free energy change of -1.18 eV, 1.65 to 2.83 eV) than C₃N₄-Mn (with a free energy change of 0.08 eV, 1.95 to 1.87 eV). This could be generated from the reception of an excited electron by Mn on C₃N₄, which slightly prevent Mn from absorbing water molecules. While given the presence of sufficient water molecules present in the solution, this will not affect the catalytic water splitting. Nevertheless, at a free energy change of -0.44 eV (1.51 to 1.95 eV), the homolysis of the O-H bond over C₃N₄-Mn is much easier than over C₃N₄ (1.12 eV, 2.77 to 1.65 eV). It is the critical step in the generation of ·OH and the significant difference indicates that the introduction of Mn can significantly enhance the catalytic capability of ·OH generation. The desorption of ·H was also calculated, and only a small amount of ·H was observed relative to ·OH (Supplementary Fig. 27, 28). Correspondingly, C₃N₄-H-Mn* demonstrates an excellent capability to reduce Pyr, which is confirmed by the lower free energy change (-2.21 eV, 2.43 to 4.64 eV). Therefore, the water

splitting catalyzed by C_3N_4 -Mn SACs can generate $\cdot OH$ effectively, and subsequently $\cdot H$ can be released to facilitate the reduction of Pyr by C_3N_4 -H-Mn*. Consequently, along with maintaining of the photocatalytic cycles, cellular respiration of cancer cells can be blocked upon consuming a significant energy substance of Pyr.

(Page S24, Supplementary Figure 32, Revised Supplementary Information)

Supplementary Figure 32. (A) Energy profiles of photocatalytic water splitting for $\cdot\text{OH}$ generation on the surface of C_3N_4 and $\text{C}_3\text{N}_4\text{-Mn}$, along with the comparison of $\cdot\text{H}$ generation and the reduction of Pyr. Illustrations of reaction pathways of the photocatalytic water splitting over $\text{C}_3\text{N}_4\text{-Mn}$ (B) and C_3N_4 (C). Atom colors in catalyst: C (blue), N (gray), Mn (purple), O (red), H (white).

6. Other than that, there are other problems with this manuscript, starting with English language with so many flaws that sometimes the meaning is lost.

Response:

Thanks a lot for the Reviewer's comments. According to the comment, we have tried our best to brush the English language, which was once assisted by the English editing service of Mogo Internet Technology Co., LTD. We hope that the revised manuscript is substantially improved from a linguistic point of view.

Reviewers' Comments:

Reviewer #1:

Remarks to the Author:

Authors have clearly addressed my concern. I would like to recommend to accept the manuscript at current form.

Reviewer #2:

Remarks to the Author:

The authors have partially addressed my concerns, but not totally. It seems that they didn't go into the literature that I suggested, or they would find critical issues on their calculations. After going through the revised manuscript, I have the following comments.

1. Figure 1e should present the EDS pattern of O, just as being done in Supplementary Figure 4.
2. The authors stated Mn-N bond length should mostly be 1.97 Å (Supplementary Table 1), while in their model Mn-N bond length is no shorter than 2.5 Å (Supplementary Table 2). Such a large difference would greatly affect the results of DFT calculations. Also, their model is not consistent with the CN value of 3.5 in Supplementary Table 1. It is hard to know why the calculated data should match so well with the experimental data as the authors showed. Maybe it just happens to be some random event there to have such matching.
3. If the authors have read the literature that I suggest, they should know the corrugation of g-C₃N₄ should be much larger than the model they used. And in the literature, the researchers have shown the corrugation may also affect the electronic structure of the materials. In the experimental synthesis process, the corrugation may vary and affect the band edge. In the current model, all this information is missing.
4. The HOMO-LUMO gap of a molecule or small cluster cannot be called band gap, as there is no band. If the authors have read the literature that I suggest, they should know Mn doping would introduce electrons into the conduction band of C₃N₄, which is an electron injection doping process. And such a band structure would have a broad light adsorption spectrum, which is in line with the authors' experimental observations. However, the model used by the authors didn't have such behavior.
5. The EPR data cannot rule out the process for the excitation of Mn²⁺ itself, where its spin changes from 5/2 to 3/2 (J. Phys. Chem. Lett. 2020, 11, 9587–9595). As Mn²⁺ is used for red light emitting, it may also have the electronic structure to adsorb the light of 660 nm here. And the authors' model didn't consider this process in the calculations or experimental data analysis.
6. Raman spectra of C₃N₄-MnO₂ but not MnO₂ should be presented in Supplementary Figure 12 for comparing.
7. The treatment of VB XPS data in Figure 3B seems to be arbitrary, a little variation on the drawing of the lines may change the results a lot. And the authors didn't tell people why the values in Figure 3B are different in Figure 3C, where all data shift with a constant. In current condition, it seems the treatment would like to match the statement of C₃N₄ not generating ·OH.

Above all, the DFT calculations by the authors cannot help to explain the experiments, as it is far from the experimental data from the authors and a lack of physical meaning. The current work may be far from the physics in the whole process. And the experimental data is some kind of treating with bias. Rigorously, I would recommend to reject the current manuscript for those critical issues.

Reviewer #3:

Remarks to the Author:

The authors have done a substantial amount of work in their revision, and I am satisfied with most of the answers given to my comments, especially the supplementary experiments showing that in non-cancer cells with less amount of GSH the mechanism of reduction is not activated. Also, I appreciated the experiments in the dark stability and stability in DMEM, as well as the better substantiation of the mechanism and the fate of the H from water.

Two issues still remain before the manuscript is ready for acceptance.

1. As shown recently in the literature (the manuscript by Bonet et al. "Unveiling the interplay between homogeneous and heterogeneous catalytic mechanisms..." *Chemical Science* 2022, 13, 8307) while the GSH levels are 2 to 3 times higher in cancer cells than in healthy cells, the GSH levels in healthy cells are not negligible. This means that in situ synthesis of the SAC should also occur to a certain extent in healthy cells. The authors may argue that a certain level of GSH is needed to trigger the self assembly, but this should be proven with additional experiments. This discussion (about how much GSH triggers the synthesis) needs to be incorporated and the paper of Bonet et al. cited to compare the levels of GSH.
2. The biodistribution in fig Suppl. 38 is welcome, but is incomplete. We do not know how many micrograms of Mn were injected in all, and how much ended in each organ. The distribution given only represents the concentration at each organ, but is misleading, what is important is to know actually what percentage of the injected catalyst ends up in the tumor, and compare it to the percentages in other organs. Related to this, the relatively low arrival to the liver should be discussed, as this is normally the organ where 90%+ of the nanoparticles end.

After these remaining issues are satisfactorily addressed the paper can be accepted for publication.

Reply to the Reviewer 1's Comments

Reviewer #1 (Remarks to the Author):

Authors have clearly addressed my concern. I would like to recommend to accept the manuscript at current form.

Response:

Thank you very much for the positive comments from the Reviewer.

Reply to the Reviewer 2's Comments

Reviewer #2 (Remarks to the Author):

The authors have partially addressed my concerns, but not totally. It seems that they didn't go into the literature that I suggested, or they would find critical issues on their calculations. After going through the revised manuscript, I have the following comments.

Response:

Thank you very much for the valuable comments from the Reviewer. According to the comments and the suggested reference (*J. Phys. Chem. C*, 2020, 124, 4644-4651), the whole section of calculations has been recalculated upon replacing the original model of C_3N_4 with quite small corrugation molecular model into the suggested larger corrugation periodical model. Therefore, the electronic structures of the materials and corresponding energies have also been updated. The corresponding revisions are shown in the subsequent responses.

1. Figure 1e should present the EDS pattern of O, just as being done in Supplementary Figure 4.

Response:

As suggested, the EDS pattern of O has been added in Figure 1E. The revised figure is shown as follows:

(Figure 1, Line 113-118, Page 8, Highlighted Revised Manuscript):

Fig. 1. The synthesis of C_3N_4 -Mn SACs by the reduction of C_3N_4 - MnO_2 by GSH and the morphology characterizations. (A) The mechanism illustration. MnO_2 was reduced into Mn^{2+} by GSH, and Mn^{2+} was captured by C_3N_4 nanosheets via the coordination into the N_6 -cavity. (B) TEM images of C_3N_4 - MnO_2 . (C) AFM images and the heights of C_3N_4 nanosheets, C_3N_4 - MnO_2 precursor, and C_3N_4 -Mn SACs. (D) TEM images of C_3N_4 -Mn SACs. (E) EDS mapping of C_3N_4 -Mn SACs. (F) HAADF-STEM image of C_3N_4 -Mn SACs. The atomically dispersed Mn was highlighted by red circles.

2. The authors stated Mn-N bond length should mostly be 1.97 Å (Supplementary Table 1), while in their model Mn-N bond length is no shorter than 2.5 Å (Supplementary Table 2). Such a large difference would greatly affect the results of DFT calculations. Also, their model is not consistent with the CN value of 3.5 in Supplementary Table 1. It is hard to know why the calculated data should match so well with the experimental data as the authors showed. Maybe it just happens to be some random event there to have such matching.

Response:

Just as the Reviewer indicated, the calculated data didn't match well with the experimental data, due to the configuration of the unsuitable model of C_3N_4 with quite a small corrugation. As suggested by the Reviewer, a larger corrugation model of C_3N_4 was constructed for the calculations based on the suggested reference (*J. Phys. Chem. C*, 2020, 124, 4644-4651). In the new model, the Mn atom was coordinated to four pyridine-N atoms, instead of located in the center of the cavity. The calculated Mn-N bond length was updated to about 2.2 Å and the CN value was calculated to be 4, matching better with the experimental data.

The corresponding revisions are shown as follows:

(Line 174-179, Page 10-11, Highlighted Revised Manuscript):

... As a result, the Mn atom was coordinated to four pyridine-N atoms (with $d_{Mn-N} = 2.20, 2.23, 2.25,$ and 2.34 Å, respectively), instead of being located in the center of the cavity (the distance from N was about 2.50 Å). This was in accordance with the quantitative EXAFS curve fitting analysis of Mn-N, which exhibited around 3.5 of the Mn-N coordination number with the smaller d_{Mn-N} (Supplementary Table 1). Consequently, Mn with a lower atomic radius is preferably coordinated to the four nearby pyridine-N instead of located at the center of the C_3N_4 cavity upon uniform interaction with six pyridine-N.³¹ ...

(Supplementary Figure 14, Line 228-231, Page 11, Highlighted Revised Supplementary Information)

Supplementary Figure 14. Top and side views of optimized geometry of (A) C_3N_4 and (B) C_3N_4 -Mn SACs. Atom colors in catalyst: C (brown), N (gray), Mn (purple).

3. If the authors have read the literature that I suggest, they should know the corrugation of g- C_3N_4 should be

much larger than the model they used. And in the literature, the researchers have shown the corrugation may also affect the electronic structure of the materials. In the experimental synthesis process, the corrugation may vary and affect the band edge. In the current model, all those information is missing.

Response:

We appreciate the valuable comments from the Reviewer. According to the comment, we have carefully studied the suggested reference (*J. Phys. Chem. C*, 2020, 124, 4644-4651) and reconstructed the model from the original model of C_3N_4 with quite small corrugation into the suggested larger corrugation model. Just as the Reviewer indicated, the corrugation affected the electronic structure of the materials and the band edges. As a result, the thickness of C_3N_4 and C_3N_4 -Mn was 3.99 Å and 4.16 Å respectively, being much larger than the original near planar ones. Meanwhile, Mn was not in the center of the cavity, resulting in shorter Mn-N bonds to well match the EXAFS results. Consequently, all the electronic structures of the materials and corresponding band edges have also been updated in the revised manuscript.

The corresponding revisions are shown in the Highlighted Revised Manuscript as follows:

(Supplementary Figure 14, Line 228-231, Page 11, Highlighted Revised Supplementary Information)

Supplementary Figure 14. Top and side views of optimized geometry of (A) C_3N_4 and (B) C_3N_4 -Mn SACs. Atom colors in catalyst: C (brown), N (gray), Mn (purple).

4. The HOMO-LUMO gap of a molecule or small cluster cannot be called band gap, as there is no band. If the authors have read the literature that I suggest, they should know Mn doping would introduce electrons into the conduction band of C_3N_4 , which is an electron injection doping process. And such a band structure would have

a broad light adsorption spectrum, which is in line with the authors' experimental observations. However, the model used by the authors didn't have such behavior.

Response:

We really appreciate the valuable suggestions on the calculations. Based on the Reviewer's comment, we carefully studied the suggested literature (*J. Phys. Chem. C*, 2020, 124, 4644-4651, *Front. Phys.*, 2018, 13, 138108) and re-analyzed the electronic structure of the C_3N_4 and C_3N_4 -Mn. Just as the Reviewer indicated, no band can be fabricated upon the HOMO-LUMO gap of a molecule or small cluster model and the analysis of the HOMO and LUMO cannot reflect the actual electronic structure in the C_3N_4 -Mn. Therefore, the band structure and DOS of the material were analyzed to accurately analyze the electronic structures in the new periodical structure. The DOS results indicated the decrease of the band gap upon the insertion of Mn's low-energy empty orbitals into the conduction band of C_3N_4 . This was further in accordance with the broad light absorption of the materials and the LMCT process (Figure 3). Consequently, the band structures of C_3N_4 and C_3N_4 -Mn have been re-calculated in the revised manuscript and the corresponding demonstrations on HOMO and LUMO have been deleted.

The corresponding revisions are shown as follows:

(Line 215-230, Page 13, Highlighted Revised Manuscript):

To evaluate the band structure for supporting the improved photocatalytic oxidation by C_3N_4 -Mn, the calculations of C_3N_4 and C_3N_4 -Mn were employed with the first-principles simulations based on the density functional theory level.²⁸ As calculated by the Perdew-Burke-Ernzerhof (PBE) method, the Mn is completely spin-polarized and in a high spin state (Supplementary Fig. 18). To avoid the underestimating of the band gap by GGA calculations, the band gaps of C_3N_4 and C_3N_4 -Mn SACs were recalculated using the Heyd-Scuseria-Ernzerhof (HSE06) hybrid functional.^{29, 34} As shown in Supplementary Fig. 19, the band gap of C_3N_4 -Mn (2.57 eV) was determined to be lower than that of C_3N_4 (2.97 eV), which was in accordance with the band energy obtained in Fig. 3C. With the decreased band gap of C_3N_4 -Mn, the absorption of photons would be enhanced when exposed to red light irradiation (660 nm), which was consistent with the computed light-absorption spectra (Supplementary Fig. 20). Furthermore, after the coordination of Mn^{2+} with the pyridine-N atoms, the valence band (VB) level was 1.66 eV lower than C_3N_4 , which led to a significant lower VBM of C_3N_4 -Mn (-7.33 eV

versus vacuum energy level) compared to the standard oxidation potential of $\text{H}_2\text{O}/\cdot\text{OH}$ (-6.49 eV versus vacuum energy level) (Supplementary Fig. 19). This would further confirm the oxidation capacity of the photo-generated holes. Thus, with Mn coordinated with C_3N_4 , the band gap was dramatically decreased and photocatalytic oxidation capacity was significantly improved, which would consequently facilitate the efficient PDT process.

(Line 235-254, Page 13-14, Highlighted Revised Manuscript):

... For a deeper comprehension of the charge transfer under light irradiation, the density of states (DOS) for both C_3N_4 and $\text{C}_3\text{N}_4\text{-Mn}$ were determined by the DFT calculations. As a result, a number of new energy levels appeared below the CB of C_3N_4 once Mn^{2+} bound to the N atoms of C_3N_4 , which were attributed to the hybridization of Mn with the C_3N_4 (Fig. 3E, Supplementary Fig. 21). Considering the d orbital of Mn is the primary contributor to the CB of $\text{C}_3\text{N}_4\text{-Mn}$, the decrease of the band gap could be attributed to the introduction of the lower empty orbital of Mn. Simultaneously, the VB of $\text{C}_3\text{N}_4\text{-Mn}$ experienced a downward shift, mostly influenced by the p orbitals of the sp^2 -hybridized N. The occupied d orbitals of Mn exhibited a low energy level, potentially attributed to the coordination influence of C_3N_4 . This resulted in a large energy need (> 3 eV) for the excitation of Mn^{2+} itself (with spin changes from 5/2 to 3/2), thereby preventing any influence on the variations in the EPR signal induced by Mn's excitation. Therefore, based on the EPR results and calculation of $\text{C}_3\text{N}_4\text{-Mn}$, this electronic excitation from VB (sp^2 -hybridized N) to CB (Mn) corresponds to the typical LMCT mechanism. Furthermore, to provide a more detailed illustration of the LMCT process, the photoexcited charge density transition from VB to CB of $\text{C}_3\text{N}_4\text{-Mn}$ was examined. As presented in Supplementary Fig. 22, the excited electrons in C_3N_4 (sp^2 -hybridized N) were transferred to the d orbital of Mn, illustrating the characteristic photo-induced LMCT process. Notably, compared with the electron excitation process from C to N in C_3N_4 (Fig. 3E, Supplementary Fig. 21),³⁷ this LMCT process in $\text{C}_3\text{N}_4\text{-Mn}$ would further enhance the separation of electrons and holes. This would facilitate the construction of the photoexcited charge-separation states, which are crucial for an efficient photocatalytic process. ...

(Figure 3, Line 287-296, Page 17, Highlighted Revised Manuscript):

Fig. 3. Examinations on $\cdot\text{OH}$ generation by $\text{C}_3\text{N}_4\text{-Mn}$ SACs under light irradiation. (A) UV-Vis absorption of C_3N_4 and $\text{C}_3\text{N}_4\text{-Mn}$ SACs. The experiment was repeated three times independently with similar results. (B) VB-XPS spectra of C_3N_4 and $\text{C}_3\text{N}_4\text{-Mn}$ SACs. (C) Band position of C_3N_4 and $\text{C}_3\text{N}_4\text{-Mn}$ SACs versus NHE. The band position versus NHE was calculated according to the following formula: $E_{\text{NHE}} = \phi + E_{\text{XPS}} - 4.44$ (NHE level), where ϕ is the work function of the instrument (4.2 eV). (D) EPR spectra of $\text{C}_3\text{N}_4\text{-Mn}$ SACs before and after light irradiation. Density of states (DOS) of (E) $\text{C}_3\text{N}_4\text{-Mn}$ and (F) C_3N_4 obtained by DFT calculations. (G) EPR spectra of $\text{C}_3\text{N}_4\text{-Mn}$ SACs solution before and after the light irradiation for 30 min (660 nm, 0.4 W/cm²). DMPO acted as the trapping agent. (H) The LMCT-based photocatalytic generation of $\cdot\text{OH}$ by water splitting over $\text{C}_3\text{N}_4\text{-Mn}$. A charge reparation state was formed to produce $\cdot\text{OH}$ via O-H homolysis.

(Supplementary Figure 18, Line 372-375, Page 18, Highlighted Revised Supplementary Information)

Supplementary Figure 18. PBE calculated band diagram of C_3N_4 (left) and C_3N_4 -Mn (right). The gray dashed lines represent the Fermi energy level.

(Supplementary Figure 19, Line 377-382, Page 19, Highlighted Revised Supplementary Information)

Supplementary Figure 19. HSE06 calculated band diagram of C_3N_4 (left) and C_3N_4 -Mn (right). The gray dashed lines represent the Fermi energy level, and the red dashed line represents the oxidation potential of H_2O/OH .

(Supplementary Figure 20, Line 384-387, Page 20, Highlighted Revised Supplementary Information)

Supplementary Figure 20. The light-absorption spectra of C_3N_4 and C_3N_4 -Mn based on the PBE level of theory.

(Supplementary Figure 21, Line 389-392, Page 22, Highlighted Revised Supplementary Information)

Supplementary Figure 21. PBE calculated partial density of states of C_3N_4 -Mn and C_3N_4 .

(Supplementary Figure 22, Line 393-395, Page 23, Highlighted Revised Supplementary Information)

Supplementary Figure 22. PBE calculated photoexcited charge density transition from VB to CB of C_3N_4 -Mn, indicating the LMCT process. The yellow bubble represents the electron population.

5. The EPR data cannot rule out the process for the excitation of Mn^{2+} itself, where its spin changes from $5/2$ to $3/2$ (J. Phys. Chem. Lett. 2020, 11, 9587–9595). As Mn^{2+} is used for red light emitting, it may also have the electronic structure to adsorb the light of 660 nm here. And the authors' model didn't consider this process in the calculations or experimental data analysis.

Response:

Just as the Reviewer indicated, the excitation of Mn^{2+} could be generated from changes of electronic structures upon absorbing red light at 660 nm, as demonstrated by the suggested reference (J. Phys. Chem. Lett. 2020, 11, 9587–9595). Consequently, to exclude the self-excitation of Mn^{2+} under red light irradiation, additional calculations have been carried out upon the reconstruction of the suggested larger corrugation model. As demonstrated by the DOS analysis, the electron energy of Mn itself is very low, which requires more than 3 eV for the excitation of Mn itself (Supplementary Fig. 21). This excitation energy is much higher than the energy of the red light at 660 nm (about 1.9 eV). In fact, this low energy of Mn was attributed to the coordination effect of C_3N_4 , which was demonstrated by the disappearance of the catalytic effect in the absence of light (Supplementary Figure 31).

Consequent, the corresponding revisions are shown in the Highlighted Revised Manuscript as follows:

(Line 239-247, Page 13-14, Highlighted Revised Manuscript):

... Considering the d orbital of Mn is the primary contributor to the CB of $\text{C}_3\text{N}_4\text{-Mn}$, the decrease of the band gap could be attributed to the introduction of the lower empty orbital of Mn. Simultaneously, the VB of $\text{C}_3\text{N}_4\text{-Mn}$ experienced a downward shift, mostly influenced by the p orbitals of the sp^2 -hybridized N. The occupied d orbitals of Mn exhibited a low energy level, potentially attributed to the coordination influence of C_3N_4 . This resulted in a large energy need (> 3 eV) for the excitation of Mn^{2+} itself (with spin changes from 5/2 to 3/2), thereby preventing any influence on the variations in the EPR signal induced by Mn's excitation. Therefore, based on the EPR results and calculation of $\text{C}_3\text{N}_4\text{-Mn}$, this electronic excitation from VB (sp^2 -hybridized N) to CB (Mn) corresponds to the typical LMCT mechanism. ...

6. Raman spectra of $\text{C}_3\text{N}_4\text{-MnO}_2$ but not MnO_2 should be presented in Supplementary Figure 12 for comparing.

Response:

As suggested by the Reviewer, the additional experiments on collecting the Raman spectra of $\text{C}_3\text{N}_4\text{-MnO}_2$ have been carried out and added in Supplementary Figure 12 as follows:

(Supplementary Figure 12, Line 204-205, Page 9-10, Highlighted Revised Supporting Information):

Supplementary Figure 12. Raman spectra of MnO_2 , $\text{C}_3\text{N}_4\text{-MnO}_2$, and $\text{C}_3\text{N}_4\text{-Mn}$ SACs.

7. The treatment of VB XPS data in Figure 3B seems to be arbitrary, a little variation on the drawing of the lines may change the results a lot. And the authors didn't tell people why the values in Figure 3B are different in Figure 3C, where all data shift with a constant. In current condition, it seems the treatment would like to match the statement of C_3N_4 not generating $\cdot\text{OH}$.

Response:

We are sorry for the unclear descriptions on the demonstration of VB-XPS data (Figure 3B) and the obtaining of band position based on the XPS data (Figure 3C). In fact, the line in Figure 2B was obtained based on the tangent line of the experimental XPS data. In addition, the band position of both C_3N_4 and $\text{C}_3\text{N}_4\text{-Mn}$ in Figure 3C (the potential versus NHE) was obtained according to the data of VB-XPS in Figure 3B (the potential obtained by XPS analysis). The potentials versus NHE (E_{NHE}) in Figure 3C were calculated according to the

following formula: $E_{\text{NHE}} = \varphi + E_{\text{XPS}} - 4.44$ (NHE level), where φ was the work function of the instrument (4.2 eV). Consequently, the obtained E_{NHE} values of both C_3N_4 and $\text{C}_3\text{N}_4\text{-Mn}$ were 0.24 eV lower than the corresponding VB-XPS data in Figure 2B. Actually, the Y-axis values in Figure 2C were kinds of relative data, which normally presented a constant shift during the conversion of VB-XPS data into band positions (*Nat. Commun.* 2023, 14, 7115, *Adv. Energy Mater.* 2019, 9, 1901505, *Adv. Funct. Mater.* 2023, 33, 2302824). Consequently, the band position was just relative data, which was mainly used to indicate the lowest energy position was higher or lower than the corresponding data of $\text{H}_2\text{O}/\cdot\text{OH}$. As a result, with atomically dispersed Mn^{2+} captured by C_3N_4 , the band position became lower than that of $\text{H}_2\text{O}/\cdot\text{OH}$, further confirming the possibility of generating $\cdot\text{OH}$ for the subsequent therapy. This was also in accordance with the newly calculated band diagram (Figure S18 in the Revised Supporting Information), obtained with the suggested larger corrugation model of C_3N_4 .

Therefore, the corresponding explanation was added in the legend of Figure 3 for better demonstration. The revisions are shown as follows:

(Figure 3, Line 287-296, Page 17, Highlighted Revised Manuscript):

Fig. 3. Examinations on $\cdot\text{OH}$ generation by $\text{C}_3\text{N}_4\text{-Mn}$ SACs under light irradiation. (A) UV-Vis absorption of C_3N_4 and $\text{C}_3\text{N}_4\text{-Mn}$ SACs. The experiment was repeated three times independently with similar results. (B) VB-XPS spectra of C_3N_4 and $\text{C}_3\text{N}_4\text{-Mn}$ SACs. (C) Band position of C_3N_4 and $\text{C}_3\text{N}_4\text{-Mn}$ SACs versus NHE. The band position versus NHE was calculated according to the following formula: $E_{\text{NHE}} = \varphi + E_{\text{XPS}} - 4.44$ (NHE level), where φ is the work function of the instrument (4.2 eV). (D) EPR spectra of $\text{C}_3\text{N}_4\text{-Mn}$ SACs before and after light irradiation. Density of states (DOS) of (E) $\text{C}_3\text{N}_4\text{-Mn}$ and (F) C_3N_4 obtained by DFT calculations. (G) EPR spectra of $\text{C}_3\text{N}_4\text{-Mn}$ SACs solution before and after the light irradiation for 30 min (660 nm, 0.4 W/cm²). DMPO acted as the trapping agent. (H) The LMCT-based photocatalytic generation of $\cdot\text{OH}$ by water splitting over $\text{C}_3\text{N}_4\text{-Mn}$. A charge reparation state was formed to produce $\cdot\text{OH}$ via O-H homolysis.

Above all, the DFT calculations by the authors cannot help to explain the experiments, as it is far from the experimental data from the authors and a lack of physical meaning. The current work may be far from the physics in the whole process. And the experimental data is some kind of treating with bias. Rigorously, I would recommend to reject the current manuscript for those critical issues.

Response:

Thank you very much for the valuable comments on the calculations. Just as the Reviewer indicated, the original model of C_3N_4 with quite small corrugation was not suitable for the examinations, which led to the experimental data were not in satisfactory agreement with the calculations. Therefore, in the revised works, we have tried our best to revise the whole calculation section after careful studies on the suggested reference (*J. Phys. Chem. C*, 2020, 124, 4644-4651). Excitingly, based on the new DFT calculations with the suggested larger corrugation model of C_3N_4 , we successfully updated the electronic structures of the materials and corresponding energy data, which were in accordance with the corresponding experimental data. Based on the comments from the Reviewer, we think the revised manuscript is substantially improved and we hope the Reviewer could be satisfied with our revisions.

Reply to the Reviewer 3's Comments

Reviewer #3 (Remarks to the Author):

The authors have done a substantial amount of work in their revision, and I am satisfied with most of the answers given to my comments, especially the supplementary experiments showing that in non-cancer cells with less amount of GSH the mechanism of reduction is not activated. Also, I appreciated the experiments in the dark stability and stability in DMEM, as well as the better substantiation of the mechanism and the fate of the H from water.

Response:

We really appreciate the positive comments from the Reviewer.

Two issues still remain before the manuscript is ready for acceptance.

1. As shown recently in the literature (the manuscript by Bonet et al. "Unveiling the interplay between homogeneous and heterogeneous catalytic mechanisms..." *Chemical Science* 2022, 13, 8307) while the GSH levels are 2 to 3 times higher in cancer cells than in healthy cells, the GSH levels in healthy cells are not negligible. This means that in situ synthesis of the SAC should also occur to a certain extent in healthy cells. The authors may argue that a certain level of GSH is needed to trigger the self assembly, but this should be proven with additional experiments. This discussion (about how much GSH triggers the synthesis) needs to be incorporated and the paper of Bonet et al. cited to compare the levels of GSH.

Response:

Thank you very much for the comments from the Reviewer. As suggested by the Reviewer, additional experiments have been carried out to evaluate the intracellular GSH levels in HeLa cells and healthy cells of HUV-EC cells, with and without the C_3N_4 - MnO_2 treated. As shown in Supplementary Figure 35, after C_3N_4 - MnO_2 treated, the GSH levels in HeLa cells exhibited a decrease of about 56 %, whereas the GSH levels in the HUV-EC cells remained unchanged. Just as the Reviewer indicated, this could be generated from the much

lower expression of GSH in healthy cells than in the cancer ones, being in accordance with the suggested report (*Chem. Sci.*, 2022, 13, 8307–8320). Consequently, the corresponding discussion has been added and the suggested reference has been cited in the revised manuscript (Ref. 41).

The corresponding revisions are shown as follows:

(Line 376-381, Page 21, Highlighted Revised Manuscript)

... Afterwards, the corresponding intracellular GSH levels in HeLa cells and healthy HUV-EC cells were measured with and without the presence of $C_3N_4-MnO_2$. After $C_3N_4-MnO_2$ was treated, the GSH levels in HeLa cells exhibited a decrease of about 56 %, whereas the GSH levels in the HUV-EC cells remained unchanged (Supplementary Fig. 36). This could be generated from the much lower expression of GSH in healthy cells than in the cancer cells.⁴¹ ...

(Supplementary Figure 36, Line 517-520, Page 33, Highlighted Revised Supplementary Information)

Supplementary Figure 36. Intracellular GSH concentration levels in HeLa cells and HUV-EC cells before and after 24 h of the incubation with $C_3N_4-MnO_2$. ns: $p > 0.05$, not significant, *: $p \leq 0.05$, **: $p \leq 0.01$, ***: $p \leq 0.001$, ****: $p \leq 0.0001$. $[C_3N_4] = 10 \mu\text{g/mL}$.

2. The biodistribution in fig Suppl. 38 is welcome, but is incomplete. We do not know how many micrograms of Mn were injected in all, and how much ended in each organ. The distribution given only represents the concentration at each organ, but is misleading, what is important is to know actually what percentage of the injected catalyst ends up in the tumor, and compare it to the percentages in other organs. Related to this, the relatively low arrival to the liver should be discussed, as this is normally the organ where 90%+ of the nanoparticles end.

Response:

Just as the Reviewer indicated, the percentage of the injected catalysts should be provided, rather than how many micrograms of Mn. Therefore, according to the comment, Supplementary Figure 41 (Supplementary Figure 38 in the previous manuscript) has been revised by changing the Y-axis unit from the mass concentration ($\mu\text{g/g}$) into the percentage (ID%/g).

The corresponding revisions are shown as follows:

(Supplementary Figure 41, Line 552-554, Page 37, Highlighted Revised Supplementary Information)

Supplementary Figure 41. The biodistribution of $C_3N_4-MnO_2$ by measuring Mn concentrations over 12, 24, and 48 h after the intravenous injection.

In addition, the discussion on the relatively low biodistribution at liver was added in the revised manuscript as follows:

(Line 558-562, Page 29, Highlighted Revised Manuscript):

... Furthermore, the relatively lower biodistribution at the liver (Supplementary Fig. 41) could be generated from the structural features of $C_3N_4-MnO_2$ (such as thinness, flexibility, and high dispersibility) for passing through the glomerular filtration barrier (DFB) and accumulating in the kidney.⁵⁰ Thereby, $C_3N_4-MnO_2$ could be rapidly eliminated by passing the renal filtration, decreasing the accumulation in normal tissues to avoid toxic side effects.⁵¹ ...

After these remaining issues are satisfactorily addressed the paper can be accepted for publication.

Response:

Thank you very much for the Reviewer's comments. We have made a careful revision based on the comments from the Reviewer and we hope the Reviewer is satisfied with our improved manuscript.

Reviewers' Comments:

Reviewer #2:

Remarks to the Author:

The authors have addressed my concerns, while the details should be improved.

1. The authors stated that MnO₂ was reduced to Mn²⁺, but there are still Oxygen signals well distributed with Mn signals in Fig 1E. Please explain this phenomenon.
2. There is no Mn for C₃N₄ in Fig. 3F, so the legend should be modified.
3. Poisson's equations were solved under periodic conditions in current models, and the eigenvalues from the DFT calculations cannot be compared directly between different models. The authors should get the energy level in vacuum layer of the model, and align the eigenvalues with that value. This is widely used for the calculation of work-function for a 2D-material. It can also be adopted here for Fig. S18 and S19, since the oxidation potential of H₂O/OH used in the figure is versus vacuum zero level (but the energy bands are not).

Reviewer #3:

Remarks to the Author:

The authors have done a thorough revision and the manuscript has been improved. I have no further comments

Reply to the Reviewer 2's Comments

Reviewer #2 (Remarks to the Author):

The authors have addressed my concerns, while the details should be improved.

Response:

We really appreciate the positive comments from the Reviewer.

1. The authors stated that MnO_2 was reduced to Mn^{2+} , but there are still Oxygen signals well distributed with Mn signals in Fig 1E. Please explain this phenomenon.

Response:

Just as the Reviewer indicated, oxygen signals were observed in the EDS mapping of C_3N_4 -Mn SACs. To examine the distribution of oxygen along with Mn signals, additional experiments on collecting O 1s XPS spectra of C_3N_4 , C_3N_4 - MnO_2 , and C_3N_4 -Mn SACs have been carried out. As demonstrated (the new figures of Supplementary Fig 13G to I), significant signals of absorbed oxygen were observed (at 532.3 eV) in the O 1s XPS spectra, which indicated the oxygen signals could be attributed to the absorbed oxygen in these species. The absorbed oxygen could be introduced during the chemical exfoliation of bulk C_3N_4 into g- C_3N_4 . Besides, the oxygen signal of metal oxides at 528-531 eV was recorded in the O 1s XPS spectrum of C_3N_4 - MnO_2 (Supplementary Fig. 13H), while was absent in that of C_3N_4 -Mn SACs (Supplementary Fig. 13I). This further confirmed the absence of MnO or MnO_2 in the C_3N_4 -Mn SACs, which was in accordance with the *in situ* synthesis of C_3N_4 -Mn SACs upon reduction of MnO_2 into Mn^{2+} .

Consequently, as suggested by the Reviewer, the corresponding explanations on this phenomenon have been added in the revised manuscript as follows:

(Line 144-147, Page 7-8, Highlighted Revised Manuscript):

... Besides, demonstrated by the O 1s XPS spectra (Supplementary Fig. 13), the observed O species in the EDS mapping of C_3N_4 -Mn (Fig. 1E) were attributed to the absorbed oxygen of C_3N_4 . In addition, the absent

oxygen signal of metal oxides in C_3N_4 -Mn SACs (Supplementary Fig. 13) further confirmed the *in situ* synthesis of C_3N_4 -Mn SACs upon the reduction of MnO_2 into Mn^{2+}

(Supplementary Figure 13, Line 188-191, Page 9-10, Highlighted Revised Supplementary Information)

Supplementary Figure 13. XPS spectra of g- C_3N_4 (A), $C_3N_4-MnO_2$ (B), and C_3N_4-Mn (C). Mn 2p spectra of $C_3N_4-MnO_2$ (D), C_3N_4-Mn SACs (E), and C 1s spectra of C_3N_4 , $C_3N_4-MnO_2$ and C_3N_4-Mn (F), O 1s spectra of C_3N_4 (G), $C_3N_4-MnO_2$ (H), and C_3N_4-Mn (I) at high resolution.

(Line 192-206, Page 10, Highlighted Revised Supplementary Information):

As demonstrated in Supplementary Fig. 12, the bands of 565 and 637 cm^{-1} (MnO_2) are slightly shifted in the Raman spectra of $C_3N_4-MnO_2$, showing the significant interaction between C_3N_4 and MnO_2 .⁹ As shown in Supplementary Fig. 13, the peaks of Mn(III) ($Mn 2p_{1/2}$ at 653.1 eV and $Mn 2p_{3/2}$ at 641.5 eV)¹⁰ and Mn(IV) ($Mn 2p_{1/2}$ at 655.7 eV and $Mn 2p_{3/2}$ at 643.8 eV)^{11, 12} were recorded in $C_3N_4-MnO_2$ (Supplementary Fig. 13D). Besides, with MnO_2 deposited on C_3N_4 , the characteristic peaks of N-C₃ (400.5 eV), C-N=C (398.7 eV) as well as C-N=C

(288.2 eV) in C_3N_4 (Supplementary Fig. 13F) shifted to higher binding energies. This could be due to the electron transfer from C_3N_4 to MnO_2 .¹³ As demonstrated (Supplementary Fig. 13G to I), significant signals of absorbed oxygen were observed (at 532.3 eV) in the O 1s XPS spectra, which indicated the oxygen signals could be attributed to the absorbed oxygen in these species. The absorbed oxygen could be introduced during the chemical exfoliation of bulk C_3N_4 into g- C_3N_4 . In addition, the oxygen signal of metal oxides at 528-531 eV was recorded in the O 1s XPS spectrum of C_3N_4 - MnO_2 (Supplementary Fig. 13H), while was absent in that of C_3N_4 -Mn SACs (Supplementary Fig. 13I). This further confirmed the absence of MnO or MnO_2 in the C_3N_4 -Mn SACs, which was in accordance with the *in situ* synthesis of C_3N_4 -Mn SACs upon reduction of MnO_2 into Mn^{2+} .¹⁴

2. There is no Mn for C_3N_4 in Fig. 3F, so the legend should be modified.

Response:

We are sorry for the mistakes in the legend of Figure 3F. As suggested, Figure 3F has been corrected as follows:

(Figure 3, Line 278-286, Page 15, Highlighted Revised Manuscript):

Fig. 3. Examinations on $\cdot\text{OH}$ generation by $\text{C}_3\text{N}_4\text{-Mn}$ SACs under light irradiation. (A) UV-Vis absorption of C_3N_4 and $\text{C}_3\text{N}_4\text{-Mn}$ SACs. The experiment was repeated three times independently with similar results. (B) VB-XPS spectra of C_3N_4 and $\text{C}_3\text{N}_4\text{-Mn}$ SACs. (C) Band position of C_3N_4 and $\text{C}_3\text{N}_4\text{-Mn}$ SACs versus NHE. The band position versus NHE was calculated according to the following formula: $E_{\text{NHE}} = \phi + E_{\text{XPS}} - 4.44$ (NHE level), where ϕ is the work function of the instrument (4.2 eV). (D) EPR spectra of $\text{C}_3\text{N}_4\text{-Mn}$ SACs before and after light irradiation. Density of states (DOS) of (E) $\text{C}_3\text{N}_4\text{-Mn}$ and (F) C_3N_4 obtained by DFT calculations. (G) EPR spectra of $\text{C}_3\text{N}_4\text{-Mn}$ SACs solution before and after the light irradiation for 30 min (660 nm, 0.4 W/cm²). DMPO acted as the trapping agent. (H) The LMCT-based photocatalytic generation of $\cdot\text{OH}$ by water splitting over $\text{C}_3\text{N}_4\text{-Mn}$. A charge reparation state was formed to produce $\cdot\text{OH}$ via O-H homolysis.

3. Poisson's equations were solved under periodic conditions in current models, and the eigenvalues from the DFT calculations cannot be compared directly between different models. The authors should get the energy level in vacuum layer of the model, and align the eigenvalues with that value. This is wildly used for the

calculation of work-function for a 2D-material. It can also be adopted here for Fig. S18 and S19, since the oxidation potential of H₂O/·OH used in the figure is versus vacuum zero level (but the energy bands are not).

Response:

We appreciate the valuable comments from the Reviewer. According to the comment, additional calculations have been carried out to get the energy level in the vacuum layer of each model, and then align the eigenvalues with that value for the re-evaluation of energies versus vacuum levels.

The corresponding sections have been updated as follows:

(Line 225-229, Page 11-12, Highlighted Revised Manuscript):

... Furthermore, after the coordination of Mn²⁺ with the pyridine-N atoms, the valence band (VB) level was **1.73 eV** lower than C₃N₄, which led to a significantly lower VBM of C₃N₄-Mn (**-8.83 eV** versus vacuum energy level) compared to the standard oxidation potential of H₂O/·OH (-6.49 eV versus vacuum energy level) (Supplementary Fig. 19). ...

(Line 320-323, Page 17, Highlighted Revised Manuscript):

The mechanism of water splitting (Fig. 3G) was further explored using DFT calculations. The feasibility of hydrogen transfer by C₃N₄-Mn SACs was demonstrated by the lower hole energy (VBM level) of C₃N₄-Mn SACs (**-8.83 eV** versus vacuum energy level) than in C₃N₄ (**-7.10 eV** versus vacuum energy level) (Supplementary Fig. 19). ...

(Supplementary Figure 18, Line 232-234, Page 15, Highlighted Revised Supplementary Information)

Supplementary Figure 18. PBE calculated band diagram of C_3N_4 (left) and C_3N_4-Mn (right). The gray dashed lines represent the Fermi energy level.

(Supplementary Figure 19, Line 236-239, Page 17, Highlighted Revised Supplementary Information)

Supplementary Figure 19. HSE06 calculated band diagram of C_3N_4 (left) and C_3N_4-Mn (right). The gray dashed lines represent the Fermi energy level, and the red dashed line represents the oxidation potential of $H_2O/\cdot OH$.

Reply to the Reviewer 3's Comments

Reviewer #3 (Remarks to the Author):

The authors have done a thorough revision and the manuscript has been improved. I have no further comments.

Response:

Thank you very much for the positive comments from the Reviewer.

Reviewers' Comments:

Reviewer #2:

Remarks to the Author:

The authors have made their efforts to improve the manuscript. However, there are still some mismatches between the experimental data and DFT results, for example, H₂O/ \cdot OH oxidation potential's position versus VBM in Fig. 3C and Fig. S19. And the DFT results are also not self-consistent while the exchange-correlation functional changes, for example, spin-polarized bands changed to non-polarized ones in Fig. S18 and Fig. S19. Therefore, the authors should be careful with their statements in the manuscript, and double check their calculations.

Reply to the Reviewer 2's Comments

Reviewer #2 (Remarks to the Author):

The authors have made their efforts to improve the manuscript. However, there are still some mismatches between the experimental data and DFT results, for example, H₂O/·OH oxidation potential's position versus VBM in Fig. 3C and Fig. S19. And the DFT results are also not self-consistent while the exchange-correlation functional changes, for example, spin-polarized bands changed to non-polarized ones in Fig. S18 and Fig. S19. Therefore, the authors should be careful with their statements in the manuscript, and double check their calculations.

Response:

We appreciate the valuable comments from the Reviewer. As suggested by the Reviewer, we have made a careful check on the statements in the manuscript and especially double checked the calculation sections. Upon careful evaluation throughout the whole manuscript, we found that the interference of surface charges on the surface of C₃N₄ with high solubility in water could have effect on the simulation. Thereby, the DFT calculations have been modified by adopting the implicit solvent model (VASPsol) to simulate material states in aqueous solutions and rectify energies by compensatory charges. In this way, all the corresponding data of calculations have been recalculated and updated in the Revised Manuscript and Revised Supporting Information, including the calculated band gaps and VBM levels (Figure S18 and S19), DOS (Figure 3C), Charge Density Difference Analysis (Figure S16), computed light-absorption spectra (Figure S20), exciting charge separation state (Figure S21 and S22) and the detailed reaction pathways (Figure S34). Consequently, based on the revised calculations, the experimental data have been in accordance with the DFT results.

The corresponding revisions are highlighted in the Highlighted Revised Manuscript and Highlighted Revised Supporting Information as follows:

(Line 221-229, Page 11-12, Highlighted Revised Manuscript)

... As shown in Supplementary Fig. 19, the band gap of C₃N₄-Mn (2.54 eV) was determined to be lower

than that of C_3N_4 (2.96 eV), which was in accordance with the band energy obtained in Fig. 3C. With the decreased band gap of C_3N_4 -Mn, the absorption of photons would be enhanced when exposed to red light irradiation (660 nm), which was consistent with the computed light-absorption spectra (Supplementary Fig. 20). Furthermore, after the coordination of Mn^{2+} with the pyridine-N atoms, the valence band (VB) level was 0.44 eV lower than C_3N_4 , which led to a significantly lower VBM of C_3N_4 -Mn (-6.84 eV versus vacuum energy level) compared to the standard oxidation potential of $H_2O/\cdot OH$ (-6.49 eV versus vacuum energy level) (Supplementary Fig. 19). ...

(Line 278-286, Page 15, Figure 3, Highlighted Revise Manuscript)

Fig. 3. Examinations on $\cdot OH$ generation by C_3N_4 -Mn SACs under light irradiation. (A) UV-Vis absorption of C_3N_4 and C_3N_4 -Mn SACs. The experiment was repeated three times independently with similar results. **(B)** VB-XPS spectra of C_3N_4 and C_3N_4 -Mn SACs.

(C) Band position of C_3N_4 and C_3N_4 -Mn SACs versus NHE. The band position versus NHE was calculated according to the following formula: $E_{NHE} = \phi + E_{XPS} - 4.44$ (NHE level), where ϕ is the work function of the instrument (4.2 eV). (D) EPR spectra of C_3N_4 -Mn SACs before and after light irradiation. Density of states (DOS) of (E) C_3N_4 -Mn and (F) C_3N_4 obtained by DFT calculations. (G) EPR spectra of C_3N_4 -Mn SACs solution before and after the light irradiation for 30 min (660 nm, 0.4 W/cm²). DMPO acted as the trapping agent. (H) The LMCT-based photocatalytic generation of $\cdot OH$ by water splitting over C_3N_4 -Mn. A charge reparation state was formed to produce $\cdot OH$ via O-H homolysis.

(Line 320-323, Page 17, Highlighted Revised Manuscript)

... The feasibility of hydrogen transfer by C_3N_4 -Mn SACs was demonstrated by the lower hole energy (VBM level) of C_3N_4 -Mn SACs (-6.84 eV versus vacuum energy level) than in C_3N_4 (-6.40 eV versus vacuum energy level) (Supplementary Fig. 19). ...

(Line 104-107, Page 2, Highlighted Revised Supplementary Information)

... To implicitly incorporate the solvation effect, the linearized Poisson Boltzmann model (PBM) was utilized to represent the double layer. This approach has been incorporated into VASPsol and the compensatory charge was represented by adjusting the Debye screening length to 3.04 Å. The relative permittivity was fixed at 78.4 to simulate an aqueous environment.⁷ ...

(Line 112-116, Page 2, Highlighted Revised Supplementary Information)

... The DFT energy change of each elementary step was defined by VASPsol, and the Gibbs free energy was calculated as: $\Delta G = \Delta E + \Delta ZPE - T\Delta S$, where ΔE was the total energy difference obtained by DFT calculation, ΔZPE was the zero-point energy change and $T\Delta S$ represents the energy correction by entropy. T was set to 298.15 K and the pH-induced free energy change was not considered.

(Line 227-230, Page 13, Supplementary Figure 16, Highlighted Revised Supplementary Information)

Supplementary Figure 16. Top and side views of the Charge Density Difference Analysis for the C₃N₄-Mn SACs. The yellow and the cyan areas represent charge accumulation and depletion, respectively. The isosurface value is taken as 0.0025 e*bohr⁻³.

(Line 236-238, Page 16, Supplementary Figure 18, Highlighted Revised Supplementary Information)

Supplementary Figure 18. PBE calculated band diagram of C_3N_4 (left) and C_3N_4-Mn (right). The gray dashed lines represent the Fermi energy level.

(Line 240-243, Page 18, Supplementary Figure 19, Highlighted Revised Supplementary Information)

Supplementary Figure 19. HSE06 calculated band diagram of C_3N_4 (left) and C_3N_4-Mn (right). The gray dashed lines represent the Fermi energy level, and the red dashed line represents the oxidation potential of H_2O/OH .

(Line 245-247, Page 19, Supplementary Figure 20, Highlighted Revised Supplementary Information)

Supplementary Figure 20. The light-absorption spectra of C_3N_4 and C_3N_4 -Mn based on the PBE level of theory.

(Line 249-250, Page 20, Supplementary Figure 21, Highlighted Revised Supplementary Information)

Supplementary Figure 21. PBE calculated partial density of states of C_3N_4 -Mn and C_3N_4 .

(Line 252-255, Page 22, Supplementary Figure 22, Highlighted Revised Supplementary Information)

Supplementary Figure 22. PBE calculated photoexcited charge density transition from VB to CB of C₃N₄-Mn, indicating the LMCT process. The yellow bubble represents the electron population. The isosurface value is taken as 0.02 e*bohr⁻³.

(Line 330-334, Page 30, Supplementary Figure 34, Highlighted Revised Supplementary Information)

Supplementary Figure 34. (A) Energy profiles of photocatalytic water splitting for $\cdot\text{OH}$ generation on the surface of C_3N_4 and $\text{C}_3\text{N}_4\text{-Mn}$, along with the comparison of the reduction of Pyr. Illustrations of reaction pathways of the photocatalytic water splitting over C_3N_4 (B) and $\text{C}_3\text{N}_4\text{-Mn}$ (C). Atom colors in catalyst: C (brown), N (gray), Mn (purple), O (red), H (white).

(Line 335-349, Page 30-31, Revised Supplementary Information)

The detailed reaction path of the water splitting on the surface of C_3N_4 and $\text{C}_3\text{N}_4\text{-Mn}$ was determined. As

shown in Supplementary Fig. 33, C_3N_4 -Mn was easier to adsorb water molecules (with a free energy change of -0.43 eV) than C_3N_4 (with a free energy change of -0.13 eV). This could be generated from the positive charge on Mn, which slightly enhanced the absorption of C_3N_4 -Mn for polar small molecules (water). Clearly, C_3N_4 -Mn had much lower Gibbs free energy for the $\cdot OH$ generation. In detail, at a free energy change of 0.31 eV (-0.43 to -0.12 eV), the homolysis of the O-H bond over C_3N_4 -Mn is much easier than over C_3N_4 (0.63 eV, -0.13 to 0.50 eV). It is the critical step in the generation of $\cdot OH$ and the significant difference indicates that the introduction of Mn can significantly enhance the catalytic capability of $\cdot OH$ generation. Correspondingly, C_3N_4 -H-Mn demonstrates an excellent capability to reduce Pyr, which is confirmed by the lower free energy change (-0.62 eV, 2.93 to 2.31 eV). Therefore, the water splitting catalyzed by C_3N_4 -Mn SACs can generate $\cdot OH$ effectively, and subsequently Pyr can be reduced by C_3N_4 -H-Mn. Consequently, along with maintaining the photocatalytic cycles, cellular respiration of cancer cells can be blocked upon consuming a significant energy substance of Pyr.

Reviewers' Comments:

Reviewer #2:

Remarks to the Author:

The concerns have been addressed. I would like to recommend this manuscript for publication.